EMBO
Molecular Medicine

# Breast cancer secretes anti-ferroptotic MUFAs and depends on selenoprotein synthesis for metastasis

Tobias Ackermann[1,2], Engy Shokry[1], Ruhi Deshmukh [1], Jayanthi Anand [1], Laura C A Galbraith [1], Louise Mitchell[1], Giovanny Rodriguez-Blanco [1], Victor H Villar [1,3], Britt Amber Sterken[1], Colin Nixon [1], Sara Zanivan[1,2], Karen Blyth [1,2], David Sumpton [1] & Saverio Tardito [1,2,4 ✉]

## Abstract

The limited availability of therapeutic options for patients with triple-negative breast cancer (TNBC) contributes to the high rate of metastatic recurrence and poor prognosis. Ferroptosis is a type of cell death caused by iron-dependent lipid peroxidation and counteracted by the antioxidant activity of the selenoprotein GPX4. Here, we show that TNBC cells secrete an anti-ferroptotic factor in the extracellular environment when cultured at high cell densities but are primed to ferroptosis when forming colonies at low density. We found that secretion of the anti-ferroptotic factors, identified as monounsaturated fatty acid (MUFA) containing lipids, and the vulnerability to ferroptosis of single cells depends on the low expression of stearyl-CoA desaturase (SCD) that is proportional to cell density. Finally, we show that the inhibition of Sec-tRNAsec biosynthesis, an essential step for selenoprotein production, causes ferroptosis and impairs the lung seeding of circulating TNBC cells that are no longer protected by the MUFA-rich environment of the primary tumour.

**Keywords** Breast Cancer; Ferroptosis; Lipid Metabolism; Metastasis; Selenium Metabolism
**Subject Categories** Autophagy & Cell Death; Cancer; Metabolism

## Introduction

Ferroptosis is a form of cell death caused by iron-dependent peroxidation of polyunsaturated fatty acids (PUFA) within membranes. Peroxidised lipids cause stiffening and thinning of the membrane bilayer making it prone to rupture (Yang and Stockwell, 2016; Zheng and Conrad, 2020). In recent years, several cell-intrinsic pathways that prevent or revert lipid peroxidation have been discovered. To reduce the free radical species generated from lipidic hydroperoxide, cells rely on lipophilic antioxidants such as tetrahydrobiopterin, 7-dehydrocholesterol, ubiquinone, vitamin K and E, while the membrane-associated peroxidase selenoprotein GPX4, acts through a distinct mechanism that can directly reduce hydroperoxides (Bersuker et al, 2019; Mishima et al, 2022; Yang et al, 2014; Ursini et al, 1985; Freitas et al, 2024; Li et al, 2024; Mao et al, 2021; Soula et al, 2020). In addition, the fatty acid composition of the membrane influences the cellular susceptibility to ferroptosis. The synthesis of saturated fatty acids, the desaturation of PUFAs mediated by Fatty Acid Desaturases 1 and 2 (FADS1/2), and the membrane incorporation of PUFAs mediated by Acyl-CoA Synthetase Long Chain Family Member 4 (ACSL4), have all been shown to promote ferroptosis (Doll et al, 2017; Song et al, 2021; Lorito et al, 2024). Furthermore, exogenous diacyl-PUFA phosphatidylcholines are pro-ferroptotic molecules that accumulate in cell membranes and correlate with cancer cell sensitivity to GPX4 inhibition (Qiu et al, 2024). On the other hand, membrane incorporation of the exogenously supplemented or adipocyte-provided monounsaturated fatty acid (MUFA) oleic acid, as well as its cell-autonomous production mediated by the stearoyl-CoA desaturase (SCD), has been shown to prevent ferroptosis in vitro and vivo (Magtanong et al, 2019; Tesfay et al, 2019; Ubellacker et al, 2020; Xie et al, 2022).

Selenocysteine is a genetically encoded amino acid, and a structural analogue of cysteine with a selenium atom instead of the sulphur one. Proteins that incorporate selenocysteine are defined as selenoproteins. During mRNA translation, selenocysteine is incorporated by a recoding event. If an mRNA contains a selenocysteine insertion sequence (SECIS), the canonical UGA stop codon is translated to selenocysteine. In addition, selenocysteine incorporation into peptides depends on the presence of specific translation factors, the eukaryotic elongation factor selenocysteine (eEFsec) and SECIS-binding protein (SECISBP2) (Labunskyy et al, 2014). Despite the chemical similarity, cysteine and selenocysteine do not share the biosynthetic pathway. In fact, selenocysteine is synthesised on the selenocysteine tRNA (tRNA$^{sec}$) that is firstly charged with serine by seril-tRNA synthetase. The serine on the tRNA$^{sec}$ is then phosphorylated by phosphoseryl-tRNA kinase (PSTK). The selenophosphate (SePO$_3^{(3-)}$) synthesised by the Selenophosphate Synthetase 2 (SEPHS2) is used by O-phosphoseryl-tRNA$^{Sec}$ selenium transferase (SEPSECS) to synthesise selenocysteinilated tRNA$^{sec}$ (Sec-tRNA$^{sec}$). This is the only

[1]Cancer Research UK Scotland Institute, Garscube Estate, Switchback Road, Glasgow G61 1BD, UK. [2]School of Cancer Sciences, University of Glasgow, Glasgow G611QH, UK. [3]Present address: School of Medicine, University of St Andrews, St. Andrews KY16 9TF, UK. [4]Present address: Center for Cancer Research, Medical University of Vienna, Comprehensive Cancer Center, Vienna 1090, Austria. ✉E-mail: saverio.tardito@meduniwien.ac.at

known pathway to produce the Sec-tRNA$^{sec}$ and free selenocysteine obtained from selenoproteins degradation cannot be directly charged onto tRNA$^{sec}$ (Burk and Hill, 2015).

The supplementation with an excess of selenium and the inhibition of Sec-tRNA$^{sec}$ synthesis can both lead to the accumulation of reactive selenium species toxic to cancer cells (Carlisle et al, 2020; Chen et al, 2013). On the other hand, selenium deprivation has been shown to induce ferroptosis in cells from breast cancer, acute myeloid leukaemia, and neuroblastoma (Eagle et al, 2022; Vande Voorde et al, 2019; Alborzinia et al, 2023). Several preclinical and clinical studies assessed selenium supplementation as an antioxidant cancer-preventive intervention and selenium is frequently found in fortified food, and multivitamin/multimineral supplements (Vinceti et al, 2014; National Institutes of Health—Office of Dietary Supplements, 2023).

However, susceptibility to ferroptosis does not only depend on selenium availability. In this study, we show that triple-negative breast cancer (TNBC) cells under selenium starvation die of ferroptosis selectively when seeded at low density. We found that this phenotype can be rescued by transferring medium conditioned by high-density cultures of breast cancer cells or cancer-associated fibroblasts, but not normal cells. By means of analytical methods coupled with mass spectrometry-based lipidomics, we identified SCD-derived MUFA-containing lipids as the anti-ferroptotic factors released by TNBC cells. Finally, we interfered with the expression of genes of the selenocysteine biosynthesis pathway to prove that in vivo TNBC cells require the antioxidant action of selenoproteins to overcome the pro-ferroptotic environment encountered during the metastatic cascade.

# Results

## Medium conditioned by breast cancer cells increases colony formation efficiency of triple-negative breast cancer (TNBC) cells

We have previously shown that TNBC cells seeded at low density are enabled to form colonies by the sodium selenite present in the physiological cell culture medium Plasmax™ (Vande Voorde et al, 2019). To directly visualise if TNBC cells die when seeded at low density, we performed live-cell imaging on MDA-MB-468 cells (Fig. 1A). Four days after seeding, the frequency of cell death was significantly decreased if the medium was supplemented with sodium selenite or Ferrostatin-1, demonstrating that TNBC cells seeded at low density die of ferroptosis impairing their colony formation (Fig. 1B).

However, TNBC cells' susceptibility to ferroptosis has been shown to be cell density-dependent, with cells undergoing ferroptosis selectively when seeded at low density (Vande Voorde et al, 2019; Panzilius et al, 2018; Wu et al, 2019). To dissect the mechanism underpinning the dependency of ferroptosis on culture density, we tested whether the ferroptosis-resistance observed in selenium-restricted cultures at high density (Appendix Fig. S1A) was transferrable with the conditioned medium. The medium conditioned for 48 h by high-density cultures of MDA-MB-468 cells, was employed to seed cells at low density. Similarly to the anti-ferroptotic agents, sodium selenite and Ferrostatin-1, the conditioned medium conferred the capacity to survive ferroptosis

and form colonies to MDA-MB-468 cells (Fig. 1C; Appendix S1B). The medium conditioned by high-density cultures of human BT549, CAL120, MDA-MB-231 and murine EO771 TNBC cells also promoted colony formation in the respective cell lines seeded at low density (Fig. 1D). However, this effect was not observed with the luminal A, ER-positive MCF7 breast cancer cells, for which the number of colonies was not altered by the conditioned medium (Fig. 1D). Moreover, the medium conditioned by BT549, CAL120 and MCF7 cells stimulated MDA-MB-468 colony-forming capacity, demonstrating that the anti-ferroptotic effect of conditioned medium is preserved across cell lines (Fig. 1E).

This observation prompted us to test if conditioned medium from untransformed cells could also support the clonal growth of cancer cells. Cancer-associated fibroblasts (CAFs) are known to support the proliferation of cancer cells in vivo (Sahai et al, 2020; Kojima et al, 2010). Conditioned medium from CAFs increased the colony formation capacity of MDA-MB-468 comparably to the anti-ferroptotic agent sodium selenite (Fig. 1F). Conversely, neither the media conditioned by human normal fibroblasts from the mammary gland (MF), or from the dermis (DF), significantly increased the colony-forming capacity of MDA-MB-468 cells (Fig. 1F). These data suggest that the medium conditioned by transformed or cancer-reprogrammed cells supports clonal growth of cancer cells by preventing ferroptosis.

## Breast cancer cells cultured at high-density produce and secrete an anti-ferroptotic factor

To test whether cells cultured at high-density condition the medium by depleting a ferroptosis-inducing molecule (e.g. iron) or produce a factor that prevents ferroptosis, we collected medium conditioned by high-density MDA-MB-468 cells and diluted with fresh medium (1:3). The diluted conditioned medium enabled colony formation of MDA-MB-468 cells comparably to medium supplemented with the anti-ferroptotic agent Ferrostatin-1 (Fig. 2A). This factor is also produced and secreted by TNBC cells cultured in a sodium selenite-free version of the physiological culture medium Plasmax™ (Vande Voorde et al, 2019) (Fig. 2A). These results suggest that breast cancer cells produce an anti-ferroptotic factor that compensates for the selenium deficiency. To test whether the conditioned medium regulates the expression of known ferroptotic mediators in the recipient cells we measured the mRNA levels of AIFM2, ACSL4, DHFR, DHODH, and protein expression of GPX4 in cells cultured at low density and exposed to the condition medium. However, the expression pattern of these genes did not explain the anti-ferroptotic activity of the conditioned medium (Fig. EV1A–F), suggesting that the secreted factor does not activate a transcriptional anti-ferroptotic response in the recipient cells.

To identify this factor, we first fractionated the conditioned medium with a size exclusion column with a 10 kDa cut-off (Fig. 2B). The flow-through fraction of the condition medium depleted of molecules >10 kDa did not significantly increase the colony-forming capacity of MDA-MB-468 cells compared to unconditioned medium. On the contrary the concentrated fraction of the conditioned medium enriched in molecules >10 kDa significantly boosted the colony formation when used to supplement the fresh medium (Fig. 2C). Moreover, its activity was retained when incubated for 15 min at 95 °C (Fig. 2D). To further

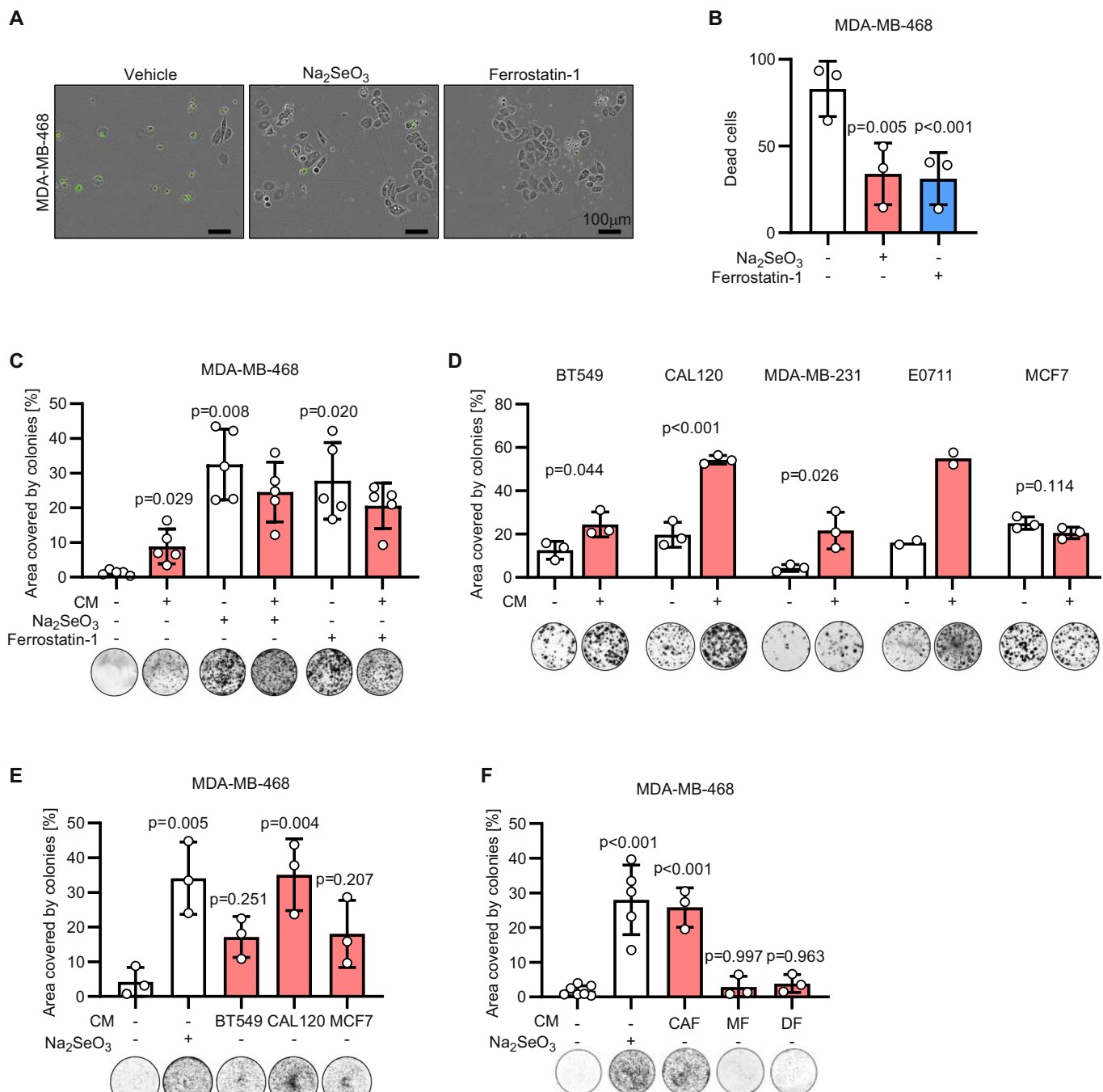

investigate the chemical properties of this factor we applied a Bligh and Dyer lipid extraction to the > 10 kDa fraction of the condition medium. The extracted lipidic fraction used to supplement the medium of the MDA-MB-468 cells increased their colony-forming capacity comparably to sodium selenite supplementation (Fig. 2E). Altogether these results suggest that the active anti-ferroptotic factor produced by TNBC cells at high density is bound to a dispensable carrier >10 kDa and is a heat-resistant lipid sufficient to confer anti-ferroptotic effects to the conditioned medium. Consistently with this hypothesis, the deletion of the long-chain-fatty acid-CoA ligase 3 (ACSL3), an enzyme that plays a key role in lipid

biosynthesis, ablated the protective effect of the conditioned medium (Fig. 2F,G), strengthening the hypothesis that the secreted pro-survival factor is of lipidic nature.

## Monounsaturated fatty acids prevent ferroptosis of TNBC cells

To gain an insight into the lipidic composition of the conditioned medium, we performed a lipidomic analysis on its size-separated fractions. Overall, we could detect 213 lipids from 12 classes, and all of them were more abundant in the >10 kDa fraction (Fig. 3A).

◀ **Figure 1. Medium conditioned by breast cancer cells and cancer-associated fibroblasts enhances the clonogenicity of TNBC cells.**

(A) Representative images of MDA-MB-468 cells seeded at low density and incubated for 96 h with 50 nM Na$_2$SeO$_3$ or 2 μM Ferrostain-1 as indicated. Phase contrast and fluorescence images were overlayed and dead cells were identified by green fluorescence emitted by the Incucyte® Cytox Green Dye. (B) Quantification of the dead cells stained with the green fluorescent dye in the conditions described in (A). *P* values refer to a one-way ANOVA test for paired samples with Geisser–Greenhouse correction. (C) Well area covered by colonies formed by MDA-MB-468 cells incubated for 7 days with mock medium or medium conditioned by MDA-MB-468 cells seeded at high density (CM). Media used for the colony-forming assay were supplemented with 50 nM Na$_2$SeO$_3$ or 2 μM Ferrostain-1 as indicated. *P* values refer to a two-way ANOVA test for unpaired samples with Dunnett's multiple comparisons test. (D) Well area covered by colonies formed by the indicated cell lines incubated for 7 days with mock medium or medium conditioned by BT549, CAL120, MDA-MB-231, EO771 and MCF7 cells seeded at high density. Conditioned medium was used on the respective cell line. *P* values refer to a two-tailed, homoscedastic Student's *t* tests for unpaired samples. (E) Well area covered by colonies formed by MDA-MB-468 incubated for 7 days with mock medium or medium conditioned by BT549, CAL120 or MCF7 cells seeded at high density. Media used for the colony-forming assay were supplemented with 50 nM Na$_2$SeO$_3$ as indicated. *P* values refer to a one-way ANOVA test for unpaired samples with Dunnett's multiple comparisons test. (F) Well area covered by colonies formed by MDA-MB-468 incubated for 7 days with mock medium or medium conditioned by cancer-associated fibroblasts (CAF), immortalised mammary fibroblasts (MF), or immortalised human dermal fibroblasts (DF) seeded at high density. Media used for the colony-forming assay were supplemented with 50 nM Na$_2$SeO$_3$ as indicated. *P* values refer to a one-way ANOVA test for unpaired samples with Dunnett's multiple comparisons test. (C–F) Representative images of wells with colonies are shown for each experimental condition. (B–F) $n_{exp}$ = 2-7 as indicated by the data points in each panel. Bars that represent mean ± s.d. are shown. Source data are available online for this figure.

Notably, 86% of the acyl chains of the detectable lipids had one (35%) or more double bonds (Fig. 3B). In addition to the lipidic classes reported in (Fig. 3A), we could detect free C18:1 monounsaturated fatty acid (MUFA), that was significantly more abundant in the >10 kDa fraction of the conditioned medium (Fig. 3C).

Based on this evidence, we tested if individual fatty acid (FA) supplementation influenced the capacity of MDA-MB-468 cells to form colonies. At 50 μM, the saturated FA 16:0 and 18:0 did not have a significant effect on colony formation, while out of the 8 MUFAs tested, 7 significantly increased the colony-forming capacity (Figs. 3D and EV2B). Of the 7 MUFAs with colony-stimulating activity, 4 (C16:1n7, C18:1n7, C18:1n9 and C20:1n9) retained their activity at a lower concentration (10 μM, Figs. 3E and EV2C). In addition, 10 μM C18:1n9 (oleic acid) increased the colony formation capacity in BT549 and CAL120 TNBC cells but not in MCF7 cells, thereby phenocopying the effects of conditioned medium across these breast cancer cell lines (Figs. 3F and EV2D).

To profile the effects of oleic acid and conditioned medium on the lipidome of recipient cells we performed lipidomic analysis on MDA-MB-468 cells cultured at low cell density and exposed to those agents. The results show that 350 out of the 790 lipidic species identified were consistently and coherently regulated by the oleic acid and the conditioned medium compared to cells exposed to mock medium (Fig. EV2A). Triglycerides emerged as the class of lipids whose abundance was consistently increased by both oleic acid and conditioned medium. Conversely, polyunsaturated phosphatidylcholines were downregulated in cells cultured with conditioned medium or oleic acid. These results show that the lipidome changes elicited by oleic acid and the condition medium largely overlap, suggesting a common mode of action for these anti-ferroptotic agents.

## SCD activity is required to condition the medium with an anti-ferroptotic factor

Notably, all the MUFAs stimulating the clonogenic growth at 10 μM (Fig. 3E) were products of the stearoyl-CoA desaturase (SCD), therefore we hypothesised that the presence of the anti-ferroptotic factor in the conditioned medium could be SCD-mediated. To test this hypothesis, we compared the colony-stimulating capacities of media conditioned by MDA-MB-468 cells pre-treated with the SCD inhibitor CAY10566 or vehicle control (Fig. 4A). SCD inhibition had no significant effects on the viability

of high-density cultures employed to condition the medium (Fig. 4B). Nevertheless, the medium conditioned by SCD-inhibited cells was significantly less effective in stimulating colony formation compared to medium conditioned by vehicle-treated cells (Fig. 4C). Consistently with this observation, the levels of C16:1n7 (palmitoleic acid), a product of SCD, were significantly lower in the medium conditioned by SCD-pre-inhibited cells (Fig. 4D) while other fatty acids were not significantly affected (Fig. EV3A).

To further validate the essential role of SCD to produce the anti-ferroptotic factor secreted in the medium, we generated two SCD knockout (ko) clones derived from MDA-MB-468 cell lines (Fig. 4E). Constitutive *SCD* deletion (SCDko) impaired cell proliferation (Fig. 4F), therefore the number of cells seeded was adjusted to obtain comparable levels of medium-conditioning in 48 h (Fig. 4F). Compared to medium conditioned by NTC cells, the media from SCDko cells was significantly less potent in stimulating colony formation (Fig. 4G). Moreover, media conditioned by SCDko cells was less effective than medium from NTC cells in supporting colony formation of cells that lack the selenoprotein P (SELENOP) receptor, LRP8 (Fig. EV3B,C). These results indicate that SCD activity in TNBC cells is required to produce an anti-ferroptotic factor transferrable with the conditioned medium, whose action is not dependent on the uptake of a major selenium carrier, SELENOP.

To test if the SCD-dependent secretion of lipids occurs in the tumour microenvironment in vivo, SCDwt and SCDko MDA-MB-468 cells were orthotopically transplanted in the mammary fat pad of mice, and the interstitial fluid was collected from the respective tumours. The deletion of SCD (Fig. 4H) did not affect tumour growth (Fig. 4I,J), but it selectively decreased the levels of the SCD-derived MUFAs oleic, palmitoleic and vaccenic acids in the tumour interstitial fluid (Fig. 4K). These results show that breast cancer cells release SCD-produced MUFAs in the extracellular tumour microenvironment where, similarly to what we observed in high cell density cultures, MUFAs and selenium-dependent pathways have overlapping anti-ferroptotic functions.

## Cell density-dependent loss of SCD expression sensitises TNBC cells to ferroptosis

Having demonstrated the anti-ferroptotic role of SCD-derived lipids secreted in the conditioned medium, we assessed whether

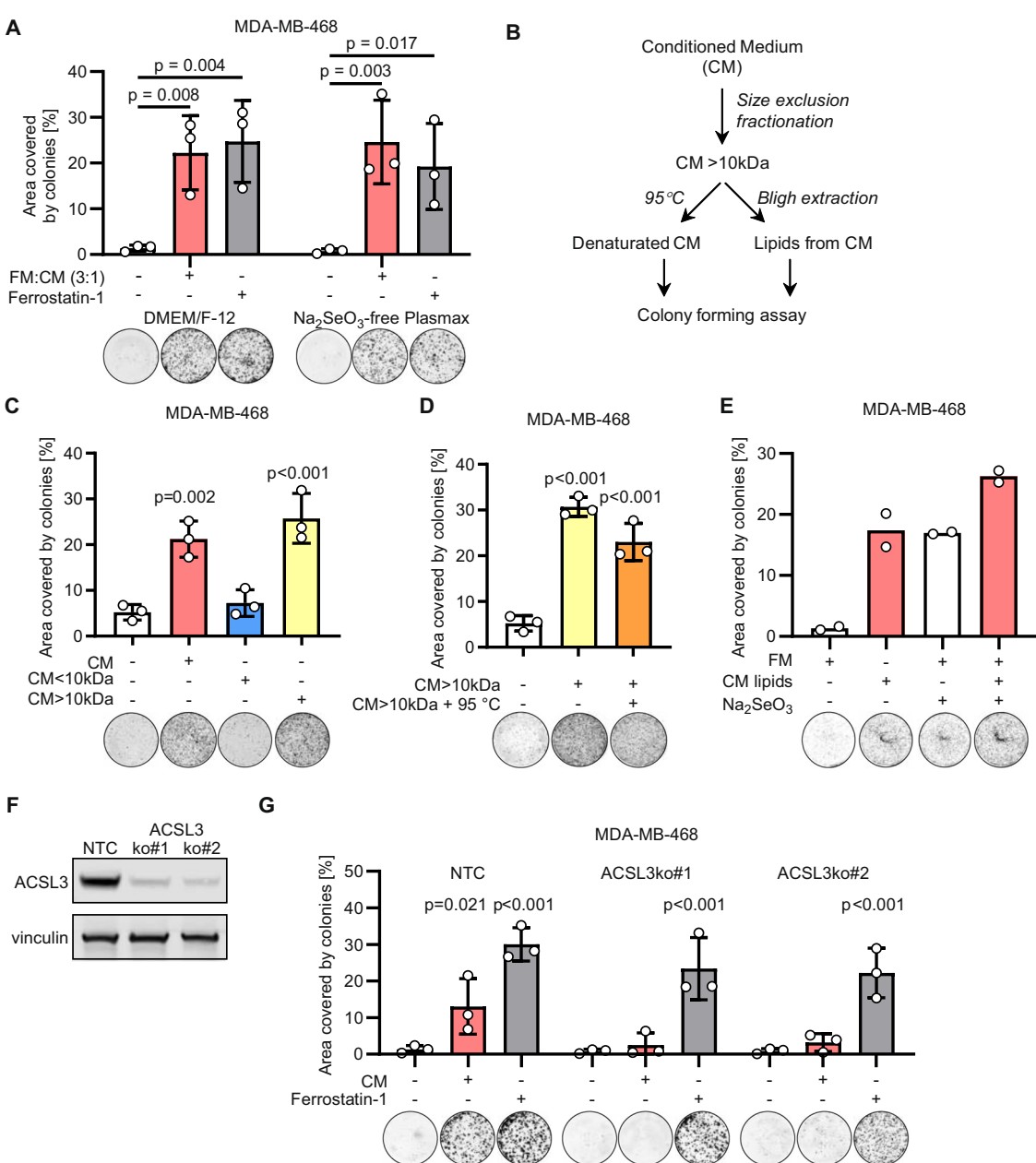

**Figure 2. Breast cancer cells produce an anti-ferroptotic molecule at high density.**

(A) Well area covered by colonies formed by MDA-MB-468 incubated for 7 days with fresh medium (FM) without or with supplementation of 2 μM ferrostatin-1, or with FM mixed 3:1 with medium conditioned by MDA-MB-468 cells seeded at high density (CM). Experiments were performed in DMEM/F-12 or $Na_2SO_3$-free Plasmax™. P values refer to a two-way ANOVA test for unpaired samples with Dunnett's multiple comparisons test. (B) Schematic diagram of the analytical procedures applied to CM to identify the factor rescuing colony formation. (C, D) Well area covered by colonies formed by MDA-MB-468 incubated for 7 days with CM or with medium supplemented with the fractions separated by a size exclusion column as described in "Methods" section (C), and further incubated for 15 min at 95 °C (D). P values refer to a one-way ANOVA test for unpaired samples with Dunnett's multiple comparisons test. (E) Well area covered by colonies formed by MDA-MB-468 incubated for 7 days with fresh medium (FM) supplemented with lipidic extracts obtained from the conditioned medium (CM) as shown in (B). (F) Immunoblot of ACSL3 and vinculin (loading control) in NTC and two ACSL3 ko clones obtained from MDA-MB-468 cells. (G) Well area covered by colonies formed by MDA-MB-468 control (NTC) or ACSL3 ko cells incubated for 7 days with fresh medium, conditioned medium (CM) or fresh medium supplemented with 2 μM ferrostatin-1 as indicated. P values refer to a two-way ANOVA test for paired samples with Sidak's multiple comparisons test. (A, C–E, G) Representative images of wells with colonies are shown for each experimental condition. $n_{exp} = 2$–3 as indicated by the data points in each panel. Bars that represent mean ± s.d. are shown. Source data are available online for this figure.

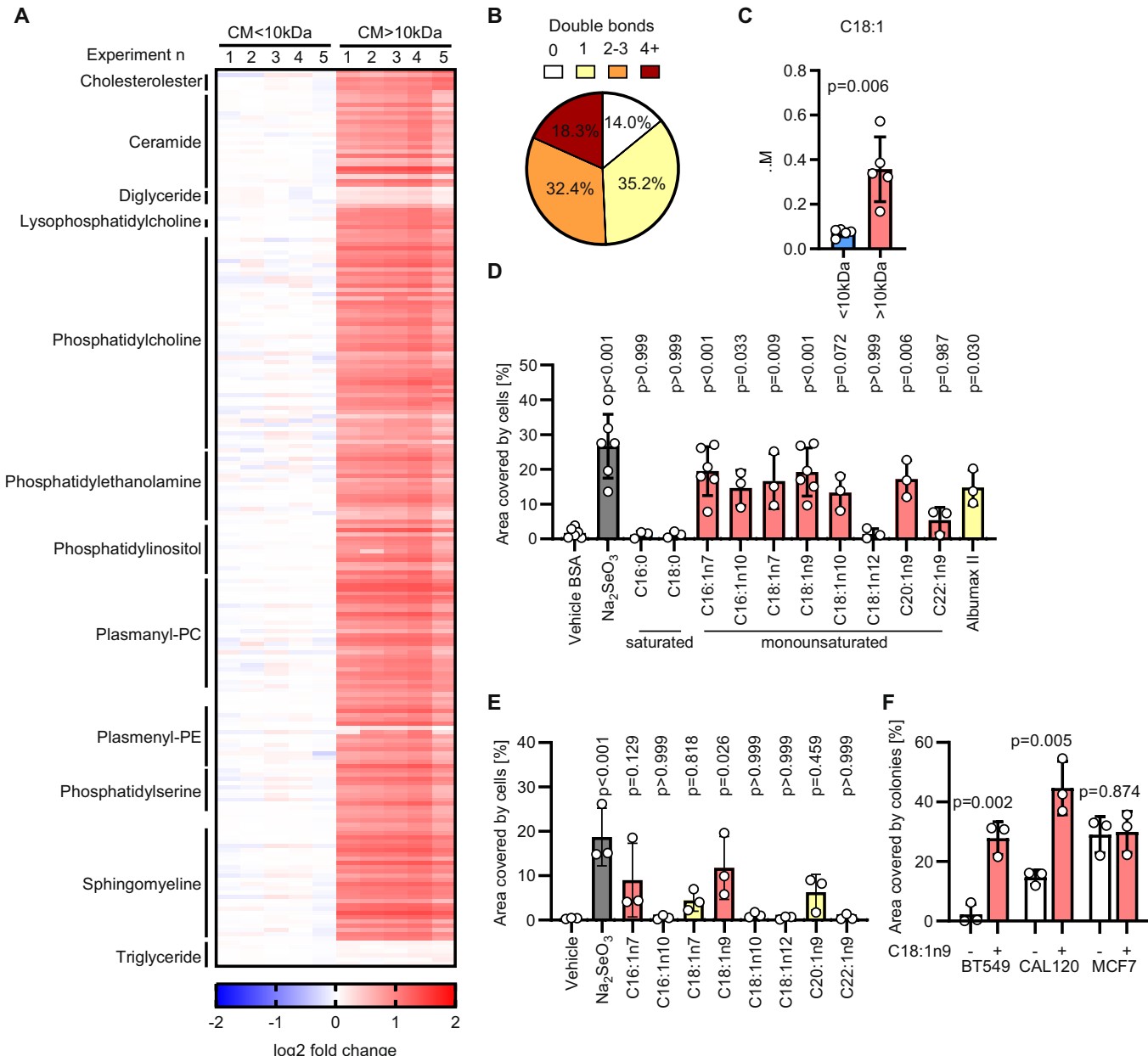

**Figure 3. Monounsaturated fatty acids are enriched in the conditioned medium and prevent ferroptosis.**

(A) Heatmap of lipids identified in the fractions of condition medium (CM) described in Fig. 2B,C. Selected classes of lipids are indicated. $n_{exp} = 5$. (B) Pie chart depicting the proportion of lipid-bound fatty acids with 0, 1, 2–3, or ≥4 double bonds detected by LC-MS in the CM > 10 kDa fraction. (C) Concentration of free C18:1 fatty acid measured in the fractions of conditioned medium described in Fig. 2B,C. P value refers to a two-tailed, homoscedastic Student's t tests for unpaired samples. (D) Well area covered by colonies formed by MDA-MB-468 cells incubated for 7 days with medium supplemented with 0.015% BSA (vehicle), 0.015% BSA + 50 nM Na₂SeO₃, 0.015% BSA + 50 μM of the indicated fatty acids, 1.6 g/L lipid-rich BSA (Albumax II). P values refer to a one-way ANOVA test for unpaired samples with Dunnett's multiple comparisons test. (E) Well area covered by colonies formed by MDA-MB-468 cells incubated for 7 days with fresh medium supplemented with vehicle control, 50 nM Na₂SeO₃ or 10 μM of the indicated fatty acid. P values refer to a one-way ANOVA test for unpaired samples with Dunnett's multiple comparisons test. (F) Well area covered by colonies formed by the indicated cell lines incubated for 7 days with or without the supplementation of 10 μM oleic acid (C18:1n9). P values refer to a two-tailed, homoscedastic Student's t tests for unpaired samples. (C–F) $n_{exp} = 3$–5 as indicated by the data points. Bars represent mean ± s.d. Source data are available online for this figure.

SCD deletion could sensitise to ferroptosis TNBC cells cultured at high density upon selenium restriction (i.e. 2.5% FBS). The number of SCDko cells cultured at high density was partially rescued by the individual supplementation of oleic acid, sodium selenite,

ferrostatin-1 and deferoxamine (Fig. 5A) demonstrating that SCDko promotes ferroptosis upon selenium restriction. In addition, the survival of the SCDko cells, but not that of NTC control cells, was significantly impaired by the treatment with 50 nM of

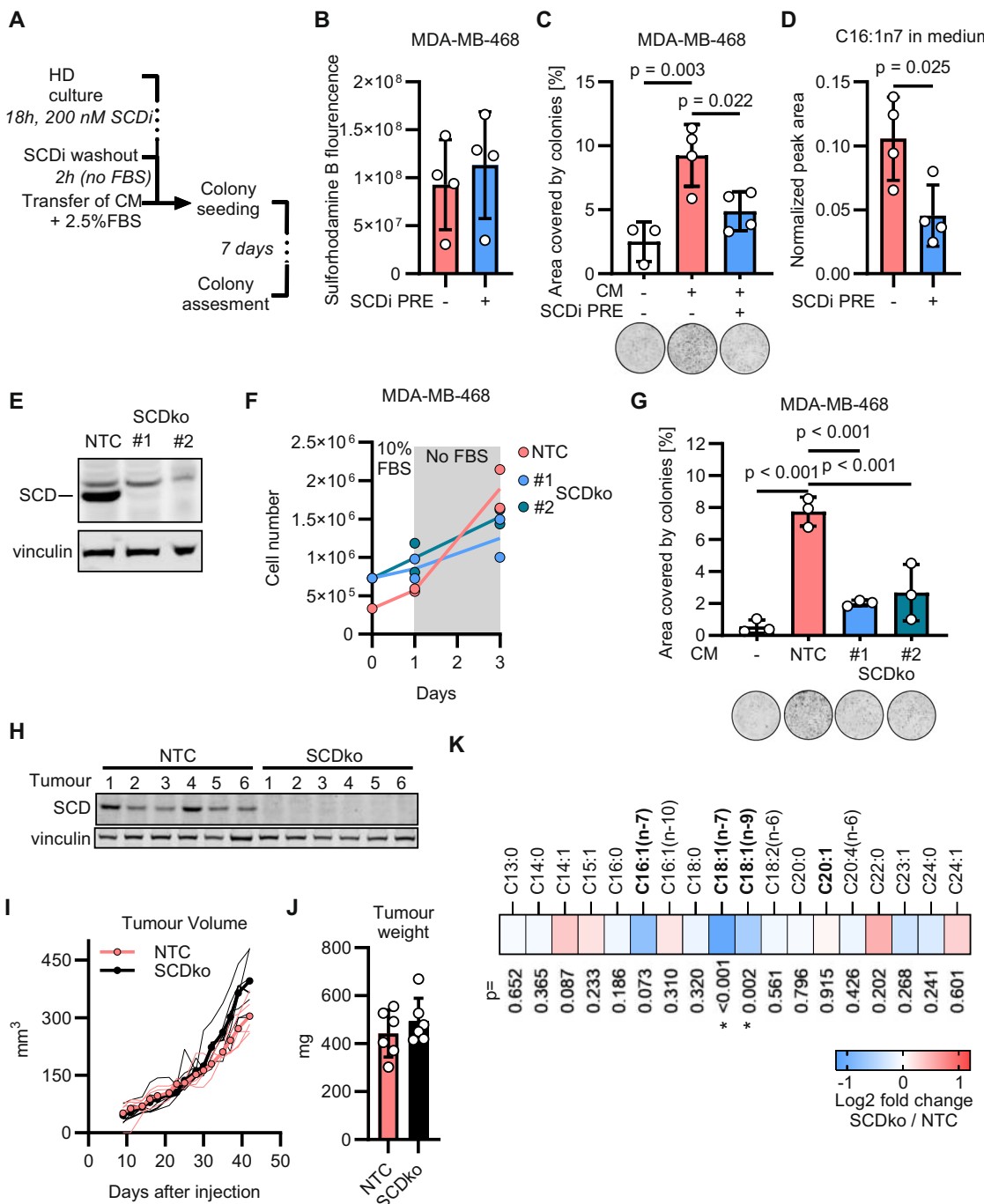

GPX4 inhibitor RSL3 and rescued by oleic acid supplementation (Fig. 5B; Appendix S2A), demonstrating that the SCD-dependent production of MUFAs such as palmitoleic and oleic acids (Fig. 5C,D) becomes essential for cell survival when the selenium-dependent anti-ferroptotic function is hindered. Unexpectedly, the levels of oxidised C11 BODIPY, a probe for lipid peroxidation, were not significantly affected by *SCD* deletion (Fig. 5E) suggesting that SCD-derived fatty acids desensitise cells from ferroptosis acting downstream of lipid peroxides.

To find a mechanistic rationale for the cell density-dependent susceptibility to ferroptosis observed in TNBC cells (i.e. maximal at low density), we assessed the levels of SCD mRNA and protein expression and found that it was proportional to cell density in all four TNBC lines tested (Fig. 5F,G). Consistently with the unresponsiveness of luminal A MCF7 cells to the conditioned medium (Fig. 1B), these cells did not modulate *SCD* expression depending on cell density (Fig. 5F,G). Furthermore, the mRNA expression of key enzymes for fatty acid synthesis, elongation, and

**Figure 4. SCD is required for the anti-ferroptotic capacities of the conditioned medium.**

(A) Schematic depicting the experimental workflow to condition the medium by MDA-MB-468 at high density (HD) pre-incubated with SCD inhibitor (SCDi, CAY10566). (B) Quantification of Sulforhodamine B staining performed on MDA-MB-468 cells immediately after the conditioning of the medium that followed the SCD inhibitor pre-treatment (SCDi PRE) as described in (A). (C) Well area covered by colonies formed by MDA-MB-468 cells incubated for 7 days with mock medium and medium conditioned for 2 h by MDA-MB-468 cells without or with SCD inhibitor pre-treatment (SCDi PRE). *P* values refer to a one-way ANOVA test for unpaired samples with Dunnett's multiple comparisons test. Representative images of wells with colonies are shown for each experimental condition. (D) Quantification of total palmitoleic acid levels (free and lipid-bound C16:1n7) in medium conditioned by MDA-MB-468 cells without or with SCD inhibitor pre-treatment (SCDi PRE). Peak area values are normalised on the signal from internal standard (C17:0). *P* value refers to a two-tailed, homoscedastic Student's *t* tests for unpaired samples. (E) Immunoblot of SCD and vinculin (loading control) in NTC and two SCDko clones from MDA-MB-468 breast cancer cells. Images representative of three independent experiments are shown. (F) The number of cells obtained with MDA-MB-468 NTC and SCDko clones cultured as described in the "Methods" section "Conditioned medium". The number of seeded NTC and SCDko cells was adjusted to reach a comparable conditioning of the medium employed for the colony-forming assays shown in (G). $n_{exp} = 2$ as indicated by data points, lines represent mean values. (G) Well area covered by colonies formed by MDA-MB-468 cells incubated for 7 days with mock medium or medium conditioned by NTC or SCDko MDA-MB-468 clones cultured as described in (F) and in "Methods" section "Conditioned medium" *P* values refer to a one-way ANOVA test for unpaired samples with Dunnett's multiple comparisons test. Representative images of wells with colonies are shown for each experimental condition. (H) Immunoblot of SCD and vinculin (loading control) in NTC and SCDko tumours ($n = 6$) harvested 7 weeks after that $3 \times 10^6$ MDA-MB-468 cells were transplanted unilaterally in the mammary fat pad of NSG female mice. (I) The calliper-measured volume of the tumours obtained as described in (H). Thinner lines represent the volumes of tumours obtained from each transplantation, thicker lines and symbols represent group means. $n_{tumour} = 6$. (J) Ex vivo tumour weight of NTC and SCDko tumours described in (H). $n_{tumour} = 6$. Bars represent mean ± s.d. (K) Quantification of total (free and lipid-bound) fatty acid species in interstitial fluid of the tumours described in (H). SCD-derived fatty acids are highlighted in bold. Peak area values normalised on the signal from the internal standard (C17:0) were used to calculate the Log2 fold change. * indicate significance and *P* values refer to a two-tailed, homoscedastic Student's *t* tests for unpaired samples. $n_{tumour} = 6$. (B–D, G) $n_{exp} = 3–4$ as indicated by the data points in each panel. Bars represent mean ± s.d. Source data are available online for this figure.

desaturation (FASN, ELOVL3, FADS1 and FADS2) was also proportional to TNBC cell density (Fig. 5H).

To strengthen the relevance to human tumour biology of our results on cell density, we mined four studies with breast cancer patients from an online database (Zhao et al, 2020; Gkountela et al, 2019; Szczerba et al, 2019; Aceto et al, 2014; Yu et al, 2014; Data ref: Szczerba et al, 2019; Aceto et al, 2014; Yu et al, 2014). This analysis revealed that SCD expression is indeed lower in metastasis and circulating tumour cells from breast cancer patients, compared to the primary tumours (Fig. 5I).

Overall, these data show that TNBC cells have an impaired fatty acid synthesis and desaturation capacity at low cell density. Moreover, the pattern of SCD expression observed in primary breast tumour compared to circulating and metastatic cells resemble that observed in low- and high-density cultures, respectively. These results suggest an increased vulnerability to ferroptosis mediated by selenium starvation in cells cultured at low density or in blood circulation.

## Inhibition of selenocysteine synthesis induces ferroptosis and impairs lung metastatic colonisation

GPX4 is a potent anti-ferroptotic enzyme whose catalytic activity requires selenocysteine in its active site. The amino acid selenocysteine is only synthesised on its tRNA and cannot be re-cycled as such for the synthesis of selenoproteins (Fig. 6A). This makes selenocysteine synthesis a limiting step to counteract ferroptosis and it could constitute a novel therapeutic target for ferroptosis-primed TNBC cells. On these bases, we designed guide RNAs (gRNAs) against the three genes encoding the enzymes required for the synthesis of selenocysteinilated tRNA$^{sec}$ (PSTK, SEPHS2 and SEPSECS, Fig. 6A). The interference with the expression of each of the enzymes resulted in low GPX4 levels in MDA-MB-468 cells, comparable to those observed upon selenium starvation (Fig. 6B). The loss of SEPHS2 or SEPSECS expression as well as Ferrostain-1 supplementation had no significant effects on the viability of cells seeded at high density, demonstrating that the anti-ferroptotic function of selenoproteins is redundant under these

conditions (Figs. 6C and EV4A). On the contrary, at low cell density survival depends on ferroptosis inhibition, achieved by either selenium or ferroststatin-1 supplementation (see NTC in Figs. 6D and EV4B). In these ferroptosis-priming conditions, the interference with SEPSECS and SEPHS2 ablates the protective effect of selenium, while Ferrostatin-1 retains its anti-ferroptotic effect. A similar trend was observed for PSTK interference, indicating that selenocysteine synthesis inhibition triggers ferroptosis selectively in TNBC cells cultured at low density (Figs. 6D and EV4B).

Next, we orthotopically transplanted NTC control, sgSEPHS2 and sgSEPSECS MDA-MB-468 cell pools into the mouse mammary fat pad to investigate the effect of selenocysteine synthesis on tumour growth (Fig. 6E,F). The growth rate of sgSEPHS2 and sgSEPSECS tumours was not different from that of NTC controls, an observation confirmed ex vivo by tumour weights (Fig. 6F,G). However, compared to NTC controls the lower levels of SEPHS2 and GPX4 expression observed pre-implantation (Fig. 6E) were restored in sgSEPHS2 or sgSEPSECS tumours at endpoint (Fig. EV4D,E), suggesting that the counterselection of the interfered cells did not cause a tumour growth delay appreciable with the experimental settings employed.

To assess whether the colonisation of distant organs by TNBC cells is affected by selenocysteine synthesis inhibition, we injected luciferase-expressing NTC, sgSEPHS2 and sgSEPSECS MDA-MB-468 cells into the tail vein of female NSG mice. One hour after tail vein injection, the luciferase signal was detectable in the lungs of all mice, and it was comparable between experimental groups (Fig. EV4C). However, 9 days after injection, the luciferase signal was significantly lower (~twofold) in animals injected with sgSEPHS2 and sgSEPSECS compared to NTC cells (Fig. 6H). This difference was further exacerbated 35 days after injection (~three- to fivefold, Fig. 6I). To validate the luciferase-based results, we stained the lungs of the tail vein-injected animals with antibodies against Cas9 and human Ku80 to visualise individual metastatic cells (Fig. 6J). We found two to three times more Ku80-positive cells in lungs of mice injected with NTC cells (~1.15% of all cells in the lung sections examined) compared to sgSEPHS2(~0.55%) or

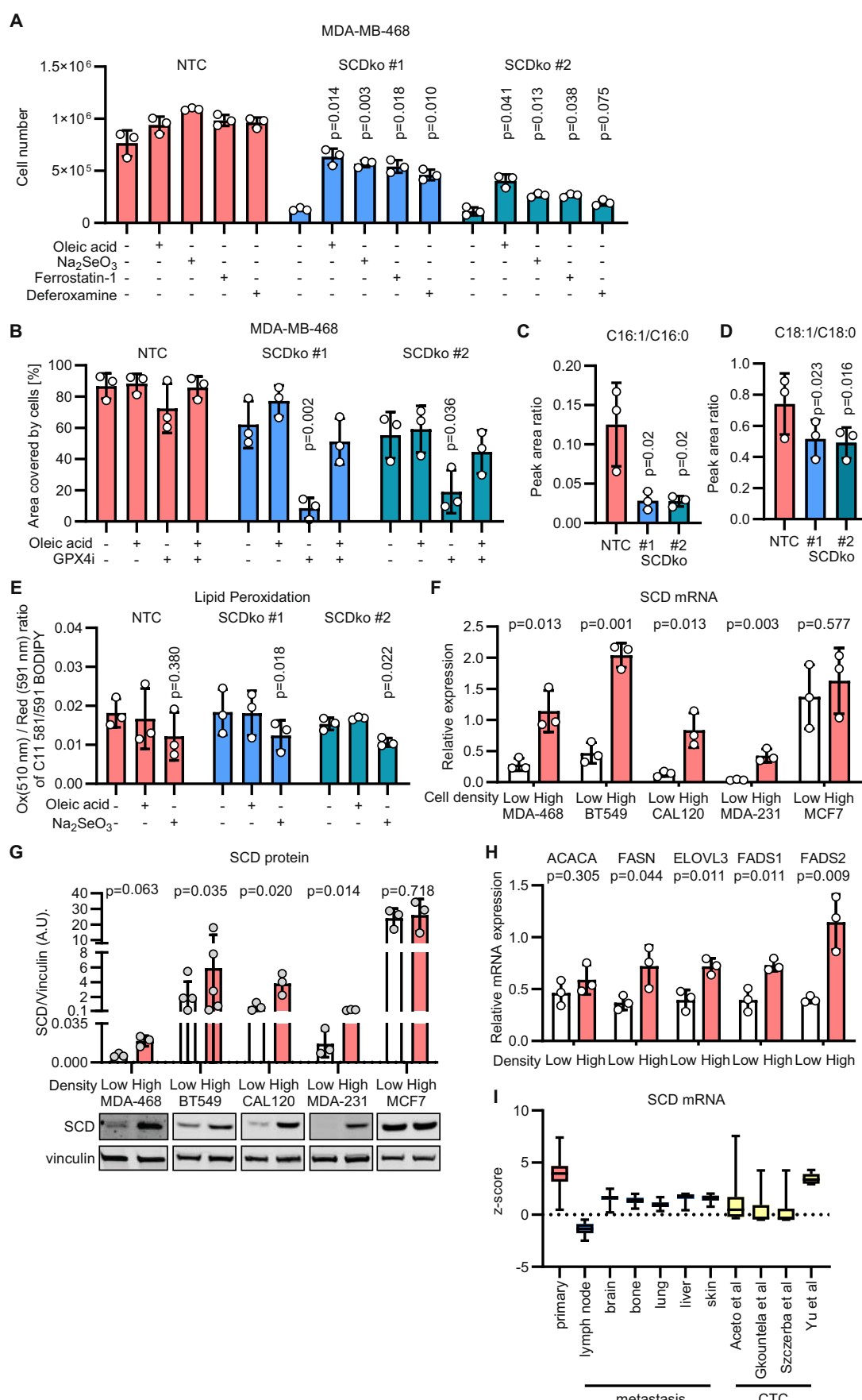

**Figure 5. Loss of SCD sensitises cells to ferroptosis.**

(A) The number of NTC and SCDko MDA-MB-468 cells seeded at high density and incubated for 5 days without or with 10 µM oleic acid, 50 nM Na$_2$SeO$_3$, 2 µM Ferrostatin-1 or 2.5 µM Deferoxamine as indicated. $P$ values refer to a two-way ANOVA test for unpaired samples with Dunnett's multiple comparisons test comparing to the respective unsupplemented controls. (B) Well area covered by NTC and SCDko MDA-MB-468 cells seeded at high density and incubated for 5 days without or with 50 nM GPX4 inhibitor (RSL3), 10 µM Oleic acid or their combination (50 nM + 10 µM, respectively). $P$ values refer to a two-way ANOVA test for unpaired samples with Dunnett's multiple comparisons test comparing to the respective unsupplemented controls. (C, D) Peak Area ratio for C16:1/C16:0 (C) and C18:1/C18:0 (D) fatty acids in NTC and SCDko MDA-MB-468 cells seeded at high density. $P$ values refer to a one-way repeated measures ANOVA test for paired samples with Dunnett's multiple comparisons test compared to the NTC controls. (E) Ratio between oxidised (510 nm) and reduced (591 nm) BODIPY 581/591 C11 (lipid peroxidation sensor) in NTC and SCDko MDA-MB-468 cells seeded at high density and incubated for 2 days without or with 10 µM oleic acid or 50 nM Na$_2$SeO$_3$ as indicated. $P$ values refer to a one-way repeated measures ANOVA test for paired samples with Dunnett's multiple comparisons test, comparing to the respective unsupplemented controls. (F) qPCR quantification of SCD mRNA expression in MDA-MB-468, BT549, CAL120, MDA-MB-231 and MCF7 cells seeded at low and high density and cultured for 2 days. The expression is relative to Actin, lamin B1 and TBP mRNA abundance. $P$ values refer to a two-tailed, homoscedastic Student's $t$ tests for unpaired samples. (G) Immunoblot analysis of SCD levels in MDA-MB-468 (antibody Alpha Diagnostic International Inc., #SCD11-A, 1:1000) and BT549, CAL120, MDA-MB-231, MCF7 cells (antibody Abcam, #ab19862, 1:1000) seeded at low and high density and cultured for 2 days. The lower inset shows representative images of the western blot for SCD and vinculin (loading control) from one of the experiments quantified in the upper panel. $P$ value refers to a two-tailed, homoscedastic Student's $t$ tests for unpaired samples comparing low and high densities. (H) qPCR quantification of ACACA, FASN, ELOVL3, FADS1 and FADS2 mRNA expression in MDA-MB-468 cells seeded at low and high density and cultured for 2 days. The expression is normalised on the mean mRNA abundance of ACTB, LMNB1 and TBP (Appendix Fig. S2B–D). $P$ values refer to a two-tailed, homoscedastic Student's $t$ tests for unpaired samples comparing low and high densities. (I) Relative expression of SCD in breast cancer patient-derived samples from primary tumours, metastatic lesions and circulating tumour cells. The analysed data were obtained from the ctcRbase database. Whiskers represent highest and lowest values. Boxes span from first quartile values to third quartile values with line representing the median expression level. (A–H) $n_{exp} = 3$–6 as indicated by the data points in each panel. Bars represent mean ± s.d. Source data are available online for this figure.

sgSEPSECS cells (~0.4%, Fig. 6K). The magnitude of these differences was enhanced if Cas9-positive cells were counted in the lung sections of the same mice (NTC cells ~0.55%, sgSEPHS2 ~0.1%, and sgSEPSECS ~0.2%). Overall, these in vivo results demonstrate that the expression of SEPHS2 and SEPSECS is less critical for mammary breast tumour growth than for breast cancer cells to survive in the bloodstream and colonise the lungs. These findings recapitulate the conditional essentiality of seleno-cysteine synthesis for clonogenic growth observed in vitro.

## Discussion

Understanding the regulation of ferroptotic cell death will provide new avenues to treat cardiovascular disease, neurodegenerative diseases, and cancer (Chen et al, 2021; Yan et al, 2021). In recent years, several redundant anti-ferroptosis pathways have been described (Freitas et al, 2024; Mishima et al, 2022; Bersuker et al, 2019; Yang et al, 2014). However, it remains to be elucidated which of those pathways are conserved in specific tissues and cell types. We and others have previously shown that selenium deprivation and uptake inhibition sensitises to ferroptosis selectively when cells are cultured at low density (Vande Voorde et al, 2019; Wu et al, 2019; Alborzinia et al, 2023). Here we demonstrate that in triple-negative breast cancer (TNBC) cells, the seeding density positively regulates the expression of genes involved in fatty acid synthesis and desaturation (i.e., SCD). Intriguingly, we found that medium conditioned by cancer cells and cancer-associated fibroblasts cultured at high density has a pro-survival effect on TNBC cells seeded at low density for colony-forming assays (Fig. 1). Specifically, we showed that the pro-clonogenic activity of the conditioned medium is thermostable and is retained in its lipidic fraction (Fig. 2E). Moreover, the supplementation of specific monounsaturated fatty acids (MUFAs) was effective in preventing ferroptosis occurring during colony formation (Fig. 3D). Indeed, SCD-derived MUFAs are the most potent in rescuing ferroptosis when supplemented to TNBC cells at low density. Consistently, the loss of SCD activity in TNBC cells cultured at high-density sensitises to

ferroptosis (Fig. 5A–C). These results suggest that diets low in SCD-derived MUFAs might enhance the efficacy of pro-ferroptotic agents in TNBC (Tesfay et al, 2019). In line with this hypothesis, Dierge et al showed that oleate-rich and PUFA-rich diets have opposite effects on the ferroptosis susceptibility of acidic cancer cells, and Ubellacker et al described the anti-ferroptotic effects of oleic acid in melanoma (Dierge et al, 2021; Ubellacker et al, 2020). We showed that unlike normal cells, high-density cultures of breast cancer cells or cancer-associated fibroblasts secrete lipids containing monounsaturated fatty acid in the extracellular environment in quantities that prevent ferroptosis when supplemented to colony-forming cells (Figs. 1 and 2E). Recently, it has been shown that primary breast tumours can secrete a factor that increases palmitate production in the lung, conditioning the metastatic niche to favour cancer cell growth (Altea-Manzano et al, 2023). Furthermore, it has been shown that the pro-metastatic effect of high-fat diets relies on CD36-dependent MUFA uptake (Terry et al, 2023). Our results show that mammary tumours, where cells are at high density, secrete MUFA-containing lipids in the interstitial fluid altering the lipid composition of tumour microenvironment (Fig. 4K). On these bases, we hypothesised that selenocysteine biosynthesis could be a novel target to kill ferroptosis-primed triple-negative breast cancer (TNBC) cells that have left the MUFA-rich environment of the primary tumour (Fig. 7). Indeed, the interference with SEPHS2 or SEPSECS, enzymes of the selenocysteine biosynthesis pathway (Fig. 6A), strongly impairs the lung seeding of cancer cells injected into the bloodstream (Fig. 6H–K). In culture, we show that each of the three enzymes synthesising the Sec-tRNA$^{sec}$ (PSTK, SEPHS2 and SEPSECS), as well as selenium supplementation, are all required to prevent ferroptosis in cells at low density (Fig. 6). Carlisle et al have shown that high SEPHS2 expression correlates with poor survival in patients with breast carcinoma and that its loss induces glioblastoma cell death by the accumulation of toxic H$_2$Se upon supraphysiological supplementation of sodium selenite (Carlisle et al, 2020). Our data show that loss of SEPHS2 in TNBC cells phenocopies the cytotoxic effects of selenium deprivation and does not require the accumulation of toxic selenium metabolites to prime cells to ferroptosis (Fig. 6C,D).

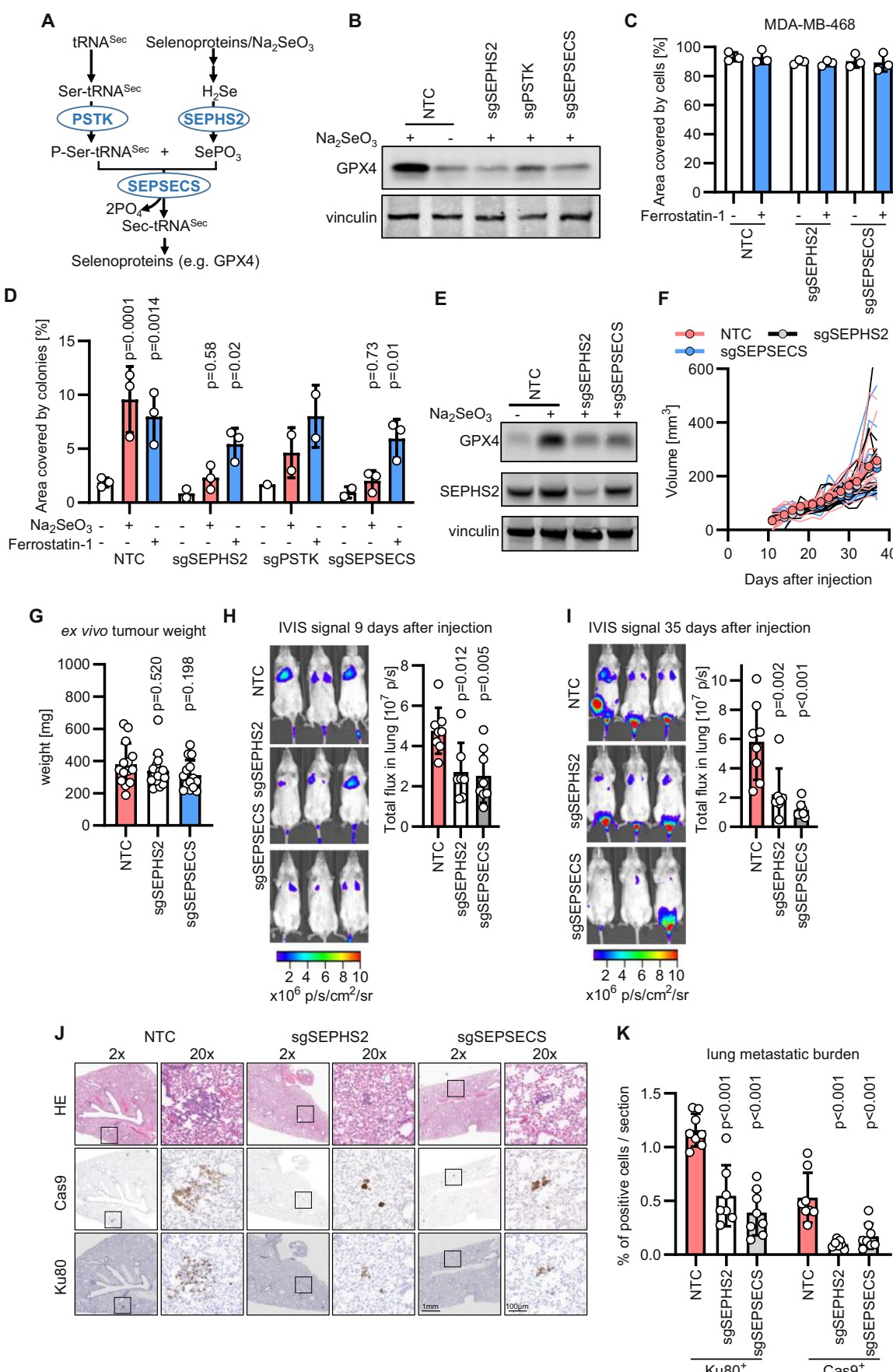

**Figure 6. Targeting selenocysteine biosynthesis impairs lung metastasis of TNBC.**

(A) Schematic depicting selenocysteine synthesis on tRNA[sec]. (B) Immunoblot images of GPX4 and vinculin (loading control) in control MDA-MB-468 cells (NTC) or in cells depleted of PSTK, SEPHS2 and SEPSECS supplemented with 50 nM $Na_2SeO_3$ as indicated. (C) Well area covered by NTC, sgSEPHS2 and sgSEPSECS MDA-MB-468 cells seeded at high density and incubated for 5 days without or with 1 μM Ferrostatin-1. $n_{exp} = 3$. Bars represent mean ± s.d. (D) Well area covered by colonies formed by NTC, sgSEPHS2, sgPSTK and sgSEPSECS MDA-MB-468 cells seeded at low density and incubated for 7 days without and with 50 nM $Na_2SeO_3$ or 1 μM Ferrostatin-1. $n_{exp} = 1$–3 as indicated by the data points. P values refer to a two-way ANOVA test for unpaired samples with Dunnett's multiple comparisons test comparing to the respective unsupplemented controls. Bars represent mean ± s.d. (E) Immunoblot images of GPX4, SEPHS2 and vinculin (loading control) in MDA-MB-468 cells (NTC) or in cells depleted of SEPHS2 or SEPSECS supplemented with 50 nM $Na_2SeO_3$ as indicated. Images are representative of 3 independent experiments. (F) Calliper-measured volume of tumours in the mammary fat pad of female NSG mice transplanted with NTC, sgSEPHS2 or sgSEPSECS MDA-MB-468 cells ($n = 8$ mice/group). All mice were transplanted bilaterally with $3 × 10^6$ cells. One transplanted mouse of each NTC and sgSEPSECS group had to be culled before the completion of the experiment for licence reasons due to the location of the tumour. When multiple tumours were formed from a single transplantation of cells their combined volumes were reported as one data point. Thinner lines represent the volumes of tumours obtained from each fat pad transplantation, thicker lines and symbols represent group means ($n = 14$ tumours for NTC, $n = 16$ for sgSEPHS2, $n = 14$ for sgSEPSECS). A CGGC permutation test (https://bioinf.wehi.edu.au/software/compareCurves/) was used to assess the significant differences between growth curves. Adjusted P value comparing the NTC to sgSEPHS2 or sgSEPSECS obtained with a test with 1000 permutations were 0.66 and 0.67, respectively. (G) Ex vivo weight of resected NTC, sgSEPHS2 or SEPSECS tumours taken 38 days after transplantation. P values refer to a one-way ANOVA test for unpaired samples with Dunnett's multiple comparisons test compared to the NTC control. $n = 14$ for NTC or sgSEPSECS tumours and $n = 16$ for sgSEPHS2 as indicated by the data points. (H, I) IVIS pictures and quantification of lung metastasis burden 9 days (H) and 35 days (I) after tail vein injection of $2 × 10^6$ NTC, sgSEPHS2 or sgSEPSECS MDA-MB468 cells. The same mice are shown in (H, I). P value refers to a one-way ANOVA test for unpaired samples with Dunnett's multiple comparisons test compared to the NTC control. $n = 7$–8 female NSG mice as indicated by data points. One injected mouse of sgSEPHS2 group had to be culled due to husbandry reasons. (J) Representative images of staining with haematoxylin and eosin (HE), human Ku80 and Cas9 of lungs of mice described in (H, I) seven weeks after tail vein injection. Black squares frame the areas magnified at ×20. (K) Lung metastasis burden assessed by quantifying the % of human Ku80 and Cas9-positive cells in lungs of mice described in (H, I) seven weeks after tail vein injection. P values refer to a one-way ANOVA test for unpaired samples with Dunnett's multiple comparisons test comparing to the NTC control. $n = 7$–8 female NSG mice as indicated by data points. Source data are available online for this figure.

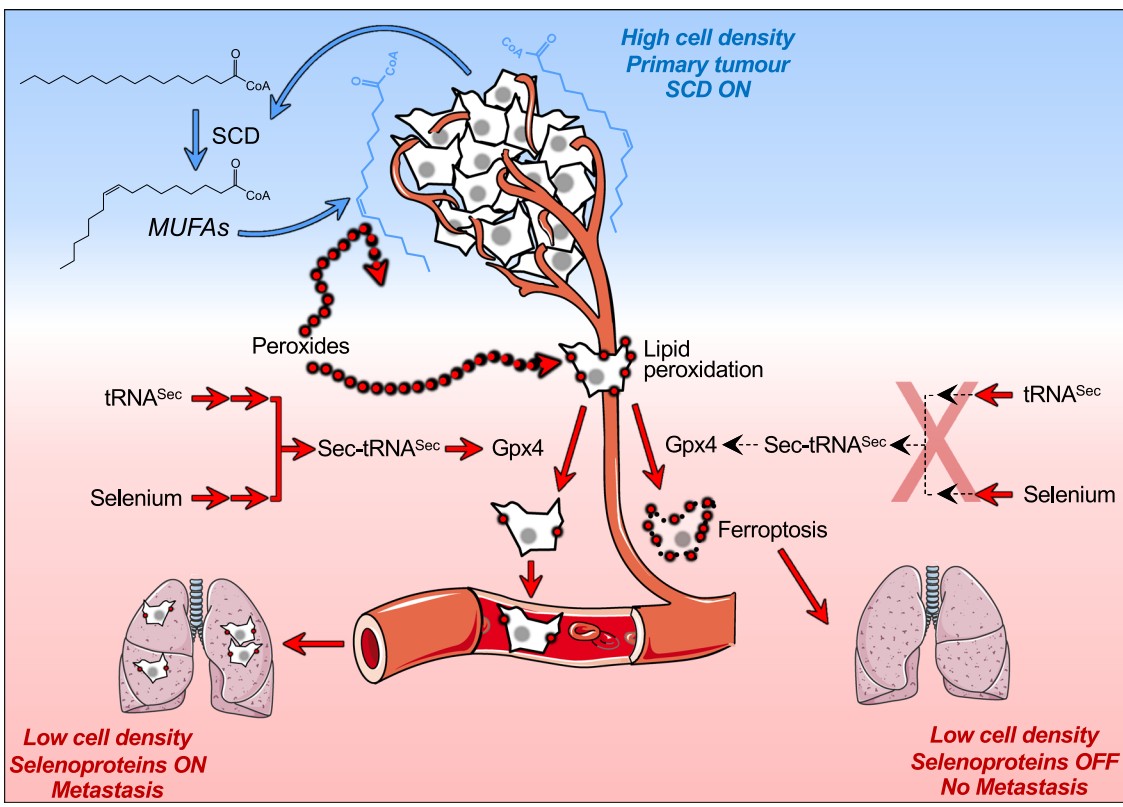

**Figure 7. The lack of a MUFAs shield sensitizes metastatic TNBC cells to selenoprotein synthtesis inhibition.**

Conversely, TNBC cells grown at low density or metastasising in the bloodstream downregulate SCD expression resulting in MUFAs deficiency. This metabolic shift renders metastatic cells dependent on the anti-ferroptotic action of selenoproteins, in particular glutathione peroxidase 4 (GPX4), exposing a novel conditional vulnerability tied to the inhibition of Sec-tRNA[sec] biosynthesis. Indeed, in preclinical models targeting the enzymes of the Sec-tRNAsec biosynthesis effectively impede TNBC lung metastasis.

Breast cancer tissue shows increased levels of protein-bound selenium compared to healthy tissue surrounding the tumour (Charalabopoulos et al, 2006), suggesting that unrestricted selenium availability is promoting tumour cells survival. Indeed, GPX4 inhibition in combination with immunotherapy showed promising results in murine models of TNBC (Yang et al, 2023). While selenium-deprived diets have not been tested in breast cancer models, Eagle et al showed a therapeutic effect of selenium deprivation in leukaemia models, while Alborzinia et al inhibited cellular selenoprotein P uptake to decrease the expression of selenoproteins and induce ferroptosis in neuroblastoma (Alborzinia et al, 2023; Eagle et al, 2022).

On the contrary, selenium-supplemented diets have been assessed with clinical trials (Select trial, NCT00006392) as preventive interventions for breast and prostate cancer with the rationale to boost the antioxidant capacity of the host. However, whether diet intervention can lead to selenium concentrations that modulate selenocysteine synthesis has yet to be established. Moreover, the pharmacologic inhibition of Sec-tRNA$^{sec}$ synthesis might have a more precise effect on the selenoprotein production in cancer cells that enter the blood circulation and depend on the antioxidant function of selenoproteins for their survival. Interestingly, PSTK shows low similarity to other eukaryotic kinases thereby favouring the design of specific small-molecule inhibitors (Sherrer et al, 2008).

Taken together, our in vitro data show that SCD expression increases with cell density in TNBC cells and mechanistically, the loss of the pro-survival effect of SCD in cells at low density enhances their dependency on the antioxidant action of seleno-proteins to survive ferroptosis and form colonies. This principle translates in vivo where we show that TNBC cells injected in the bloodstream metastasize less efficiently to the lung when selenoprotein synthesis is impaired by the deletion of SEPHS2 and SEPSECS, two key enzymes in the Sec-tRNA$^{sec}$ synthesis. In line with the working model proposed, the deletion of SEPHS2 and SEPSECS does not affect the proliferation of TNBC cultured at high density, and a transient interference with their expression does not appreciably delay the growth of orthotopic mammary tumours. Overall, these results suggest that inhibition of the Sec-tRNA$^{sec}$ could be a valid therapeutic target to eradicate metastatic TNBC cells primed to ferroptosis by an imbalance in fatty acid saturation.

# Methods

**Reagents and tools table**

| Reagent/resource | Reference or source | Identifier or catalogue number |
| --- | --- | --- |
| **Experimental models** | | |
| HEK-293 cells (*H. sapiens*) | ATCC | CRL-1573 |
| MDA-MB-468 cells (*H. sapiens*) | ATCC | HTB-132 |
| MDA-MB-231 cells (*H. sapiens*) | ATCC | CRM-HTB-26 |
| BT549 (*H. sapiens*) | ATCC | HTB-122 |
| CAL120 (*H. sapiens*) | DSMZ | ACC 459 |
| MCF7 (*H. sapiens*) | ATCC | HTB-22 |
| Dermal fibroblasts (*H. sapiens*) | Thermo Fisher Scientific | C0045C |
| Mammary fibroblasts (*H. sapiens*) | Prof. Akira Orimo | N/A |
| Cancer-associated fibroblast (*H. sapiens*) | Prof. Akira Orimo | N/A |
| EO711 (*M. musculus*) | CH3 BioSystems | 94A001 |
| NSG mice (*M. musculus*) | CRUK SI animal facility | N/A |
| NSG mice (*M. musculus*) | Charles river | 614NSG |
| **Recombinant DNA** | | |
| lentiCRISPRv2 | Addgene | 52961 |
| pVSV-G (pMD2.g) | Addgene | 12259 |
| psPAX2 | Addgene | 12260 |
| pGL4.50 | Promega | E1310 |
| pLenti6NEO | Invitrogen (discontinued) | N/A |
| **Antibodies** | | |
| Mouse anti-vinculin | Merck | SAB4200080, 1:2000 |
| Rabbit anti-GPX4 | Abcam | ab125066, 1:1000 |
| Rabbit anti-ACSL3 | Abcam | ab151959, 1:1000 |

| Reagent/resource | Reference or source | Identifier or catalogue number |
|---|---|---|
| Rabbit anti-LRP8 | Abcam | ab108208, 1:1000 |
| Rabbit anti-SEPHS2 | Proteintech | 14109-1-AP, 1:1000 |
| Rabbit anti-SCD | Alpha Diagnostic International Inc | SCD11-A, 1:1000 |
| Mouse anti-SCD | Abcam | ab19862, 1:1000 |
| Mouse anti-Cas9 | Cell Signaling | 14697, 1:250 |
| Rabbit anti-Ku80 | Cell Signaling | 2180, 1:400 |
| Mouse Envision secondary antibody HRP | Agilent | K4001 |
| Rabbit Envision secondary antibody HRP | Agilent | K4003 |
| **Oligonucleotides and other sequence-based reagents** | | |
| **gRNA** | **5′-3′** | |
| Non-targeting control | GTAGCGAACGTGTCCGGCGT | |
| ACSL3 | GAGCTATCATCCACTCGGCCC | |
| LRP8 | GGCCACTGCATCCACGAACGG | |
| PSTK | AAACTGATCAGACACTCCGA | |
| SCD | GCAGCCGAGCTTTGTAAGAG | |
| SEPHS2 | GAGGGACGGCAGTGACCGG | |
| SEPSECS | AACCGCGAGAGCTTCGCGG | |
| **PCR primers** | **Forward primer (5′-3′)** | **reverse primer (5′-3′)** |
| ACACA | TCACACCTGAAGACCTTAAAGCC | AGCCCACACTGCTTGTACTG |
| ACSL4 | CATCCCTGGAGCAGATACTCT | TCACTTAGGATTTCCCTGGTCC |
| ACTB | GGCATGGGTCAGAAGGATT | ACATGATCTGGGTCATCTTCTC |
| AIFM2 | AGACAGGGTTCGCCAAAAAGA | CAGGTCTATCCCCACTACTAGC |
| DHFR | CGCTCAGGAACGAGTTCAAGT | TGCCAATTCCGGTTGTTCAATAA |
| DHODH | CCACGGGAGATGAGCGTTTC | CAGGGAGGTGAAGCGAACA |
| ELOVL3 | CTGTTCCAGCCCTATAACTTCG | GAATGAGGTTGCCCAATACTCC |
| FADS1 | CTACCCCGCGCTACTTCAC | CGGTCGATCACTAGCCACC |
| FADS2 | GACCACGGCAAGAACTCAAAG | GAGGGTAGGAATCCAGCCATT |
| FASN | AAGGACCTGTCTAGGTTTGATGC | TGGCTTCATAGGTGACTTCCA |
| LMNB1 | AAGCAGCTGGAGTGGTTGTT | TTGGATGCTCTTGGGGTTC |
| SCD | TCTAGCTCCTATACCACCACCA | TCGTCTCCAACTTATCTCCTCC |
| SEPHS2 | GCGGCTGAGGAAGGAGGGACG | ACGGCGCTGTCCGGCATTATG |
| TBP | AGTGACCCAGCATCACTGTTT | TAAGGTGGCAGGCTGTTGTT |
| **Chemicals, enzymes and other reagents** | | |
| DMEM/F-12 | Thermo Fisher Scientific | 21331046 |
| Glutamine | Thermo Fisher Scientific | 25030149 |
| Foetal bovine serum (FBS) | Thermo Fisher Scientific | 10270106 |
| MycoAlert Mycoplasma Detection Kit | Lonza | LT07-318 |
| Puregene Gentra Kit | Qiagen | 158845 |
| Promega Geneprint Kit | Promega | B9510 |
| CAY10566 | Cayman Chemicals | 10012562 |
| Sulforhodamine B | Merck | 230162 |
| Trichloroacetic acid | Merck | T6399-500G |
| Amicon Ultra-15 Ultracel-10 Centrifugal Filter Unit | Merck | UFC901024 |
| Meth-Prep II | Alpha Aesar | 43284 |
| BODIPY 581/591 C11 lipid peroxidation sensor | Thermo Fisher Scientific | D3861 |

| Reagent/resource | Reference or source | Identifier or catalogue number |
|---|---|---|
| RIPA buffer | Millipore | 20-188 |
| Laemmli buffer | Bio-rad | 1610747 |
| SDS-polyacrylamide gel | Invitrogen | NP0336BOX |
| Nitrocellulose membrane | Amersham | 10600001 |
| Waters CSH C18 analytical column | Waters | 186005297 |
| Na$_2$SeO$_3$ | Merck | S5261 |
| Deferoxamine | Merck | D9533 |
| Ferrostatin-1 | Merck | SML0583 |
| oleic acid | Merck | O1008 |
| RSL3 | Merck | SML2234 |
| Incucyte® Cytotox Green Dye | Sartorius | 4633 |
| FAME Standard Mixture | Merck | CRM47885 |
| JetPrime | Polyplus | 101000015 |
| Polybrene | Merck | H9268 |
| Puromycin | Merck | P8833 |
| AlbuMAX II lipid-rich BSA | Thermo Fisher Scientific | 11021029 |
| G-418S sulphate | Formedium | G-418S |
| Luciferin | Abcam | ab143655 |
| SuperScript VILO MasterMix | Thermo Fisher Scientific | 11755-050 |
| Applied Biosystems Fast SYBR Green MasterMix | Thermo Fisher Scientific | 385612 |
| Mini Cell Strainer II | Funakoshi | HT-AMS-04002 |
| antigen retrieval using ER2 solution | Leica | AR9640 |
| on-board dewaxing solution | Leica | AR9222 |
| Leica wash buffer | Leica | AR9590 |
| Intense R kit | Leica | DS9263 |
| mouse Ig blocking solution | Vector Labs | MKB-2213 |
| haematoxylin z | CellPath | RBA-4201-00A |
| DPX mountant | CellPath | SEA-1300-00A |
| **Software** | | |
| Xcalibur version 4.3 | https://www.thermofisher.com/ | |
| Compound Discoverer v.3.1 | https://www.thermofisher.com/ | |
| LipiDex | https://www.ncqbcs.com/resources/software/ | |
| MSConvert | https://proteowizard.sourceforge.io/ | |
| Applied Biosystems Genemapper v4.1 software | https://www.thermofisher.com/ | |
| Incucyte 2022B software | https://www.sartorius.com/ | |
| MassHunter Quantitative Analysis software | https://www.agilent.com/ | |
| HALO image analysis software | https://indicalab.com/halo/ | |
| GraphPad Prism 9.0 | https://www.graphpad.com/ | |
| **Other** | | |
| Li-COR Odyssey DLx imaging system | https://www.cambridgescientific.com/ | |
| Incucyte S3 | https://www.sartorius.com/ | |
| Agilent 7890 GC chromatograph and 7693 autosampler | https://www.agilent.com/ | |
| Agilent 7000 GC/MS | https://www.agilent.com/ | |
| Thermo Fisher Scientific Ultimate 3000 binary UPLC | https://www.thermofisher.com/ | |

| Reagent/resource | Reference or source | Identifier or catalogue number |
|---|---|---|
| Q Exactive Orbitrap mass spectrometer | https://www.thermofisher.com/ | |
| Heated Electrospray Ionisation (HESI-II) source | https://www.thermofisher.com/ | |
| Applied Biosystems 3130xl DNA analyser | https://www.thermofisher.com/ | |
| Casy cell counter | https://www.ols-bio.de/ | |
| Tecan Spark multiplate reader | https://lifesciences.tecan.com/ | |
| Leica Bond Rx autostainer | https://www.leicabiosystems.com/ | |

## Cell culture

Cultures of breast cancer cells (BT549 [RRID:CVCL_1092], Cal120 [RRID:CVCL_1104], EO771 [RRID:CVCL_GR23], MCF7 [RRID:CVCL_0031], MDA-MB-231 [RRID:CVCL_0062] and MDA-MB-468 [RRID:CVCL_0419]), human dermal fibroblast (DF), human mammary fibroblast (MF) and cancer-associated fibroblast (CAF) were maintained in DMEM/F-12 (Thermo Fisher Scientific, #21331046) supplemented with 2 mM Glutamine (Thermo Fisher Scientific, #25030149) and 10% foetal bovine serum (FBS, Thermo Fisher Scientific, #10270106). MF and CAF cell lines were kindly provided by Prof. Akira Orimo and have been previously characterised (Kojima et al, 2010; Hernandez-Fernaud et al, 2017). All cell lines were tested negative for mycoplasma (Venor GeM qOneStep Mycoplasma Detection Kit). Cell lines were authenticated using genomic DNA extracted with Puregene Gentra Kit and multiplexed using the Promega Geneprint Kit and multiplexed with a STR-based method (Promega Geneprint System). Samples were run on an Applied Biosystems 3130xl DNA analyser and the results analysed using the Applied Biosystems Genemapper v4.1 software. Profiles were matching the references reported by ATCC (LGC standards), Cellosaurus and DSMZ databases.

## Conditioned medium

To condition medium from high-density cultures of BT549, Cal120, EO771, MCF7, MDA-MB-231, MDA-MB-468, DF, MF and CAF cell lines, $6 \times 10^6$ cells were seeded in DMEM/F-12 with 10% FBS in a dish with a diameter of 145 mm. The day after seeding, the medium was replaced with 20 ml of DMEM/F-12 without FBS and conditioned for 2 days. For non-targeting control (NTC) and SCDko MDA-MB-468 cells, $5 \times 10^6$ and $1.1 \times 10^7$ cells were seeded, respectively, to condition the medium. For experiments with acute SCD inhibition, $2 \times 10^7$ MDA-MB-468 cells were seeded in DMEM/F-12 with 10% FBS in a dish with a diameter of 145 mm. The day after seeding the medium was replaced by DMEM/F-12 without FBS and supplemented with 200 nM CAY10566 (SCD inhibitor, Cayman chemicals, #10012562). After 18 h of incubation with 200 nM CAY10566, cells were washed once with PBS, and 15 ml DMEM/F-12 medium without FBS was conditioned for 2 h. For experiments shown in Fig. 2A, $6 \times 10^6$ MDA-MB468 cells were seeded in DMEM/F-12 with 10% FBS in a dish with a diameter of 145 mm. The day after seeding, the medium was replaced with 20 ml of DMEM/F-12 or selenite-free Plasmax™ (Vande Voorde et al, 2019), both supplemented with 10% FBS and conditioned for 2 days. The conditioned medium was diluted with three volumes of serum-free unconditioned medium and used for the colony-forming assays.

To generate mock medium, cell culture medium was incubated in a cell culture dish maintained at 37 °C without cells for the respective experimental time periods (2 h or 2 days).

## Colony formation assays

Colony formation assays were performed as described previously (Vande Voorde et al, 2019). For MDA-MB-468 cells, we seeded 5000 cells/well in a six-well plate with 2 ml/well of DMEM/F-12 supplemented with 2.5% FBS. To test the effects of lipid extracts from medium conditioned by MDA-MD-468 cells, 1000 cells/well were seeded in a 24-well plate with 0.5 ml of medium/ well. After 7 days of incubation, colonies from MDA-MB-468 cells were fixed by replacing the medium with a solution of 3% trichloroacetic acid in water (Merck, #T6399-500G), rinsed twice with water, stained with 0.057% Sulforhodamine B solution in 1% acetic acid (Merck, #230162) and washed twice with a 1% acetic acid solution in water. For BT549, Cal120, MCF7, MDA-MB-231 and EO771 cells, 500 cells/well were seeded in a six-well plate and fixed after 14 days of culture with the exception of EO771 cells that were fixed 7 days after seeding. Plates stained with Sulforhodamine B were scanned with Li-Cor Odyssey® DLx imaging system and the fluorescent signal quantified with ImageJ as previously described (Vande Voorde et al, 2019). In figures reporting colony area or number, each data point represents an independent colony-forming assay mean of two or three replicate wells.

## Cell proliferation assay

To determine the number of NTC control and SCDko cells during the medium-conditioning experiments, $3.32 \times 10^5$ NTC control cells per well and $7.3 \times 10^5$ SCDko cells per well were seeded in a six-well plate with DMEM/F-12 containing 10% FBS. One day after seeding, cells were trypsinized and counted with a Casy counter. At this time, the medium of parallel plates seeded and incubated in the same conditions was replaced with DMEM/F-12 without FBS, and 2 days after, the cells were counted with a Casy counter.

For proliferation assays, NTC control or SCDko MDA-MB-468 cells were seeded at $3 \times 10^4$/well in a six-well plate with DMEM/F-12 supplemented with 2.5% FBS, 50 nM Na₂SeO₃ (Merck, #S5261), 2.5 μM Deferoxamine (Merck, # D9533), 1 μM Ferrostatin-1 (Merck, #SML0583), 10 μM oleic acid (Merck, #O1008), or vehicles (0.05% DMSO and 0.02% ethanol). For proliferation assays with GPX4 inhibitor (RSL3), cells were seeded and incubated as described above and supplemented with 50 nM RSL3 (Merck, #SML2234) and 10% FBS. After 5 days, cells were counted with a

## Live-cell imaging

For Live-cell imaging, MDA-MB-468 cells were seeded at $5 \times 10^3$/ well in a 24-well plate with DMEM/F-12 containing 2.5% FBS and 3 nM Incucyte® Cytotox Green Dye (Sartorius, #4633), 50 nM $Na_2SeO_3$ or 2 µM Ferrostatin-1 as indicated in figures. Twenty-four hours after seeding and every 2 h thereafter, nine images per well were acquired with phase contrast and green fluorescence (acquisition time: 200 ms) with a 10× objective using an Incucyte S3 (Sartorius). To quantify the number of dead cells, the fluorescent objects were counted using Incucyte 2022B software.

## Medium fractionation and heat inactivation

Medium conditioned by MDA-MB-468 cells was loaded into size exclusion columns (Amicon® Ultra-15 Ultracel-10 Centrifugal Filter Unit, Merck, #UFC901024) centrifuged at $4000 \times g$ for 30 min and the fractions stored at $-20\,°C$ until further analysis. The concentrated fraction (CM > 10 kDa) was diluted 1:40 in unconditioned DMEM/F-12 medium, supplemented with 2.5% FBS and used for colony-forming assays. The column flow-through (CM < 10 kDa) was supplemented with 2.5% FBS and used undiluted for colony-forming assays.

For heat inactivation and protein denaturation, the concentrated fraction (CM > 10 kDa) was incubated for 15 min at 95 °C, afterwards diluted 1:40 in unconditioned DMEM/F-12 medium supplemented with 2.5% FBS and used for colony-forming assays.

## Lipid extraction

Lipids were extracted from the concentrated fraction (CM > 10 kDa) of the conditioned medium following the Bligh and Dyer method and used for colony-forming assays or directly from the conditioned medium with a methyl tertiary-butyl ether (MTBE) solution and used for lipidomic analysis(Matyash et al, 2008). For the Bligh and Dyer extraction, 250 µl of the concentrated fraction (CM > 10 kDa) from the size exclusion columns were mixed with 960 µl of a 1:2 chloroform:methanol mixture. Afterwards, 310 µl chloroform and 310 µl water were added, and the solution was mixed to achieve a phase separation. The lower, chloroform-containing phase was transferred to a new glass vial, dried at room temperature by nitrogen flow and resuspended in 25 µl of ethanol. In total, 5 µl of lipid-containing ethanol was supplemented to 2 ml of DMEM/F-12 with 2.5% FBS to perform colony-forming assays.

For the MTBE extraction, 5 ml MTBE and 1 ml methanol were added to 2.5 ml of conditioned medium. After mixing the solution three times for 30 s, the upper phase was transferred to a new glass vial, dried under nitrogen flow at 35 °C and resuspended in 25 µl ethanol. In all, 1 µl of the lipid ethanol solution was supplemented to 1 ml of DMEM/F-12 with 2.5% FBS and used for colony-forming assays.

## Lipidomic analysis

For lipidomic analysis, $5 \times 10^5$ NTC control and SCDko cells were seeded in 20 ml of DMEM/F-12 supplemented with 2.5% FBS in plates of 145 mm diameter. After 2 days, cells were scraped off the plate in the culture medium, collected in a tube and centrifuged at $1000 \times g$ for 3 min. The cell pellet was resuspended in 1 ml of ice-cold PBS, transferred to 1.5 ml Eppendorf vial, and centrifuged at $10,000 \times g$ for 10 s. Lipids were extracted from the cell pellet with 200 µl of butanol:methanol solution (1:1) centrifuged at $16,000 \times g$ for 10 min, and analysed using high-resolution mass spectrometry.

For lipidomic analysis of low-density cultures, $1.6 \times 10^5$ MDA-MB-468 cells were seeded in plates of 145 mm diameter with 20 ml of mock or conditioned DMEM/F-12 supplemented with 2.5% FBS and 2 µM Ferrostatin-1. Vehicle control or 10 µM oleic acid was added as indicated in Fig. EV2A. After 2 days, cells were scraped off the plate in the culture medium, collected in a tube and centrifuged at $1000 \times g$ for 3 min. The cell pellet was resuspended in 1 ml of ice-cold PBS, quickly transferred to 1.5-ml Eppendorf vial, and centrifuged at $10,000 \times g$ for 10 s. Lipids were extracted from the cell pellet with 100 µl of butanol:methanol solution (1:1), centrifuged at $16,000 \times g$ for 10 min, and analysed using high-resolution mass spectrometry.

For the lipidomic analysis of media, 10 µl of conditioned medium enriched (CM > 10 kDa) or depleted (CM < 10 kDa) fractions were diluted with 190 µl of butanol:methanol solution (1:1), centrifuged at $16,000 \times g$ for 10 min, and analysed using high-resolution mass spectrometry.

For the lipidomic analysis of interstitial fluid (IF), 5 µl of IF from individual NTC or SCDko tumours were diluted with 45 µl of butanol:methanol solution (1:1), centrifuged at $16,000 \times g$ for 10 min, and analysed using high-resolution mass spectrometry.

Lipidomic analyses were performed using a Thermo Fisher Scientific Ultimate 3000 binary UPLC coupled to a Q Exactive Orbitrap mass spectrometer equipped with a Heated Electrospray Ionisation (HESI-II) source (Thermo Fisher Scientific, Massachusetts, USA). For each sample, MS data were acquired using full MS/ dd-MS2 in positive and negative modes to maximise the number of detectable species. Details on the parameters for the MS methods using different polarities are provided in Table 1. Chromatographic separation was achieved using a Waters CSH C18 analytical column ($100 \times 2.1$ mm, 1.7 µm) maintained at 55 °C. The mobile phase consisted of 60:40 (v/v) acetonitrile:water containing 10 mM ammonium formate and 0.1% formic acid (phase A) and 90:10 (v/v) isopropanol:acetonitrile containing 10 mM ammonium formate and 0.1% formic acid (phase B) at a flow rate of 400 µl/min. The gradient elution consisted of 30% B for 0.5 min, increasing linearly to reach 50% at 4 min, then 80% at 12 min, then 99% B at 12.1 min then held at 99% for 1 min, then returned to starting condition in 1 min and kept constant for 2 min. The total run time was 20 min. The whole system was controlled by Xcalibur version 4.3. Quality control (QC) samples prepared by mixing equal volumes of experimental samples were injected at regular interval throughout the whole batch to monitor the instrument performance. Lipidomics data were analysed with Compound Discoverer v.3.1 (Thermo Fisher Scientific) and LipiDex(Hutchins et al, 2019), an open-source software suite available at http://www.ncqbcs.com/ resources/software/. Briefly, raw files were loaded into Compound Discoverer and processed using two workflows (aligned and unaligned) as previously described (Hutchins et al, 2018). Compound result tables were exported for further processing using the "Peak picking" tab in Lipidex. In addition, the raw data files were converted to .mgf files using MSConvert (ProteoWizard, P.

**Table 1. Parameters of the mass spectrometry methods using positive and negative polarities.**

| Parameters | Positive mode | Negative mode |
|---|---|---|
| **Source parameters** | | |
| -Capillary temperature | 262.5 °C | 300.0 °C |
| -Sheath gas | 50.0 | 50.0 |
| -Aux gas | 12.5 | 7.0 |
| -Spare gas | 2.5 | 5.0 |
| -Probe heater temperature | 425.0 °C | 300.0 °C |
| -S-Lens RF level | 50 | 50 |
| **Full MS** | | |
| -Chromatographic peak width | 15 s | 10 s |
| -Default charge | 1 | 1 |
| -Resolution | 70,000 | 70,000 |
| -AGC target | 3e6 | 3e6 |
| -Maximum IT | 100 ms | 100 ms |
| -Scan range | 240–1200 $m/z$ | 240–1600 $m/z$ |
| **dd-MS2** | | |
| -Resolution | 17,500 | 17,500 |
| -AGC target | 2e5 | 1e5 |
| -Loop count | 5 | 5 |
| -Isolation window | 1.4 $m/z$ | 1.4 $m/z$ |
| -Stepped nce | 25 | 25 |
| -Spectrum data | Centroid | Centroid |
| **dd-settings** | | |
| -Minimum AGC | 4e3 | 1e3 |
| -Exclude Isotopes | On | On |
| -Dynamic exclusion | 8 s | 5 s |

Mallick, Stanford University) (Chambers et al, 2012) and imported into the "Spectrum Searcher" tab in Lipidex, where the following libraries were searched "LipidBlast_Formate", "LipiDex_HCD_Formate", "LipiDex_Splash_ISTD_Formate", "LipiDex_HCD_ULCFA" using the default search tolerances for MS1 and MS2. For a lipid to be considered identified it required a minimum of 75% spectral purity, an MS2 search dot product score of at least 500 and a reverse dot product of at least 700.

For the determination of the total fatty acid content of the lipid species, samples were analysed using GC-MS after transesterification using a methanolic solution of (m-trifluoromethylphenyl) trimethyl ammonium hydroxide (Meth-Prep II), to provide fatty acid methyl esters (FAME) in a one-step reaction followed by monitoring of the generated FAME using GC/MS in SIM mode. Briefly, samples were first spiked with heptadecanoic (C17:0) as an internal standard, dried under nitrogen stream and resuspended in 27 μL of chloroform. Overall, 3 μL of Meth-Prep II were added, and samples were mixed and analysed within 2 days. Analysis was performed using an Agilent 7890 GC chromatograph and 7693 autosampler coupled with an Agilent 7000 GC/MS provided with (Agilent, CA, USA). High-purity helium (99.999%) was used as carrier gas at an initial flow rate of 1 mL/min increased to 2 mL/min in 12 s and kept constant for 1 min and then returned to 1 mL/min for the rest of the run time. The chromatographic

column used was Phenomen BPX70 (60 m × 250 μm, 0.25 μm) (Fisher Scientific, USA). The injector was set at a temperature of 300 °C, a pressure of 21 psi and a septum purge flow of 3 mL/min. Injection was performed in a pulsed splitless mode at 60 psi for 1.2 min and a purge flow to split vent of 50 mL/min at 1.2 min. The injection volume was 1 μl. The oven temperature was set to an initial temperature of 100 °C held for 2 min followed by a linear increase to 172 °C at 8 °C/min (held for 6 min), then 196 °C (held for 9 min) then 204 °C (held for 15 min). The total run time was 45 min. For the separation of the palmitoleic acid and oleic acid isomers, a longer chromatographic run was used applying the same starting conditions, followed by a stepwise increase in temperature initially to 160 °C at 8 °C/min (held for 6 min), then to 185 °C at 0.5 °C/min (held for 9 min), then 204 °C at 8 °C/min (held for 15 min). The MS was operated in electron ionisation (EI) mode at 70 eV, and the MSD transfer line was set to 260 °C. MS data were acquired in SIM mode for monitoring of the individual fatty acid derivatives. MS recording started at a cut-off of 8 min. Data were processed using Agilent MassHunter Quantitative analysis software (Agilent, CA, USA). Initially, fatty acids were annotated by comparison to FAME Standard Mixture (Merck Life Science UK Limited, UK) using a combination of retention time and fragmentation pattern using full scan. A SIM mode for selected fragments was applied to improve the sensitivity of detection.

## Cloning and CRISPR-based gene editing

The following gRNA sequences: Non-Targeting Control (NTC): 5′-GTAGCGAACGTGTCCGGCGT-3′, ACSL3 5′-GAGCTATCATC-CACTCGGCCC-3′, LRP8 5′-GGCCACTGCATCCACGAACGG-3′, *PSTK*: 5′-AAACTGATCAGACACTCCGA-3′, *SCD*: 5′-GCAGCC-GAGCTTTGTAAGAG-3′, *SEPHS2*: 5′-GAGGGACGGCAGT-GACCGG-3′, *SEPSECS*: 5′-AACCGCGAGAGCTTCGCGG-3′ were cloned into lentiCRISPRv2 vector (Addgene, Plasmid #52961) using BsmBI restriction sites. For lentivirus production, $2 \times 10^6$ HEK293T cells were transfected with 5 μg lentiCRISPR plasmid (NTC or on target), 1 μg pVSV-G (viral envelope) and psPAX2 (2nd generation lentiviral packaging plasmid) using JetPrime (Polyplus, #101000015) according to the manufacturer's protocol. Six hours after transfection the medium was replaced, incubated for 18 h and harvested for viral infection of recipient cells. The recipient MDA-MB-468 cells were cultured for 24 h with lentivirus-containing medium supplemented with 8 μg/ml Polybrene, for an additional 24 h with fresh medium and selected for 4 days with medium supplemented with 0.75 μg/mL puromycin.

After infection, SCDko pools and clones were cultured in DMEM/F-12 with 10% FBS and 0.8 g/L AlbuMAX II lipid-rich BSA (Thermo Fisher Scientific, # 11021029). After infection, ACSL3 and LRP8ko pools were cultured in DMEM/F-12 with 10% FBS and 50 nM Na$_2$SeO$_3$. In maintenance culture, ACSL3 and LRP8ko clones were grown with 2.5% FBS. sgPSTK, sgSEPHS2 and sgSEPSECS cell lines were cultured in DMEM/F-12 with 2.5% FBS, 0.8 g/l AlbuMAX II lipid-rich BSA and 1 μM Ferrostatin-1 (MERCK, # SML0583-5MG) unless otherwise indicated.

Firefly luciferase expression cassette (fluc + ) was excised from pGL4.50 vector (Promega, # E1310) using NdeI and BamHI restriction enzymes and cloned into pLenti6NEO vector using the same restriction sites. The integration of the cloned fragment was confirmed by Sanger sequencing. Virus production and infection of MDA-MB-468 target cells were performed as described above for lentiCRISPR vectors. After the infection cells were selected for

7 days with 500 µg/ml G-418S sulphate (Formedium, # G-418S). Luciferase activity was checked by supplementing the culture medium with luciferin (100 µg/ml, Abcam, #ab143655), and the bioluminescent signal was assessed with a Tecan Spark multiplate reader.

## C11 BODIPY lipid peroxidation assay

In total, $5 \times 10^5$ NTC control and SCDko cells were seeded in DMEM/F-12 supplemented with 2.5% FBS in plates of 145 mm diameter. 2 days after seeding cells were incubated for 30 min with 1 µM BODIPY 581/591 C11 lipid peroxidation sensor (Thermo Fisher Scientific, #D3861). Cells were then washed with PBS, detached by trypsinisation, pelleted by centrifugation, and resuspended in 400 µL PBS with 1 µg/ml DAPI used to stain dead cells. Peroxidation of the BODIPY probe in live cells was measured by fluorescent-activated flow cytometry (FACS). The fluorescence of the reduced probe was measured at 581/591 nm (excitation/ emission, Texas Red filter set) and the oxidised probe at 488/ 510 nm (FITC filter set). The ratio between the signals at 510 and 591 nm (oxidised/reduced) was used as a readout for lipid peroxidation. Overall, $5 \times 10^5$ NTC control and SCDko cells exposed to 100 nM RSL3 for 2 h were used as positive control for lipid peroxidation.

## Immunoblotting

Cells were lysed in radioimmunoprecipitation assay (RIPA) buffer (Millipore, #20-188). Lysates were incubated in Laemmli buffer (Bio-rad, #1610747) at 95 °C for 3 min and loaded onto a SDS-polyacrylamide gel (4–12%, Invitrogen NuPAGE, #NP0336BOX). After size separation, proteins were transferred onto nitrocellulose membrane (0.2 µM pore size, Amersham, # 10600001), and the membrane was blocked for 1 h at room temperature by 5% nonfat dry milk (in tris-buffered saline with 0.01% Tween (TBST)). Membranes were incubated overnight at 4 °C in a 5% BSA/TBST solution of primary antibody at the following dilutions: vinculin, 1:2000, Merck, # SAB4200080; GPX4, 1:1000, Abcam, # ab125066; ACSL3, 1:1000, Abcam, #ab151959; LRP8, 1:1000, Abcam, # ab108208; SEPHS2, 1:1000, Proteintech, # 14109-1-AP; SCD, 1:1000, Alpha Diagnostic International Inc. # SCD11-A in Figs. 4E and 5G for MDA-MB-468; SCD, 1:1000, Abcam, # ab19862 in Figs. 4H and 5G for BT549, CAL120, MDA-MB-231, MCF7. The next day membranes were washed and stained with species-specific near-infrared fluorescent, secondary antibodies (Li-COR) for 1 h at room temperature. After additional washing steps, membranes were imaged with Li-Cor Odyssey® DLx imaging system.

## RNA isolation and qRT-PCR analysis

The same number of MDA-MB-468 cells ($1.6 \times 10^5$) were seeded per well in a six-well plate or in plates of 145 mm diameter for high- and low-density cultures, respectively. For BT549, CAL120, MDA-MB-231 and MCF7 cells, low- and high-density cultures were achieved by seeding $4 \times 10^4$ cells in plates of 145 mm diameter and $1.6 \times 10^5$ per well in a six-well plate, respectively. Two days after seeding, the cells were washed with ice-cold PBS, scraped off the plate and pelleted at 4 °C at $10{,}000 \times g$ for 30 s. RNA was isolated from cell pellets and tumour fragments following the kit manufacturer's protocol (Qiagen RNeasy, # 74104). Overall, 500 ng RNA was used for cDNA synthesis (SuperScript VILO MasterMix, # 11755-050). In total, 5 ng cDNA and 8 pmol of each primer were used in each quantitative real-time polymerase chain reaction (Applied Biosystems Fast SYBR Green MasterMix, #4385612). Primer sequences were obtained from primer bank (https:// pga.mgh.harvard.edu/primerbank/). A standard curve method with linear regressions $R^2 > 0.8$ was used to obtain relative quantification of mRNAs expression.

## Xenograft experiments

Animal experiments were performed in accordance with UK Home Office Regulations and Directive 2010/63/EU and subjected to review by the Animal Welfare and Ethical Review Board of the University of Glasgow (Project licence PP6345023 and P38F4A67E). In house-bred or commercially sourced (Charles River) NOD SCID gamma (NSG) mice were housed at temperatures between 19 and 23 °C in ventilated cages with ad libitum food and water access and 12 h light/dark cycles. To minimise pain and distress, Rimadyl was added to the drinking water 24 h prior to xenotransplantation and removed 3 days post implantation. For experiments with sgSEPHS2 and sgSEPSECS MDA-MB-468 cells, 24 female NSG mice aged between 81–170 days were anaesthetised and transplanted into the inguinal mammary fat pad with 50 µl per transplantation of 1:1 PBS:Matrigel solution containing $3 \times 10^6$ luciferase-expressing mycoplasma-negative cells. Groups of eight age-matched animals were randomly assigned to the three experimental groups (NTC, sgSEPHS2 and sgSEPSECS). All the mice were transplanted with cells bilaterally. Tumours were measured by calliper three times/week by animal technicians blinded to the scientific outcome. The tumour volume was calculated using the equation [length × width$^2$]/2, where width is the smaller of the two dimensions.Thirty-eight days post-transplantation mice were culled, and mammary tumours were harvested. Tissues were frozen at −80 °C or fixed in 10% buffered formalin solution and embedded in paraffin.

To assess the metastatic seeding of breast cancer cells, $2 \times 10^6$ luciferase-expressing mycoplasma-negative MDA-MB-468 cells were passed through a 70-µm strainer, resuspended in 100 µL of 4.5% BSA PBS solution (pH 7.4), and injected into the tail vein of 24 female NSG mice aged to 79 days. Mice were randomly assigned to three experimental groups (NTC, sgSEPHS2 and sgSEPSECS) consisting of eight mice per group. Mice were imaged by IVIS bioluminescence at the specified times (indicated in figures) and culled 7 weeks after injection prior to reaching the clinical endpoint. Organs were harvested, fixed in 10% buffered formalin solution, and embedded in paraffin for immunohistochemistry.

For experiments with SCDko MDA-MB-468 cells, 12 female NSG mice of 113–137 days of age were anaesthetised and transplanted unilaterally into the inguinal mammary fat pad with 50 µl per transplantation of 1:1 PBS:Matrigel solution containing $3 \times 10^6$ cells (mycoplasma-negative). Two groups of six age-matched animals were randomly assigned to the NTC and SCDko experimental groups. Tumours were measured by calliper three times/week by animal technicians blinded to the scientific outcome. The tumour volume was calculated using the equation [length × width$^2$]/2, where width is the smaller of the two

**The paper explained**

**Problem**

The limited availability of therapeutic options for patients with triple-negative breast cancer (TNBC) contributes to the high rate of metastatic recurrence and poor prognosis.

**Results**

This study shows that TNBC cells under selenium starvation die of ferroptosis selectively when cultured at low cell density and this phenotype can be rescued by transferring medium conditioned by breast cancer cells cultured at high density. Monounsaturated fatty acid (MUFA) containing lipids derived from Stearoyl-CoA Desaturase (SCD) were identified as the anti-ferroptotic factors released by TNBC cells. In analogy to cells cultured at low-density TNBC cells in circulation, downregulate SCD expression and require the antioxidant action of selenoproteins to overcome ferroptosis during the metastatic cascade. Indeed, the inhibition of selenocysteine-tRNA$^{sec}$ biosynthesis effectively impedes TNBC lung metastasis.

**Impact**

Preclinical and clinical studies assessed selenium supplementation as an antioxidant cancer-preventive intervention, and selenium is frequently found in multivitamin/multimineral food supplements. However, this study shows that the antioxidant activity of selenium is required for efficient metastatic dissemination, identifying the incorporation of selenium into selenoproteins as a novel and druggable therapeutic target to prevent TNBC metastasis.

dimensions. Forty-three days post-transplant, mice were culled, and tumours were harvested. Tissue fragments were immediately processed for the isolation of interstitial fluid, or frozen on dry ice and stored at $-80\,°C$ or fixed in 10% buffered formalin solution and embedded in paraffin.

## Isolation of interstitial fluid

Tissues freshly isolated from mammary fat pad tumours were cut by scalpel in four slices of ~2–4 mm thickness and each slice was transferred to an individual cell strainer (Mini Cell Strainer II, Funakoshi, #HT-AMS-04002). The cell strainers were then placed into 1.5 ml Eppendorf tubes and centrifuged at $100 \times g$ for 10 min at $4\,°C$ to separate the interstitial fluid. The interstitial fluid from four slices was merged into one sample representative of one tumour, frozen on dry ice and stored at $-80\,°C$ until lipidomic analysis.

## Immunohistochemistry

All immunohistochemistry (IHC) staining was performed on 4-μm formalin fixed paraffin embedded sections (FFPE) heated at $60\,°C$ for 2 h.

The following antibodies were used to stain sections with a Leica Bond Rx autostainer: Cas9 (14697, Cell Signaling) and Ku80 (2180, Cell Signaling). All FFPE sections underwent on-board dewaxing (AR9222, Leica) and antigen retrieval using ER2 solution (AR9640, Leica) for 20 min at $95\,°C$. Sections were rinsed with Leica wash buffer (AR9590, Leica) before peroxidase block was

performed using an Intense R kit (DS9263, Leica) for 5 min. After rinsing with wash buffer, mouse Ig blocking solution (MKB-2213, Vector Labs) was applied to Cas9 sections for 20 min. Sections were rinsed with wash buffer and then the primary antibody applied at an optimal dilution (Cas9, 1:250; Ku80, 1:400) for 30 min. The sections were rinsed with wash buffer and appropriate secondary antibody was applied for 30 min (Cas9, Mouse Envision (Agilent, K4001); Ku80, Rabbit Envision (Agilent, K4003). The sections were rinsed with wash buffer and visualised using DAB with the Intense R kit.

The sections were washed in water and counterstained with haematoxylin z (RBA-4201-00A, CellPath). To complete IHC staining FFPE sections were rinsed in tap water, dehydrated through graded ethanol's, and placed in xylene. The stained sections were coverslipped in xylene using DPX mountant (SEA-1300-00A, CellPath). Slides were scanned at ×20 magnification and analysed with HALO image analysis software (Indica Labs).

## Analysis of SCD expression in patient-derived samples

Publicly available datasets on ctcRbase (http://www.origin-gene.cn/database/ctcRbase/search.html) were interrogated for the expression of SCD in samples obtained from breast cancer patients. Datasets from four breast cancer studies were interrogated (Zhao et al, 2020; Gkountela et al, 2019; Szczerba et al, 2019; Aceto et al, 2014; Yu et al, 2014; Data ref: Szczerba et al, 2019; Aceto et al, 2014; Yu et al, 2014). The data were exported from ctcRbase database and visualised with GraphPad Prism 9.0 or later versions.

## Statistical analysis

Independent experimental replicate numbers and statistical tests are described in the figure legends. For all *t* tests and ANOVA analyses, GraphPad Prism 9.0 or later versions were used. For the statistical analysis in Fig. 6F, we used the CGGC permutation test https://bioinf.wehi.edu.au/software/compareCurves/ to assess the significant differences between growth curves. Data points for animal experiments were excluded before the end of the experiment if animals had to be culled to comply with the animal licence or for husbandry-related reasons. No data were excluded on the basis of experimental outcomes.

## Data availability

Metabolomics data: MassIVE MSV000094784 (https://massive.ucsd.edu/ProteoSAFe/dataset.jsp?accession=MSV000094784).

The source data of this paper are collected in the following database record: biostudies:S-SCDT-10_1038-S44321-024-00142-x.

## Peer review information

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

## Acknowledgements

The authors thank the Core Services and Advanced Technologies at the Cancer Research UK Scotland Institute, and particularly the Metabolomics, Biological Services Unit, Histology Service, Molecular Technologies and Advanced Imaging Resource. The authors acknowledge Catherine Winchester (CRUK Scotland Institute) for critically reviewing this manuscript and all the members of the Oncometabolism lab for constructive discussions. This work was funded by Cancer Research UK core funding awarded to the CRUK Scotland Institute (grant number A31287), Stand Up to Cancer campaign for CRUK awarded to SZ (grant number A29800), Breast Cancer Now awarded to SZ (grant number 2018NovPR102), Cancer Research UK core funding awarded to KB (grant number A29799) and Cancer Research UK core funding awarded to ST (grant number A23982).

## Author contributions

**Tobias Ackermann**: Conceptualisation; Data curation; Formal analysis; Investigation; Visualisation; Writing—original draft; Writing—review and editing. **Engy Shokry**: Methodology. **Ruhi Deshmukh**: Investigation. **Jayanthi Anand**: Investigation. **Laura C A Galbraith**: Investigation. **Louise Mitchell**: Supervision. **Giovanny Rodriguez-Blanco**: Methodology. **Victor H Villar**: Methodology. **Britt Amber Sterken**: Investigation. **Colin Nixon**: Methodology. **Sara Zanivan**: Supervision; Funding acquisition. **Karen Blyth**: Supervision; Funding acquisition. **David Sumpton**: Methodology. **Saverio Tardito**: Conceptualisation; Data curation; Supervision; Funding acquisition; Visualisation; Writing—original draft; Writing—review and editing.

Source data underlying figure panels in this paper may have individual authorship assigned. Where available, figure panel/source data authorship is listed in the following database record: biostudies:S-SCDT-10_1038-S44321-024-00142-x.

## Disclosure and competing interests statement

ST is the inventor of Plasmax™ cell culture medium. The remaining authors declare no competing interests.

# Expanded View Figures

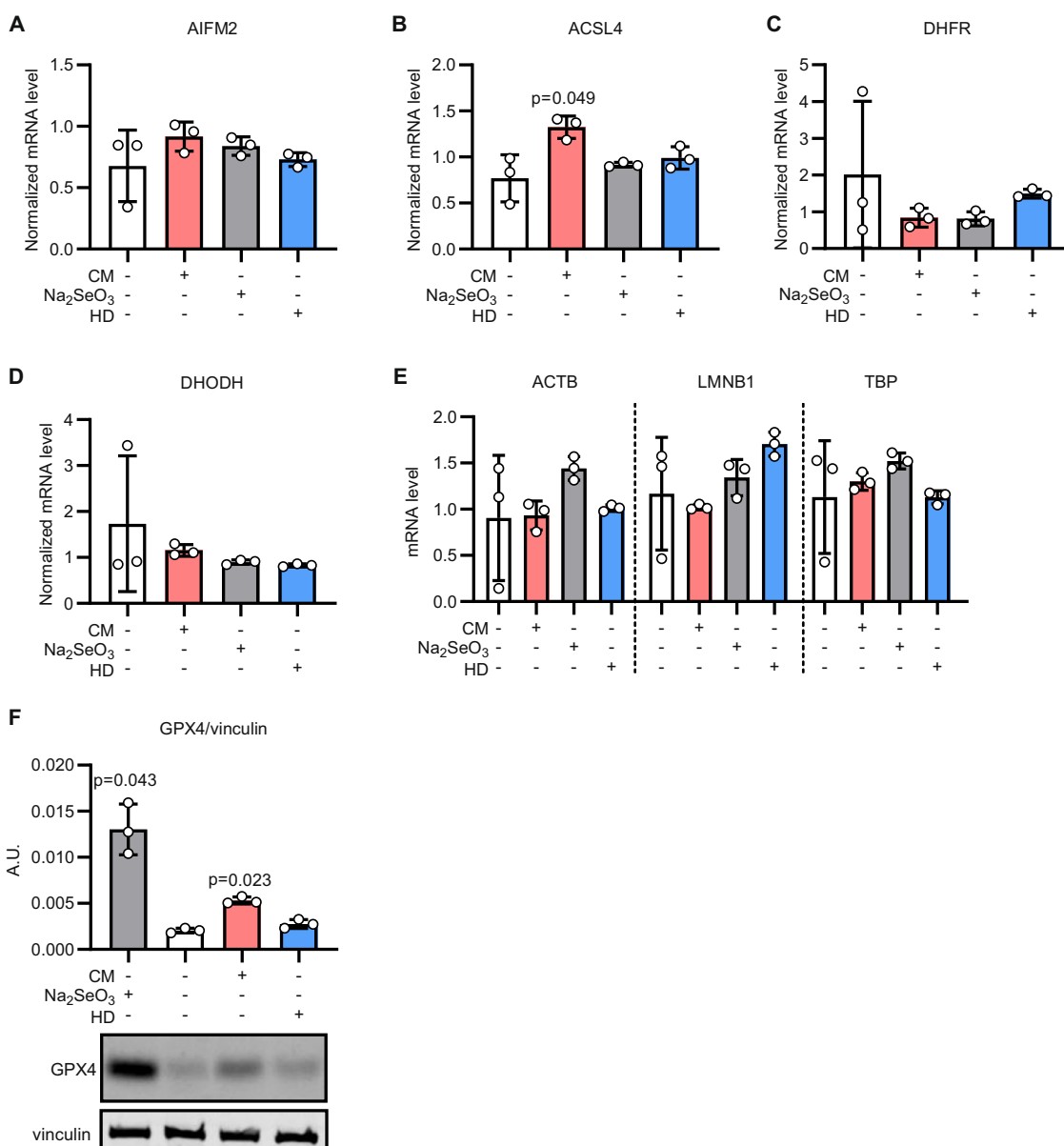

**Figure EV1. Breast cancer cells produce an anti-ferroptosis molecule at high density.**

(**A–D**) qPCR quantification of *AIFM2* (**A**), *ACSL4* (**B**), *DHFR* (**C**), and *DHODH* (**D**) mRNA expression in MDA-MB-468 cells seeded at low density in mock medium and supplemented with 50 nM selenite or conditioned medium (CM) for 2 days as indicated. The gene expression was also assessed in cells seeded at high density (HD) grown for 2 days in mock medium. The mRNA level is normalised to the mean mRNA abundance of the three housekeeping genes (*ACTB*, *LMNB1*, and *TBP*) shown in (**E**). *P* values refer to a one-way ANOVA for paired samples with Dunnett's multiple comparisons test. $n_{exp} = 3$. Bars represent mean ± s.d. (**E**) qPCR quantification of *ACTB*, *LMNB1*, and *TBP* mRNA expression in MDA-MB-468 cells seeded and treated as described in A–D. $n_{exp} = 3$. Bars represent mean ± s.d. (**F**) Immunoblot analysis of GPX4 and vinculin (loading control) in MDA-MB-468 cells MDA-MB-468 cells seeded and treated as described in (**A–D**). A.U.: arbitrary unit. Representative images of GPX4 and vinculin (loading control) from one of the three experiments quantified in the upper graph. *P* values refer to a one-way ANOVA for paired samples with Dunnett's multiple comparisons test. Bars represent mean ± s.d.

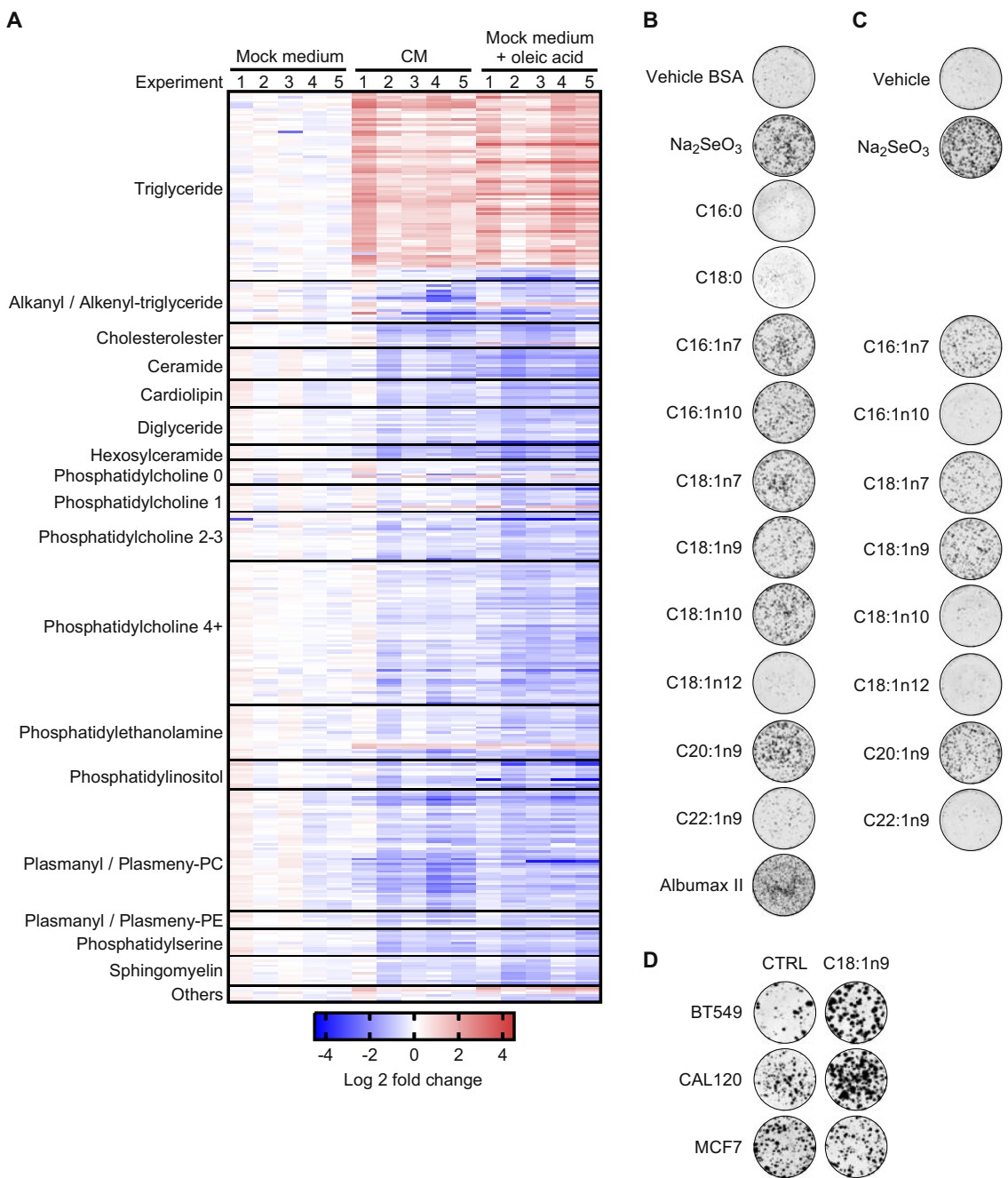

**Figure EV2. Monounsaturated fatty acids are enriched in the conditioned medium and prevent ferroptosis.**

(A) Heatmap of lipids regulated in MDA-MB-468 cells cultured at low density with mock medium, conditioned medium (CM) or mock medium with 10 µM oleic acid. Ferrostatin-1 was supplemented at 2 µM in all conditions. The lipids identified as significantly regulated with a two-tailed, homoscedastic Student's $t$ tests for unpaired samples in the comparison between conditioned medium and mock medium are reported and selected classes of lipids are indicated. For the phosphatidylcholine class the number of double bonds is also reported (0, 1, 2–3, 4 + ). The Log2 fold change refers to the comparison with mock medium supplemented cells. $n_{exp} = 5$. (B) Representative images for the colony-forming assays displayed in Fig. 3D. (C) Representative images for the colony-forming assays displayed in Fig. 3E. (D) Representative images for colony-forming assays displayed in Fig. 3F.

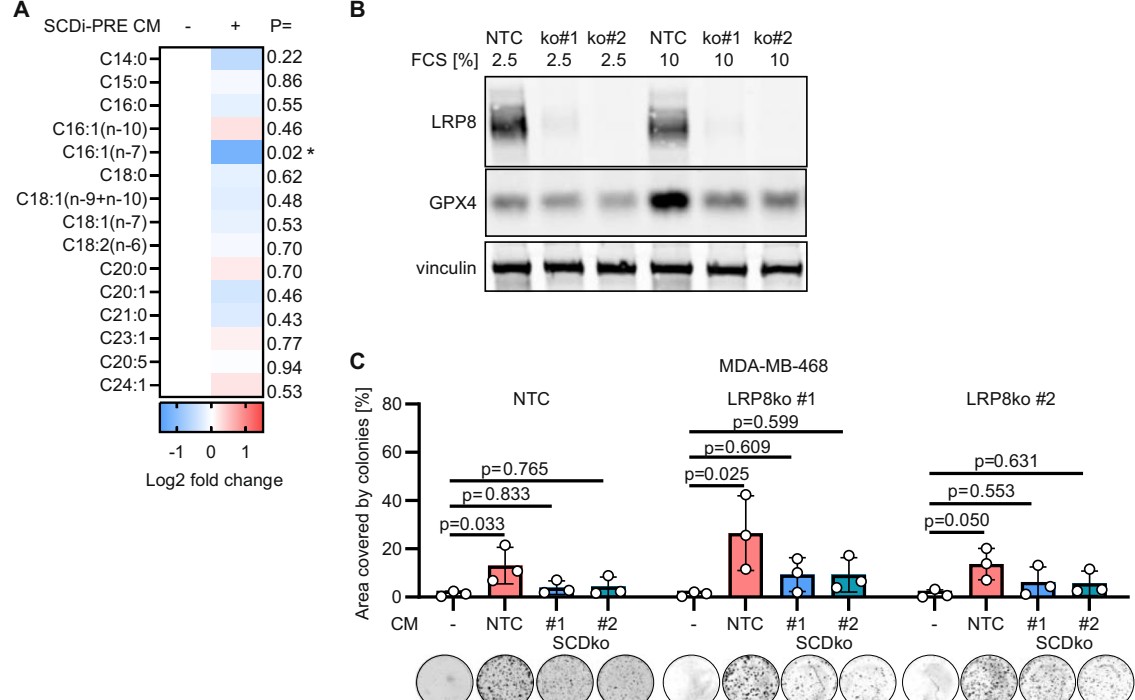

**Figure EV3.  SCD is required for the anti-ferroptotic capacities of the conditioned medium.**

(A) Quantification of total (free and lipid-bound) fatty acid species in medium conditioned by MDA-MB-468 cells without or with SCD inhibitor pre-treatment (SCDi PRE). Peak area values normalised on the signal from internal standard (C17:0) were used to calculate the Log2 fold change. *P* value refers to a two-tailed, homoscedastic Student's *t* tests for unpaired samples. These data complement Fig. 4D. $n_{exp}$ = 4. (B) Immunoblot of LRP8, GPX4, and vinculin (loading control) in NTC and LRP8ko clones (#1–2) derived from MDA-MB-468 breast cancer cells. (C) Well area covered by colonies formed by MDA-MB-468 NTC control cells and LRP8ko clones incubated for 7 days with mock medium or medium conditioned by NTC or SCDko MDA-MB-468 clones cultured as shown in Fig. 4F. *P* values refer to a one-way ANOVA test for unpaired samples with Dunnett's multiple comparisons test. $n_{exp}$ = 4. Bars represent mean ± s.d. Representative images of wells with colonies are shown for each experimental condition.

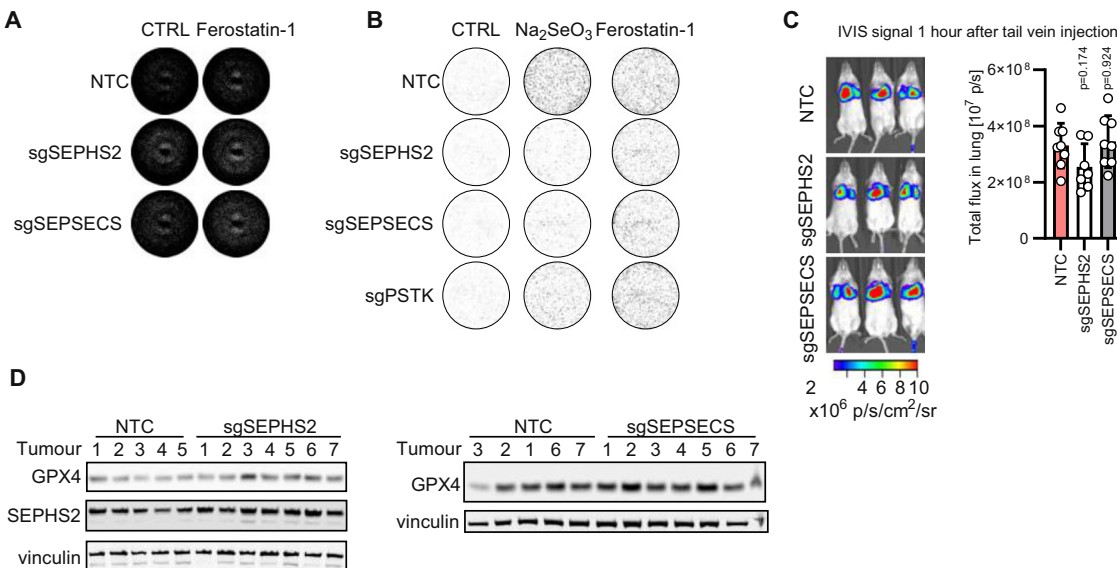

**Figure EV4.** **Targeting selenocysteine biosynthesis impairs lung metastasis of TNBC.**

(A) Representative images of the well area covered by cells at the end of the assays shown in Fig. 6C. (B) Representative images of the colony-forming assays shown in Fig. 6D. (C) IVIS pictures and quantification of lung metastasis burden 1 h after tail vein injection of $2 \times 10^6$ NTC, sgSEPHS2 or sgSEPSECS MDA-MB468 cells. The same mice are shown in Fig. 6H,I. *P* value refers to a one-way ANOVA test for unpaired samples with Dunnett's multiple comparisons test comparing to the NTC control. $n = 7$–8 female NSG mice as indicated by data points. One injected mouse of sgSEPHS2 group had to be culled due to husbandry reasons. (D) Immunoblot for GPX4, SEPHS2 and vinculin (loading control) in mammary tumours sampled 38 days after the transplantation of NTC, sgSEPHS2, or sgSEPSECS MDA-MB-468 cells. For each experimental group the lysates from 7 tumours were loaded as indicated.

