## [Peer Review File · EMBO Molecular Medicine]

Breast cancer secretes anti-ferroptotic MUFAs and depends on selenoprotein synthesis for metastasis

Tobias Ackermann, Engy Shokry, Ruhi Deshmukh, Jayanthi Anand, Laura Galbraith, Louise Mitchell, Giovanny Rodriguez-Blanco, Victor Villar, Britt Sterken, Colin Nixon, Sara Zanivan, Karen Blyth, David Sumpton, and Saverio Tardito

Corresponding author: Saverio Tardito (s.tardito@beatson.gla.ac.uk)

Review Timeline:

Submission Date:	23rd Jun 23
Editorial Decision:	13th Jul 23
Revision Received:	5th Apr 24
Editorial Decision:	8th May 24
Revision Received:	31st Jul 24
Editorial Decision:	8th Aug 24
Revision Received:	23rd Aug 24
Accepted:	4th Sep 24

Editor: Lise Roth

Transaction Report:

13th Jul 2023

Dear Dr. Tardito,

Thank you for the submission of your manuscript to EMBO Molecular Medicine. We have now received feedback from the three referees who agreed to evaluate your work.

As you will see from the enclosed reports, the referees acknowledge the potential interest of the findings, however, they also raise several major concerns on the study, and do not feel that the conclusions are sufficiently supported by the data at this point.

Given the nature of the referees' concerns and the amount of time and work that would be required to address them, and considering that at EMBO Press we encourage one round of revisions only in a reasonable time frame, I am afraid I see little choice but to return the manuscript to you at this point with the decision that we cannot offer to publish it.

Given the potential interest of the findings, we would, however, have no objection to consider a new manuscript on the same topic if at some time in the near future you obtained data that would considerably strengthen the message of the study and address the referees concerns in full. To be completely clear, however, I would like to stress that if you were to send a new manuscript this would be treated as a new submission rather than a revision and would be reviewed afresh, in particular with respect to the literature and the novelty of your findings at the time of resubmission. If you decide to follow this route, please make sure you nevertheless upload a letter of response to the referees' comments.

At this stage, though, I am sorry to have to disappoint you. I nevertheless hope that the referee comments will be helpful in your continued work in this area, and I thank you for considering EMBO Molecular Medicine.

Yours sincerely,

Lise Roth

***** Reviewer's comments *****

Referee #1 (Comments on Novelty/Model System for Author):

The study is timely and it adds to the importance and growing recognition of the importance of selenocysteine metabolism in multiple steps relevant to cancer pathology (metastasis in this case). The overall medical relevance is currently not highest due to the lack of pharmacological agents that could be used to test this, nevertheless the growing interest in these field should be able to provide these tools to be put to test in preclinical models.

Referee #1 (Remarks for Author):

The work by Ackermann et al. is motivated by the limited availability of therapeutic options for patients with triple-negative breast cancer (TNBC) and the high rate of metastatic recurrence, leading to a poor prognosis. The study reveals that TNBC cells secrete an anti-ferroptotic factor into the extracellular environment when cultured at high cell densities, but they become susceptible to ferroptosis when forming colonies from single cells. The identified anti-ferroptotic factor is monounsaturated fatty acid (MUFA)-containing lipids, and the vulnerability of single cells to ferroptosis depends on the expression of stearoyl-CoA desaturase (SCD), which is proportional to cell density.

1. To better characterize their model, I would encourage the authors to investigate the expression of critical ferroptosis regulators such as GPX4, AIFM2, and ACSL4 in cells exposed to conditioned and non-conditioned media. This analysis would provide valuable insights into the interplay between MUFA-containing lipids and the regulation of ferroptosis.
2. Based on our previous experience, it is worth noting that treatments aimed at removing lipid components from fetal bovine serum (FBS) can result in a significant loss of total selenium, likely in the form of SELENOP. Therefore, while the SCD knockout (KO) experiments are well-founded, the reported "stress" condition of this model could also impact the cell's secretome. To address this concern, a clear experiment to validate or exclude this impact would be to determine if conditioned media can rescue LRP8-deficient cells.

3. To provide additional support for the authors' hypothesis, incorporating ACSL3-deficient cells into the study would strengthen the notion that the incorporation of MUFAs plays a central protective role in their model.

4. The fractionation experiment suggests that lipids are transferred as a lipoprotein complex, but this aspect is not discussed in detail. It would be relevant to address this observation, as it potentially opens up avenues for characterizing a relevant pathway that could be explored in subsequent studies.

5. It is important for the authors to comment on the fact that the loss of any gene required for selenoprotein biosynthesis is generally expected to be lethal, as supported by data from the dependency map. This raises questions regarding the reported knockout (KO) models. It remains unclear how GPX4 is produced in these cells if they cannot produce tRNA-Sec. Providing additional clarification on this point would enhance the understanding of the study.

Minor aspects:

The term "selenocysteinilation" might generate confusion. It refers to the direct incorporation of selenocysteine into tRNA by an aminoacyl-tRNA synthetase. Since there are no specific enzymes dedicated to selenocysteine incorporation, and selenium is only incorporated into a preformed phospho-Ser-tRNA, it may be justifiable to use the term "tRNA^{Sec} biosynthesis" instead.

In line 13, "selenoproteins" should be replaced with "GPX4" since it is the only selenoprotein involved in ferroptosis (PMID: 29290465).

In line 26, it should be clarified that lipophilic antioxidants do not directly reduce peroxides; instead, they reduce the free radical species generated from the hydroperoxide group. However, GPX4, acting through a distinct mechanism, can reduce hydroperoxides (PMID: 3978121).

Line 35 seems to be incomplete. Please provide the necessary details to complete the sentence.

In line 237, please specify "Lipid peroxidation" to convey the intended meaning.

In line 401, a reference is needed to support the statement made.

Referee #2 (Comments on Novelty/Model System for Author):

The used animal model does not support the authors' conclusion. Interventions modulating circulating MUFA levels and targeting SCD would be suitable for the animal tumor experiments in this study.

Referee #2 (Remarks for Author):

The manuscript investigates the mechanism underlying the dependency of ferroptosis on the low culture density of a breast cancer cell line. The study reveals that secreted MUFA functions as an anti-ferroptotic molecule by analyzing the components in the conditioning medium. Additionally, the authors demonstrate the influence of modulating the selenocysteine incorporation pathway on tumor growth and metastasis in an animal xenograft tumor model. The main finding of the study, that secreted MUFA from the cells acts against ferroptosis, is intriguing. However, there are several remaining issues, particularly the lack of animal studies supporting the findings in an in vivo context.

1. Figure 6 appears to address a different topic from Figures 1-5. The MUFA parts (Figures 1-5) and the selenocysteine parts (Fig 6) seem to tell different stories. Therefore, the animal data in Figure 6 does not add further evidence to support the conclusion presented in Figures 1-5. Targeting the selenocysteine pathway, such as the SEPHS2 gene, for the xenograft animal model has already been reported (PMID Nat Metab. 2020 Jul;2(7):603-611). Thus, the reviewer suggests focusing on the MUFA aspect and expanding it for a more convincing argument, (while separating the animal data in Figure 6 for another paper). Interventions modulating circulating MUFA levels and targeting SCD would be suitable for the animal tumor experiments in this study.

2. To examine the anti-ferroptotic mechanism of the conditioning medium and MUFA, it would be necessary to conduct lipidomics analysis on cells treated with and without the conditioning medium or MUFA.

3. In Figure 4, lipidomics analysis of the conditioning medium samples from cells treated with SCD1 inhibitor and/or SCD KO is required. In Figure 4D, only a single lipid species was measured.

4. The anti-ferroptotic effect of selenium supplementation is likely due to the induction of GPX4. Please check the protein expression levels of GPX4 in cells treated with selenium and also in those treated with the conditioning medium. For this

purpose, to examine GPX4 expression, samples of cells treated with a ferroptosis inhibitor would also be required to exclude any sub-effects caused by cell death. Also, the effect of SCDi and SCD KO on GPX4 expression is required because Depmap data base (<https://depmap.org/portal/>) showed SCD1 is the top ranked gene in the negative association with GPX4 expression. Taken together, the present study needs to show the effect of condition medium/SCD inhibitor/SCD KO on ferroptosis sensitivity is whether due to the change of cellular MUFA contents or other mechanism (such as GPX4 expression).

5. The conditioning medium should also contain some selenium source. Was the level not changed by the treatment with SCDi or SCD KO?

6. Is the concentration of C18:1 MUFA comparable to the concentration of the single MUFA supplementation required for preventing ferroptosis (Fig3D)?

7. In Figure 1A, it would be necessary to include a ferroptosis inhibitor-treated group to demonstrate that ferroptosis is indeed occurring in the used setting.

8. Direct evidence of cell death should also be provided to demonstrate ferroptosis in some key experiments. The cell number and colony number alone are not sufficient to establish this.

Minor:

9. Line 489: "the levels of lipid peroxides were not significantly affected by SCD deletion" - The evaluated parameter is the amount of oxidized Bodipy and not lipid peroxide per se.

Referee #3 (Comments on Novelty/Model System for Author):

See details in the remarks to the author regarding the culture media used, cell density etc. Mouse experiments do not seem to have been repeated enough times for repeatability.

Referee #3 (Remarks for Author):

Overall, while this is an interesting area of study, this paper lacks sufficient evidence and a coherent story to add substantially to the current literature. It appears like it is two disconnected stories around MUFA/SCD in TNBC, and then a second story on sodium selenite. Data should be better compared with the current literature. For example, Xie et al JHO 2022 examined the role of adipocytes in protecting TNBC cells from ferroptosis, highlighting a role of oleic acid in that process - 3 papers are cited in other cancers but not this key paper in TNBC. A recent article from Yang et al Cell Metab, 35(1), 84-100.e108 looked at ferroptosis in TNBC.

Considering the key role of xCT in TNBC (Timmerman LA et al Cancer Cell 2013) and CB1 in TNBC (Li, P et al Cell Death Disease 2022), additional mechanistic insights into this process by inducing ferroptosis through other pathways would be insightful. In general, while ferroptosis is discussed throughout, no direct evidence is provided for significant changes in ferroptosis.

The authors cite their seminal paper on Plasmax in the second sentence of the results, highlighting how sodium selenite is key to ferroptosis. Why are all these experiments not done in this more physiological media? Justification of this approach and the relevance to in vivo physiology should be provided from the outset. Considering the oversupply of carbon sources such as glutamine and glucose in the traditional culture media used, this may induce artificial production and secretion of the MUFAs. Particularly for TNBC cells which so heavily utilise glutamine and glucose for their unique metabolism - which may explain why other BCa subtype cell lines do not produce these MUFAs. Testing whether these MUFAs are also produced in cells cultured in physiological nutrients of Plasmax is essential to ensure the physiological conditions are replicated. Since sodium selenite is important, Plasmax without sodium selenite could be used as a control.

There is no statement on cell line verification, or where these cells were sourced from and when.

Cells needed to be cultured at high density for the conditioning to work. Firstly, images should be taken of the cells to get a better understanding of what a high-density culture looks like, and how long they were maintained in this high-density state. A time course experiment would be useful to understand whether this is a physiological process that would occur in vivo - i.e. coupling analysis of cell growth kinetics over time, images of the cells, along with the amount of secreted components in the media for the 2 day conditioning period.

Figure 1 should also include data on colony number, not just colony area. Figure 1A the colony area does not match the images shown underneath - and the tight error bars do not suggest these data are from the same experiment. This is concerning for the first figure of the paper. In addition, how have statistics been performed for E0771 when it is only n=2? More details need to be provided to determine if these experiments are technical replicates or individual experiments.

The heatmap in Figure 3A does not provide any useful information, as all data are relative within the rows and you would expect

large differences between different sized fractions. These data should be presented in a more meaningful way, that can give an indication of how much less is present between the two fractions. As it is presented a 90% reduction could look the same as a 5% reduction.

Representative colony images should be provided for Figure 4C and G (indeed for all Figures where area is provided). It is notable that the maximal area is much lower than seen in Figure 1, even for the NTC control. Is there a reason?

Data in Figure 4F are suggested to show impaired cell proliferation in both clones, however the starting cell concentration is different. This should be undertaken with the same starting cell number. Again, images of the cell confluency should be shown to enable interpretation of the differences between the NTC and SCDko cell conditioned media on colony formation (4G).

Figure 5B should be repeated with the same assay as Figure 5A. It is unclear why this was not done, particularly as the text relates the data in 5B directly to 5A on page 14 ("in these conditions"). Similarly, the RSL3 data should also have been performed with sodium selenite "rescue" as this would assist in showing whether these protective effects are working in the same pathway or by different mechanisms.

As noted above, there are substantial difference in the area covered by the colonies in these different experiments. Some experiments are at a maximum area of 8%, others 80%. Either cell numbers were different (low vs high density), or perhaps it is the inclusion of sodium selenite - this needs to be clearly stated in the text and legends to ensure clarity for each figure panel.

Additional housekeeping genes should be used in the SCD expression analysis as confluence could affect levels of beta actin. In addition, protein expression should be shown with equal protein loading for all cell lines, not just 468. It is noted that Figure 4E western has additional bands above SCD, but these are absent in Figure 5H - are these the same antibody?

The authors should be more accurate with their wording around "loss of SCD" and "downregulation" of genes at low cell density. Their data show differences in expression, but which is "normal" and which is up or down remains to be determined. Critically, how these data have any bearing on tumours in vivo - where 2D cell density is not a factor - have not been determined in this study (see comment below about serum analysis).

The focus on SCD as "critical" is somewhat diminished by the findings around lipid peroxides in Figure 5F. For a paper on ferroptosis, further assessment of lipid peroxidation would be expected to ensure ferroptosis is indeed responsible for these phenotypes.

Figure 6C should be repeated with the other two KO lines from Figure 6B.

Tumour experiments in Figure 6 are essentially a single experiment with just 3 mice per arm. There is always variation in tumour take rate in individual animals, with these data likely to be underpowered to make the current conclusions. Particularly where the two groups are subject to individual cell counts, and a small variation in the number of injected cells could lead to changes in tumour growth. In addition, one of the control mice was subject to a "technical issue" which meant only a single tumour was implanted, which may affect the tumour size if the two tumours secrete factors that enhance/support their growth. Out of interest, it should be stated whether the "two tumours derived from a single injection" appeared in the mouse with only 1 injection, as perhaps this was actually 2 injections in the same fat pad, which would totally change the result. Additional mice in a separate experiment would assist in ensuring these data are more robust and scientifically valid. This needs to be performed since there are no significant differences in tumour size for the ex vivo analysis.

No details are provided as to why the experiment stopped at 35 days - was this a predetermined endpoint or an ethical endpoint? These are quite small tumours by normal xenograft ethical standards, which also makes these data hard to interpret. Figure 6I should include protein expression rather than just mRNA expression for the knockout to ensure the protein levels are indeed lower in the knockout post tumour growth.

In the discussion it is stated: "Our results complement these findings and suggest that primary tumours, where cells are found at very high density, could secrete MUFA-containing lipids and alter the lipid composition in circulation thereby protecting metastasizing cells from ferroptosis." This should have been looked at by analysis of the mice with tumours compared to control serum for MUFA-containing lipids (and indeed compared to SCDko cells).

As a service to authors, EMBO provides authors with the possibility to transfer a manuscript that one journal cannot offer to publish to another EMBO publication. The full manuscript and if applicable, reviewers reports are automatically sent to the receiving journal to allow for fast handling and a prompt decision on your manuscript. For more details of this service, and to transfer your manuscript to another EMBO title please click on Link Not Available

Revisions for #EMM-2023-18213 '*Breast cancer prevents ferroptosis by secreting MUFA and depends on tRNAsec synthesis for metastasis.*'

- Referees' comment to Authors.
- Authors' response.

Referee #1 (Comments on Novelty/Model System for Author):

The study is timely and it adds to the importance and growing recognition of the importance of selenocysteine metabolism in multiple steps relevant to cancer pathology (metastasis in this case). The overall medical relevance is currently not highest due to the lack of pharmacological agents that could be used to test this, nevertheless the growing interest in these field should be able to provide these tools to be put to test in preclinical models.

Referee #1 (Remarks for Author):

The work by Ackermann et al. is motivated by the limited availability of therapeutic options for patients with triple-negative breast cancer (TNBC) and the high rate of metastatic recurrence, leading to a poor prognosis. The study reveals that TNBC cells secrete an anti-ferroptotic factor into the extracellular environment when cultured at high cell densities, but they become susceptible to ferroptosis when forming colonies from single cells. The identified anti-ferroptotic factor is monounsaturated fatty acid (MUFA)-containing lipids, and the vulnerability of single cells to ferroptosis depends on the expression of stearyl-CoA desaturase (SCD), which is proportional to cell density.

1. To better characterize their model, I would encourage the authors to investigate the expression of critical ferroptosis regulators such as GPX4, AIFM2, and ACSL4 in cells exposed to conditioned and non-conditioned media. This analysis would provide valuable insights into the interplay between MUFA-containing lipids and the regulation of ferroptosis.

We would like to thank Referee#1 for the appreciation of our work and for the suggestion to look at the expression of critical ferroptosis regulators in cells exposed to conditioned and non-conditioned media. To address this point, we measured the mRNA expression of AIFM2 and ACSL4 in these conditions, as well as that of DHFR and DHODH two additional regulators of ferroptosis (Ref 10 and 09 in the revised manuscript). The expression of GPX4 was assessed at the protein level. These results are reported in the revised Figure S2 and overall show that the anti-ferroptotic effects of conditioned medium are not mediated by the expression of these genes in the recipient cells. These results strengthen the working model that we propose where the MUFA-containing lipids secreted in the conditions medium are directly responsible for the suppression of ferroptosis in cells at low density.

2. Based on our previous experience, it is worth noting that treatments aimed at removing lipid components from fetal bovine serum (FBS) can result in a significant loss of total selenium, likely in the form of SELENOP. Therefore, while the SCD knockout (KO) experiments are well-founded, the reported

"stress" condition of this model could also impact the cell's secretome. To address this concern, a clear experiment to validate or exclude this impact would be to determine if conditioned media can rescue LRP8-deficient cells.

We thank the Referee for sharing the unpublished insight. We fully agree with their view, and we also experienced in an ongoing collaborative project that the procedure to strip lipids from FBS has a profound effect on selenoP availability. However, we would like to clarify that 1) the serum supplemented in all the experiments presented in this study is not treated to remove lipids or any other component, and 2) in all the experiments the medium used to culture the cells that condition the medium is serum-free. The serum-free medium is then supplemented after the conditioning with 2.5% FBS just before being used for seeding the colony forming assays. To reassure Referee #1 we will clarify these important methodological points in the revised manuscript.

As elegantly showed by Friedmann Angeli's group in EMBO Mol Med PMID: 3743585 LRP8-dependent uptake of SELENOP has a pivotal role in modulating ferroptosis in cellular and mouse models. To test whether the secretion of SELENOP is relevant to the anti-ferroptotic properties of the medium conditioned by wild type and SCD-KO cells, we followed the Referee #1 suggestion and generated LRP8-deficient breast cancer cells. The results shown in Figure S4B show that LRP8 wild type MDA-MB-468 cells respond to increasing concentrations of FBS by increasing the abundance of GPX4 protein while LRP8 KO clones don't, demonstrating the relevance of LRP8 for selenoproteins production in this model. As requested, we tested the effect of the medium conditioned by control and SCDko cells MDA-MB-468 cells on the clonogenicity of LRP8ko cells. The results presented In Figure S4C show that LRP8ko cells respond to the medium conditioned by wild type cells increasing the number of colonies formed. Moreover, this pro-survival effect of the condition medium in the LRP8 ko cells was blunted if the medium employed for the colony forming assay was conditioned by SCDko cells. Overall, these results show that LRP8 does contribute to the selenoproteins (i.e. GPX4) production in breast cancer cells, but they clearly demonstrate that the anti-ferroptotic factors released in the condition medium by breast cancer cells are not acting through the modulation of selenoP uptake.

3. To provide additional support for the authors' hypothesis, incorporating ACSL3-deficient cells into the study would strengthen the notion that the incorporation of MUFAs plays a central protective role in their model.

We would like to thank Referee#1 for the very relevant suggestion that we followed by generating ACSL3-deficient MDA-MB-468 cells on which we tested the effects of medium conditioned by MDA-MB-468 wild type cells. The results shown in Figure 2F-G demonstrate that cells deficient for ACSL3 are not receptive for the anti-ferroptotic factors present in the conditioned medium, suggesting that these factors are ACSL3 lipidic substrates. These new results complete and strengthen the proposed mechanistic model.

4. The fractionation experiment suggests that lipids are transferred as a lipoprotein complex, but this aspect is not discussed in detail. It would be relevant to address this observation, as it potentially opens up avenues for characterizing a relevant pathway that could be explored in subsequent studies.

We would like to thank Referee#1 for the suggestion to discuss the relevant hypothesis that lipids could be transferred via lipoproteins in the conditioned medium. Our results show that the conditioned medium fully retains its anti-ferroptotic activity after being exposed to rather extreme denaturing condition (i.e. 15 minutes at 95°C as shown in Figure 2B, D).

In addition, the extracts of the conditioned medium obtained with the Bligh lipids extraction method that removes proteins, retain the ferroptotic activity of the conditioned medium.

Overall, these data convinced us that the protein binding the anti-ferroptotic secreted lipid is completely dispensable for the anti-ferroptotic effects of the conditioned medium.

We clarified this important aspect in the revised results section describing Figure 2E: *Altogether these results suggest that the active anti-ferroptotic factor produced by TNBC cells at high density is bound to a dispensable carrier > 10 kDa and is a heat-resistant lipid sufficient to confer anti-ferroptotic effects to the conditioned medium.*

5. It is important for the authors to comment on the fact that the loss of any gene required for selenoprotein biosynthesis is generally expected to be lethal, as supported by data from the dependency map. This raises questions regarding the reported knockout (KO) models. It remains unclear how GPX4 is produced in these cells if they cannot produce tRNA-Sec. Providing additional clarification on this point would enhance the understanding of the study.

We would like to thank Referee#1 for this conceptual point very important for the translational implications of our study. We fully agree with them that based on current knowledge, GPX4 can only be produced *via* selenocysteine synthesis. With regards to the residual presence of GPX4 in cells interfered for tRNA-Sec production, Figure 6B,E we would like to clarify that due to the colony formation inefficiency of the cells interfered for tRNA-Sec production, we used pools (rather than clones) of cells targeted by sgRNA against SEPHS2, PSTK or SEPSECS. Therefore, it is conceivable that even after undergoing puromycin selection, a small fraction of these cells retained expression of the targeted genes (see Figure 6E for SEPHS2) explaining the residual levels of GPX4.

We are aware of the literature showing that murine models deficient (KO) for selenoprotein biosynthesis have severe phenotypes or succumb to perinatal death. However, our new *in vivo* results included in the revised manuscript convincingly demonstrate that the interference with the expression of SEPHS2 or SEPSECS does not delay tumour growth, but severely impairs metastatic colonization (Figure 6F-K). These results suggest that the loss of genes required for selenoprotein biosynthesis is conditionally lethal for breast cancer cells in blood circulation. To our knowledge the tolerability of a partial and transient inhibition of selenoprotein biosynthesis with pharmacologic agents in adult mice has not been tested, and our results provide the rationale to develop such agents that could have a therapeutic window against metastatic TNBC cells.

The new results addressing this point from Referee#1 have been added to the results section describing Figure 6.

Minor aspects:

The term "selenocysteinilation" might generate confusion. It refers to the direct incorporation of selenocysteine into tRNA by an aminoacyl-tRNA synthetase. Since there are no specific enzymes dedicated to selenocysteine incorporation, and selenium is only incorporated into a preformed phospho-Ser-tRNA, it may be justifiable to use the term "tRNA^{Sec} biosynthesis" instead.

In the revised manuscript we replaced the term 'selenocysteinilation' with 'tRNA^{Sec} biosynthesis' as suggested by Referee#1.

In line 13, "selenoproteins" should be replaced with "GPX4" since it is the only selenoprotein involved in ferroptosis (PMID: 29290465).

At line 17 of the revised manuscript we specified that the 'selenoproteins' is 'GPX4' as suggested by Referee#1.

In line 26, it should be clarified that lipophilic antioxidants do not directly reduce peroxides; instead, they reduce the free radical species generated from the hydroperoxide group. However, GPX4, acting through a distinct mechanism, can reduce hydroperoxides (PMID: 3978121).

We would like to thank Referee#1 for this knowledgeable clarification and suggestion that we incorporated in the revised manuscript as suggested (lines 30-34).

Line 35 seems to be incomplete. Please provide the necessary details to complete the sentence.

The sentence originally at line 35 (corresponding to line 43 of the revised manuscript) has been modified to improve the clarity and facilitate the reading : '*Selenocysteine is a genetically encoded amino acid, and a structural analogue of cysteine with a selenium atom instead of the sulphur one*'.

In line 237, please specify "Lipid peroxidation" to convey the intended meaning.

The heading of the methods section at line 237 (corresponding to line 291 in the revised manuscript) has been changed to '*C11 BODIPY lipid peroxidation assay*' as requested.

In line 401, a reference is needed to support the statement made.

We agree with Referee#1 that the sentence at line 401 was imprecise and in order to avoid confusion about the physiological concentrations of the four distinct fatty acids we modified the text at lines 495-497 as follows '*Of the 7 MUFAs with colony stimulating activity, 4 (C16:1n7, C18:1n7, C18:1n9 and C20:1n9) retained their activity at a lower concentration (10 μ M, Figure 3E and S3C)*'.

Referee #2 (Comments on Novelty/Model System for Author):

The used animal model does not support the authors' conclusion. Interventions modulating circulating MUFA levels and targeting SCD would be suitable for the animal tumor experiments in this study.

Referee #2 (Remarks for Author):

The manuscript investigates the mechanism underlying the dependency of ferroptosis on the low culture density of a breast cancer cell line. The study reveals that secreted MUFA functions as an anti-ferroptotic molecule by analyzing the components in the conditioning medium. Additionally, the authors demonstrate the influence of modulating the selenocysteine incorporation pathway on tumor growth and metastasis in an animal xenograft tumor model. The main finding of the study, that secreted MUFA from the cells acts against ferroptosis, is intriguing. However, there are several remaining issues, particularly the lack of animal studies supporting the findings in an in vivo context.

1. Figure 6 appears to address a different topic from Figures 1-5. The MUFA parts (Figures 1-5) and the selenocysteine parts (Fig 6) seem to tell different stories. Therefore, the animal data in Figure 6 does not add further evidence to support the conclusion presented in Figures 1-5. Targeting the selenocysteine pathway, such as the SEPHS2 gene, for the xenograft animal model has already been reported (PMID Nat Metab. 2020 Jul;2(7):603-611). Thus, the reviewer suggests focusing on the MUFA aspect and expanding it for a more convincing argument, (while separating the animal data in Figure 6 for another paper). Interventions modulating circulating MUFA levels and targeting SCD would be suitable for the animal tumor experiments in this study.

We would like to thank Referee#2 for the relevant comments. The new results included in the revised manuscript better link Figures 1-5 to Figure 6. In summary, the data and conclusions drawn from figures 1-5 generated the rationale to target selenocysteine synthesis in metastatic TNBC. Indeed, in Figure 1-5 we show that TNBC cells lose the protective effects of MUFA when cultured at low density, and mainly rely on GPX4 activity to suppress ferroptosis, hence the idea to target selenocysteine synthesis in cells that during the metastatic dissemination loose contact with the primary tumour and recapitulate the observations made in culture at low density. To make this logic more explicit we added a schematic Synopsis of our working model, reconnecting Figure 1-5 to the results presented in the new Figure 6. Here we show that selenocysteine biosynthesis is a targetable conditional vulnerability in preclinical models of metastatic breast cancer.

With regards to the results reported in Nat Metab. 2020 Jul;2(7):603-611 (Ref. #20 cited in the introduction and discussion of our manuscript) we would like to emphasize that our results demonstrate a mechanism distinct from that proposed in the abovementioned paper. In fact, Kim et.al stated in their paper that *“Collectively, these data demonstrate that SEPHS2 is required in cancer cells for a reason other than production of selenoproteins”*. Differently, we showed in Figure 6 that the interference with the selenocysteine synthesis targeting SEPHS2 and SEPSECS, phenocopies the effects of selenium starvation

and it does not have any toxic effects in cells grown in culture at high density (revised Figure 6C-D) and *in vivo* as orthotopic mammary tumours (revised Figure 6 F-G).

To test further whether the SEPHS2 deletion requires the accumulation of toxic selenium metabolites to exert its cytotoxic activity, as shown by Kim et al., we performed supplemental experiments presented in the Figure 1 for Referee#2 below. These new results show that an excess of selenium supplemented up to 750nM does not elicit differential toxic effects in WT and SEPHS2 KO cells. These results support our mechanistic model whereby SEPHS2 prevents ferroptosis in cancer cells primarily by supporting the production of selenoproteins.

We welcomed the suggestion of Referee#2 to focus on “MUFA aspects” and we targeted SCD *in vivo*. In particular, we compared the progression of orthotopic mammary tumours obtained from WT and SCD ko MDA-MB-468 cells (Figure 4H-K). The results from these experiments show that SCD deletion does not delay the growth of primary tumours, in line with our observation that SCD ko cells grow comparably to WT controls when cultured at high density with 10% FBS (Figure 5B).

In addition, we measured the levels of total fatty acids in the interstitial fluid obtains from SCD wt and ko tumours and found that oleic acid and palmitoleic acid, two main products of SCD, are less abundant in the fluid from SCDko tumours compared to SCDwt (Figure 4K). These results strengthen the relevance of our cell culture-based results presented in Figure 1-5 for tumour biology and show that lipids produced by SCD-expressing tumours are released in the extracellular tumour microenvironment.

Figure 1 for Referee#2. The area of the well covered by NTC and sgSEPHS2 MDA-MB-468 cells seeded at high density and incubated for 5 days with the indicated concentrations of sodium selenite. P values refer to a homoscedastic T-test test for unpaired samples. N=3 experiments.

2. To examine the anti-ferroptotic mechanism of the conditioning medium and MUFA, it would be necessary to conduct lipidomics analysis on cells treated with and without the conditioning medium or MUFA.

We would like to thank Referee#2 for proposing this insightful experiment. We followed Referee#2's suggestion and perform lipidomic analysis on cells cultured at low density and treated with/without the conditioned medium or oleic acid. The results show that 350 out of the 790 lipidic species detected and identified by the analysis were consistently and coherently regulated by the conditioned medium and the oleic acid compared to cells exposed to unconditioned (mock) medium. The pattern of these lipidic species is presented in the heatmap of Figure S3A that highlight triglycerides (TG) as the class of lipids consistently more abundant in cells treated with conditioned medium or oleic acid. Indeed 70 out of 81 lipids increased by condition medium and oleic acid were TGs. These results strengthen the notion that the lipidome

changes elicited by the condition medium and oleic acid largely overlap, suggesting a common mode of action for these anti-ferroptotic agents.

3. In Figure 4, lipidomics analysis of the conditioning medium samples from cells treated with SCD1 inhibitor and/or SCD KO is required. In Figure 4D, only a single lipid species was measured.

We agree with Referee#2 that a more comprehensive lipidomic analysis of the medium conditioned by cells with interfered SCD activity is informative. To identify the fatty acids regulated by SCD we performed lipid saponification on the conditioned media samples. This analytical method allowed us to measure by gas chromatography mass spectrometry the total pool of fatty acids (i.e. free and lipid-bound fatty acids). The detailed analytical procedure is reported in the methods section of the revised manuscript.

We addressed Referee#2 request by integrating the results originally reported in Figure 4D with the new results shown in Figure S4A. The reported heatmap show all the detected fatty acids that are significantly regulated in the conditioned medium (CM) vs. mock medium (a control medium exposed to the same procedure as the CM but not exposed to cells). Amongst these fatty acids only the level of palmitoleic acid (C16:1 (n-7)) was significantly decreased in the CM from cells incubated with SCD inhibitor (SCDi) compared to WT control cells. These results highlight the specificity of the effects of SCD inhibition on the pool of secreted lipids and reinforce our hypothesis that the anti-ferroptotic effects of conditioned medium are dependent on SCD-dependent MUFA-secretion.

To further address this point *in vivo* and test the relevance of the results obtained with conditioned medium to tumour biology we performed a lipidomic analysis on the interstitial fluid obtained from orthotopic mammary tumours that were wt or deficient for SCD. The results are included in the revised manuscript (Figure 4K) and described in our response to point 1 of this Referee.

4. The anti-ferroptotic effect of selenium supplementation is likely due to the induction of GPX4. Please check the protein expression levels of GPX4 in cells treated with selenium and also in those treated with the conditioning medium. For this purpose, to examine GPX4 expression, samples of cells treated with a ferroptosis inhibitor would also be required to exclude any sub-effects caused by cell death. Also, the effect of SCDi and SCD KO on GPX4 expression is required because Depmap data base (<https://depmap.org/portal/>) showed SCD1 is the top ranked gene in the negative association with GPX4 expression. Taken together, the present study needs to show the effect of condition medium/SCD inhibitor/SCD KO on ferroptosis sensitivity is whether due to the change of cellular MUFA contents or other mechanism (such as GPX4 expression).

These are all very appropriate suggestions from Referee#2 and we fully agree with their statement that the anti-ferroptotic effect of selenium supplementation is likely due to the induction of GPX4 expression. To directly test this hypothesis, we performed Western blot analysis of GPX4 protein in MDA-MB-468 cells cultured at low density with conditioned medium, selenium supplementation in the presence of a ferroptosis inhibitor (Ferrostatin-1), as requested by Referee#2. The new results presented in Figure S2F confirm the Referee#2 hypothesis that selenium supplementation (50nM Na₂SeO₃) induces GPX4 levels (~7 fold). The treatment with the conditioned medium also increases GPX4 level yet to a much lower extent (~2-fold). We also checked GPX4 protein level in SCDko and SCD-inhibited cells as well as in SCDko tumours compared to respective controls. The new results presented in Figure 2 for Referee#2 below show

that SCD genetic and pharmacologic interference does not regulate GPX4 protein levels in cells and tumours.

Altogether these new results demonstrate that GPX4 levels do not respond to SCD activity and do not refute the hypothesis that the regulation of GPX4 levels could, at least in part, contribute to the anti-ferroptotic action of the conditioned medium. However, new orthogonal results obtained deleting ACSL3 in MDA-MB-468 cells (Figure 2F-G) show that ACSL3-deficient cells are no longer protected from ferroptosis by the conditioned medium, confirming that the anti-ferroptotic factors are ACSL3 lipidic substrates. Overall, these new results strengthen the proposed mechanistic model whereby MUFAs have an anti-ferroptotic role independent of the selenium-GPX4 axis.

Figure 2 for Referee#2. Representative Western blot images and quantification of GPX4, SCD and vinculin protein expression upon 2 days *in vitro* SCD inhibition (A), in SCDwt and SCDko cell lines (B) and in SCDwt and SCDko tumours (C). All experiments were performed with MDA-MB-468 cells and datapoints indicate number of biological replicates (n=2-6). The following antibodies and dilutions were used: GPX4, 1:1000, Abcam, # ab125066; SCD, 1:1000, Abcam, # ab19862; vinculin, 1:2000, Merck, # SAB4200080.

5. The conditioning medium should also contain some selenium source. Was the level not changed by the treatment with SCDi or SCD KO?

This is an intriguing hypothesis formulated by Referee#2. First, we would like to clarify the procedure employed to condition the medium as reported in the method section of the manuscript and in the schematic of Figure 4A. For the conditioning of the medium, cells were seeded in 10% FBS, the next day they were washed with PBS, and serum-free DMEM/F-12 that does not contain any source of selenium, was added for the conditioning and incubated for 2 hours for the experiment with the SCD inhibitor, and for 48 hours for all the other experiments. The conditioned medium was then supplemented with 2.5% FBS and used to incubate recipient cells.

As described in the response to the point 4 from Referee#2 the conditioned medium significantly increases (2-fold) the levels of GPX4 in the recipient cells. While this result is compatible with a role of GPX4 in the

antiferroptotic effect of the conditioned medium, the results obtained with ACSL3-deficient cells confirm that the anti-ferroptotic factors are of lipidic nature and they are substrates of ACSL3 that activate fatty acids for lipid biosynthesis.

In addition, to address the point 2 of Referee#1 and test whether the presence of SELENOP (the main carrier of selenium in physiological settings) in the conditions medium is relevant to its anti-ferroptotic properties, we generated MDA-MB-468 breast cancer cells deficient for SELENOP receptor LRP8. The new results shown in Figure S4B show that LRP8ko cells respond to increasing concentrations of FBS by increasing the abundance of GPX4 protein while LRP8ko cells do not, demonstrating the importance of LRP8 for selenoproteins production in this model. We then tested the effect of the conditioned medium on the clonogenicity of LRP8ko cells. The new results presented in Figure S4C show that the conditioned medium increased the colony forming capacity comparably in LRP8wt and ko cells, demonstrating that the anti-ferroptotic effects of the condition medium are not mediated by the selenium carrier, SELENOP.

6. Is the concentration of C18:1 MUFA comparable to the concentration of the single MUFA supplementation required for preventing ferroptosis (Fig3D)?

To address this interesting question from Referee#2 we quantified the concentration of free C18:1 MUFA in the 'rescuing fraction' (>10kDa) and non-rescuing fraction (<10kDa) of the condition medium and reported it in the revised Figure 3C. The concentration in the 'rescuing fraction' is ~ 0.4 μ M and this fraction stimulates colony formation when diluted 40x (Figure 2C). Therefore, the concentration of C18:1 MUFA doesn't reach the concentration of oleic acids used in Figure 3D-E. These data together with the results reported in the revised Figure 3A indicate that the MUFAs responsible for the anti-ferroptotic activity of the conditioned medium are not only free C18:1 but also lipid-bound fatty acids.

7. In Figure 1A, it would be necessary to include a ferroptosis inhibitor-treated group to demonstrate that ferroptosis is indeed occurring in the used setting.

We would like to thank Referee#2 for suggesting this important control. We repeated the experiments reported in the original Figure 1A including a condition with the ferroptosis inhibitor, Ferrostatin-1. The new results with Ferrostatin-1 included in the revised Figure 1C and S1B, show that the number of colonies significantly increase when ferroptosis is inhibited demonstrating that ferroptosis indeed occur under low cell density selenium-restricted culture conditions. In addition, Ferrostatin-1 significantly rescued the number of colonies of ACSL3-, SCD-, SEPHS2-, SEPSECS-, and PSTK-deficient cells demonstrating that also in these settings ferroptosis impairs colony formation.

8. Direct evidence of cell death should also be provided to demonstrate ferroptosis in some key experiments. The cell number and colony number alone are not sufficient to establish this.

We agree with Referee#2 that the direct evidence of cell death would strengthen the importance of ferroptosis in determining the number of colonies formed upon the experimental conditions tested. To directly visualize cell death events occurring at early time points of the colony forming assay we monitored the number of dead cells at day 4 after cell seeding with an automated live/dead cell imaging system (Incucyte). The results directly show that MDA-MB-468 cells die when seeded at low cell density and that

cells are rescued by ferrostatin-1 and selenium supplementation. These new results are reported in the revised Figure 1A-B, and the experimental details reported in the revised methods.

Minor:

9. Line 489: "the levels of lipid peroxides were not significantly affected by SCD deletion" - The evaluated parameter is the amount of oxidized Bodipy and not lipid peroxide per se.

We would like to thank Referee#2 for clarifying this aspect. We have corrected the text (line 555-556 of the revised manuscript) to acknowledge the indirect nature of measuring lipid peroxides levels with a fluorescent probe: *'the levels of oxidized C11 BODIPY, a probe for lipid peroxidation, were not significantly affected by SCD deletion'*.

Referee #3 (Comments on Novelty/Model System for Author):

See details in the remarks to the author regarding the culture media used, cell density etc. Mouse experiments do not seem to have been repeated enough times for repeatability.

Referee #3 (Remarks for Author):

1. Overall, while this is an interesting area of study, this paper lacks sufficient evidence and a coherent story to add substantially to the current literature. It appears like it is two disconnected stories around MUFA/SCD in TNBC, and then a second story on sodium selenite. Data should be better compared with the current literature. For example, Xie et al JHO 2022 examined the role of adipocytes in protecting TNBC cells from ferroptosis, highlighting a role of oleic acid in that process - 3 papers are cited in other cancers but not this key paper in TNBC. A recent article from Yang et al Cell Metab, 35(1), 84-100.e108 looked at ferroptosis in TNBC.

We would like to thank Referee#3 for the precise suggestions to integrate our study with current literature on the topic. We have added the two relevant papers mention by Referee#3 to the revised manuscript (revised introduction ref #17, and discussion ref #42).

Regarding the link between MUFA/SCD and selenium metabolism we are confident that the new results included in the revised manuscript better connect the two sets of consequential observations. Indeed, the cell culture experiments with conditioned media and the density-dependent modulation of SCD activity constitute the mechanistic rationale to efficiently target selenocysteine synthesis in metastatic TNBC, as shown in the new Figure 6. The *in vitro* results show that TNBC cells lose the protective effects of MUFA when cultured at low density. They constitute a proof of principle for the reliance of metastatic cells on selenocysteine biosynthesis to suppress ferroptosis. However, we agree with Referee#3 that this reasoning, implicitly obvious for the Authors, it might not have been clear enough for the reader. To make this logic more explicit and clear we have rewritten the concluding paragraphs of the discussion, and we added a graphical Synopsis our working model, reconnecting the cell-based experiments to the *in vivo* results. Furthermore, we have bridged the gap between the cell culture-based observations and *in vivo* studies by investigating the effect of SCD KO *in vivo* with orthotopic models, as also suggested by Referee#3 at point 18.

2. Considering the key role of xCT in TNBC (Timmerman LA et al Cancer Cell 2013) and CB1 in TNBC (Li, P et al Cell Death Disease 2022), additional mechanistic insights into this process by inducing ferroptosis through other pathways would be insightful. In general, while ferroptosis is discussed throughout, no direct evidence is provided for significant changes in ferroptosis.

We agree with Referee #3 that additional insights into the mechanism of cells death triggered by low density and selenium starvation could strengthen our study. We are very confident that our key observations are dependent on ferroptosis since they are consistently rescued by selenite and Ferrostatin-1, two well-accepted ferroptosis inhibitors.

To address Referee #3 point regarding the induction of ferroptosis through other pathways, we have tested the effects of erastin, an xCT inhibitor, on the colony formation capacity of TNBC cells plated at low density. The results shown below in the Figure 1 for Referee#3 clearly show that erastin prevents colony formation

independently of the presence of Ferrostatin-1, indicating that in our experimental setting erastin does not kill TNBC by ferroptosis.

To strengthen the notion that the colony formation is a stringent readout of ferroptosis in our experimental setting we have repeat key experiments including a condition with the ferroptosis inhibitor Ferrostatin-1.

The new results are included in the revised Figure 1C. Furthermore, to visualize cells undergoing cell death when seeded at low density we employed a live cell imaging (Incucyte). The results included in the revised Figure 1A-B demonstrate that a large proportion of TNBC cells seeded at low density die within 4 days from seeding and that the cell death is fully rescued by selenium and ferrostatin-1, allowing us to conclude that the area covered by cells is indeed a readout of ferroptosis in this setting.

Figure 1 for Referee#3. Cells were seeded at 5000 cells / well in medium supplemented with 2.5% FBS and erastin (500nM), sodium selenite (50nM) and Ferrsotatin-1 (2μM) as indicated, and incubated for 7 days. The area covered by colonies was quantified as reported in the methods section of the manuscript. A representative image of the colonies is shown for each condition. Nex=2- as indicated by the data points. Bars represent mean ± s.d..

3. The authors cite their seminal paper on Plasmax in the second sentence of the results, highlighting how sodium selenite is key to ferroptosis. Why are all these experiments not done in this more physiological media? Justification of this approach and the relevance to in vivo physiology should be provided from the outset. Considering the oversupply of carbon sources such as glutamine and glucose in the traditional culture media used, this may induce artificial production and secretion of the MUFAs. Particularly for TNBC cells which so heavily utilise glutamine and glucose for their unique metabolism – which may explain why other Bca subtype cell lines do not produce these MUFAs. Testing whether these MUFAs are also produced in cells cultured in physiological nutrients of Plasmax is essential to ensure the physiological conditions are replicated. Since sodium selenite is important, Plasmax without sodium selenite could be used as a control.

Firstly, we would like to thank Referee#3 for the appreciation of our previous work on Plasmax. The reason why we did not use Plasmax throughout this study is simply because Plasmax standard formulation contains sodium selenite, and this study would have required the production of large volumes of customized Plasmax without this trace element. However, we will not disappoint Referee#3 on this specific request that we have addressed by producing conditioned medium starting from selenite-free Plasmax. The results shown in the revised Figure 2A show that the conditioned Plasmax and DMEM-F12 have comparable colony stimulating effects.

4. There is no statement on cell line verification, or where these cells were sourced from and when.

We would like to thank Referee#3 for noting this oversight. Cell lines were authenticated using genomic DNA extracted with Puregene Gentra Kit and multiplexed using the Promega Geneprint Kit and multiplexed with a STR-based method (Promega Geneprint System). Samples were run on an Applied Biosystems 3130xl DNA analyser and the results analysed using the Applied Biosystems Genemapper v4.1 software. Profiles were matching the references reported by ATCC (LGC standards), Cellosaurus and DSMZ databases. This important methodological detail has been included in the revised methods 'Cell culture' section.

5. Cells needed to be cultured at high density for the conditioning to work. Firstly, images should be taken of the cells to get a better understanding of what a high-density culture looks like, and how long they were maintained in this high-density state. A time course experiment would be useful to understand whether this is a physiological process that would occur in vivo – i.e. coupling analysis of cell growth kinetics over time, images of the cells, along with the amount of secreted components in the media for the 2 day conditioning period.

We would like to thank Referee#3 for these relevant suggestions. Firstly, we want to clarify that for all the experiments, except those with SCDi, the medium was conditioned for 2 days by cells at high density. For the experiments with SCDi presented in Figure 4B-D the medium was conditioned for 2 hours as shown in the experimental work-flow chart in Figure 4A.

To address Referee#3 request we have provided images of high-density cultures of MDA-MB-468 at beginning and at the end of the 2 days conditioning period in the revised Figure S1A.

In addition, we would like to highlight that the time course of MDA-MB-468 cell number is reported in Figure 4F. These results show that NTC cells double twice during the 2 days period used to condition the medium, demonstrating that proliferation rate is not limited by these high-density conditions. In addition, we measured the levels of glucose and glutamine in the medium at the end of the conditioning period. The levels of these most avidly consumed nutrients show that they are depleted but not exhausted by high density cell cultures (see below Figure 2 for Referee#3). Moreover, the levels of the lactate secreted in the conditioned medium did not reach unphysiologically high levels (~11mM) as shown below in the Figure 2 for Referee#3. Finally, to test the relevance of these cell culture observations to tumour biology we measured the levels of total MUFAs (lipid bound and free) in the interstitial fluid of orthotopic TNBC tumours. The results included in Figure 4H-I show that MUFAs are present in the interstitial fluid of wild type tumours. Moreover, the abundance of vaccenic acid and oleic acid is selectively diminished in interstitial fluid from SCDko tumours, indicating that the detected MUFAs are not only blood-born but at least in part contributed by the biosynthetic activity of the cancer cells in the tumour.

Figure 2 for Referee#3. Quantification of glucose, lactate and glutamine in the medium conditioned by NTC and SCDko MDA-MB-468 cells and used in the experiment show in Figure 4G. The levels of metabolites in the medium were quantified with a YSI 2900 Biochemistry Analyzer. N experiments =3. P values refer to a one-way ANOVA test for unpaired samples with Dunnett's multiple comparisons test. Bars represent mean \pm s.d..

6. Figure 1 should also include data on colony number, not just colony area. Figure 1A the colony area does not match the images shown underneath - and the tight error bars do not suggest these data are from the same experiment. This is concerning for the first figure of the paper. In addition, how have statistics been performed for E0771 when it is only $n=2$? More details need to be provided to determine if these experiments are technical replicates or individual experiments.

To address Referee#3 concerns we have provided more data relative to the original Figure 1A.

With regards to the possible mismatch between the images of colonies and the quantified colony areas reported in the original Figure 1A, we double checked the raw data as well as the calculations and we can reassure Referee#3 that the images of the colonies presented in the original Figure 1A were obtained from one of the experiments reported as data point in the same figure. The original Figure 1A and all the images obtained from the experiments showed in that figure are reported below in the Figure 3A, B for Referee#3.

We hope that the additional data provided will solve the concerns of Referee#3 on the Figure 1A presented in the original submission of the manuscript.

Moreover, to address Referee#3's point 2, the original Figure 1A has been revised to include the experimental condition with Ferrostatin-1. The results obtained with Ferrostatin-1 strengthen the causality between ferroptosis and colony-formation (revised Figure 1C). For full transparency we present to Referee#3 all the images of all the replicates from the five independent experiments used to obtain these new results (Figure 3C for Referee#3).

In addition to the colony area obtained from 5 independent experiments shown in the revised Figure 1C, we addressed the Reviewer#3's request to provide the number of colonies, now reported in Figure S1B. Overall, the results presented in the revised manuscript (Figure 1C and Figure S1B) support the original conclusion that ferroptosis is impairing the colony forming capacity of TNBC seeded at low density.

Regarding the significance of the results originally shown in Figure 1B for the E0711 cell line, we would like to clarify that the P value was calculated with a two-tailed, homoscedastic Student's t tests for unpaired samples as stated in the original figure legend. While using a Student's t tests for 2 independent experiment is not incorrect and produced a P value indicating significant differences, after discussing this point with the Editor we agreed to remove the P value from comparisons obtained from two independent experiments.

Finally, as stated in the figure legends, each data point represents an independent experiment. Each individual experimental value for colony forming assay was mean of 2 or 3 replicate wells. We have added this information to the 'Colony formation assays' section of the revised methods.

Figure 3 for Referee#3. **A**, Figure 1A as reported in the original submission. **B**, Images of the colony forming assays quantified and presented in the Figure 1A of the original submission. The representative wells showed in the original Figure 1A are circled in green. **C**, Images of the colony forming assays quantified and presented in the revised Figure 1A and Figure S1D.

7. The heatmap in Figure 3A does not provide any useful information, as all data are relative within the rows and you would expect large differences between different sized fractions. These data should be

presented in a more meaningful way, that can give an indication of how much less is present between the two fractions. As it is presented a 90% reduction could look the same as a 5% reduction.

We appreciate the suggestion on the heatmap presented in the original Figure 3A that was based on z score values. To address Referee#3 concerns we have replaced the heatmap with one where the Log2 fold change is used instead of the z score. The interpretation of the results is more intuitive in the revised Figure 3A and we would like to thank Referee#3 for their suggestion that improved the clarity of the figure.

8. Representative colony images should be provided for Figure 4C and G (indeed for all Figures where area is provided). It is notable that the maximal area is much lower than seen in Figure 1, even for the NTC control. Is there a reason?

We have addressed Referee#3's request by including representative images of colonies in Figures 4C and G, as well as in the other figures where colony areas are provided. For the Figures 3D ,E ,F , 5B, 6C and D where the space was limited, the colony images were included as supplementary Figures (Figure S3B, C, D, S5A, S6A and B respectively).

Regarding the differences noted by Referee#3 between the % area covered by colonies in the original Figure 4C and Figure 1A, one possible reason is that in Figure 4C the medium was conditioned for 2h (as shown by the experimental flow chart in Figure 4A) while the medium used for Figure 1A was conditioned for 2 days. This information is reported in the method section of the original manuscript, and it is indicated in the revised legend of Figure 4C.

9. Data in Figure 4F are suggested to show impaired cell proliferation in both clones, however the starting cell concentration is different. This should be undertaken with the same starting cell number. Again, images of the cell confluency should be shown to enable interpretation of the differences between the NTC and SCDko cell conditioned media on colony formation (4G).

We would like to clarify that in the assay shown in Figure 4F the different number of cells plated for each cell line was deliberately chosen to obtain a comparable number of cells over the duration of the conditioning period (2 days, grayed in the graph of Figure 4F). This allowed us to achieve a level of conditioning comparable between NTC and SCDko lines. For this purpose, we deliberately seeded more SCDko cells than NTC cells to compensate for the slower growth of SCDko clones. As a result of this adjustment in the number of seeded cells, the calculated Area under the curve for the three groups reported in Figure 4F is comparable (NTC: 2468675 (+/-356289), SCDko#1: 2100483 (+/-393600), SCDko#2: 2524125 (+/-297456)). Consistently, Figure 2 for Referee#3, shown above, demonstrate that the level of nutrients consumption (glucose and glutamine) and metabolite secretion (lactate) are not significantly different between SCDko and NTC cells under these conditions.

Finally, as requested by Referee #3 we have added representative colony images in Figure 4G.

10. Figure 5B should be repeated with the same assay as Figure 5A. It is unclear why this was not done, particularly as the text relates the data in 5B directly to 5A on page 14 ("in these conditions"). Similarly, the RSL3 data should also have been performed with sodium selenite "rescue" as this would assist in showing whether these protective effects are working in the same pathway or by different mechanisms.

We agree with Referee#3 that a consistent redout between Figure 5A and B would facilitate the reader. To address this point, we have repeated the experiments originally shown in Figure 5A and 5B and used cell number as a common readout. The experiments originally reported in Figure 5A and B are summarized in the revised Figure 5A.

With regards to the presence of selenite in combination with the GPX4 inhibitor (RSL3) we would like to clarify that the experiments reported in the original Figure 5C (corresponding to the revised Figure 5B) are performed in the presence of 10% FBS. Under these conditions the SELENOP contributed by serum is sufficient to boost GPX4 expression, as shown in Figure S4B. Therefore, we assessed the behaviour of SCDko cells in the presence of an alternative ferroptosis inhibitor, the iron chelator, deferoxamine. The new results included in the revised Figure 5A show that SCDko cells cultured at high density can be partially rescued by deferoxamine.

Altogether these new results strengthen the original conclusion that SCDko cells are sensitized to ferroptosis when cultured at high density.

11. As noted above, there are substantial differences in the area covered by the colonies in these different experiments. Some experiments are at a maximum area of 8%, others 80%. Either cell numbers were different (low vs high density), or perhaps it is the inclusion of sodium selenite - this needs to be clearly stated in the text and legends to ensure clarity for each figure panel.

Referee#3 is correct in attributing to the seeding density the differences in area covered by cells that ranges from ~8% (e.g. Figure 4C, low density) to ~80% (e.g. Figure 5B, high density). Indeed, the Y axes of Figure 4C is labelled 'Area covered by colonies' while the Y axis of Figure 5B is 'Area covered by cells'. Moreover, the legends of these figures specify the cell density employed for each relevant panel (i.e. legend of Figure 4C 'Well area covered by colonies' and legend of Figure 5C 'Well area covered by NTC and SCDko MDA-MB-468 cells seeded at high density'. The different cell density chosen in different assays is key to demonstrate that ferroptosis is cell density dependent in our cell model.

The revised figure legends consistently define the experiments performed with cells at high density adding clarity on this important methodological detail.

12. Additional housekeeping genes should be used in the SCD expression analysis as confluence could affect levels of beta actin. In addition, protein expression should be shown with equal protein loading for all cell lines, not just 468. It is noted that Figure 4E western has additional bands above SCD, but these are absent in Figure 5H - are these the same antibody?

We thank Referee #3 for noting the lack of additional controls in Figure 5F. We have included additional housekeeping genes, to normalize the qPCR results shown in the revised Figure 5F. The expression of the three housekeeping genes (ACTB, LMNB1, TBP) was assessed at high and low densities for the five cell lines and the results reported in Figure S5B-D. The mean values for the mRNA levels of the three housekeeping genes were then used to normalize SCD expression. These new results reported in Figure 5F confirm the cell density-dependent regulation of SCD in the TNBC cell lines. In addition to the SCD protein expression showed for the MDA MB 468 in the original Figure 5H, as requested by Referee#3, we performed Western blot analysis for SCD in the remaining four cell lines reported in Figure 5F (BT549, CAL 120, MDA MB 231, MCF7). The results included in the revised Figure 5G confirm the cell density-dependent regulation of SCD at the protein level.

Regarding the additional band above SCD noted by Referee#3 in Figure 4E and not evident in the original Figure 5H (corresponding to revised Figure 5G), we can reassure them that the same antibody (Alpha Diagnostics, #SCD11-A, 1:1000) was used for the detection of SCD in these two blots. For transparency we provided uncropped western blot images of these analysis in the Figure 4 for Referee#3, below.

In the WBs performed for the revision experiments (including those requested by Referee#3 for the four cell lines (BT549, CAL 120, MDA MB 231, MCF7) shown in the revised Figure 5) we used a different SCD antibody (Abcam, 1:1000 # ab19862), as stated in the figure legend and in the 'Immunoblotting' section of the Methods.

Figure 4 for Referee#3. Uncropped Western blot (WB) images of the bands corresponding to SCD protein reported in Figure 4E and Figure 5G (originally submitted as Figure 5H) for the MDA MB 468 cells. The green rectangles highlight the bands reported in the respective figure panels. The SCD antibody used for both WB was from Alpha Diagnostics, #SCD11-A and the dilution used was 1:1000.

13. The authors should be more accurate with their wording around "loss of SCD" and "downregulation" of genes at low cell density. Their data show differences in expression, but which is "normal" and which is up or down remains to be determined. Critically, how these data have any bearing on tumours in vivo - where 2D cell density is not a factor - have not been determined in this study (see comment below about serum analysis).

We agree with Referee#3 that is difficult to identify a 'normal' reference when comparing high and low densities and the language should reflect this uncertainty. Accordingly we revised the relevant Manuscript text to "we assessed the levels of SCD mRNA and protein expression and found that it was proportional to cell density in all four TNBC lines tested (Figure 5F-G)".

In the revised manuscript the use of terms such as "loss of SCD" is restricted to the SCDko cells.

To strengthen the relevance to human tumour biology of our cell density data we mined online databases (ctcRbase, doi: 10.1093/database/baaa020) and found that SCD expression is indeed lower in metastasis and circulating tumour cells from breast cancer patients, compared to the primary tumours. These new

results presented below in the Figure 5 for Referee#3 suggest that the pattern of SCD expression observed in primary breast tumour compared to circulating and metastatic cells resemble the pattern observed in low- and high-density cultures. These data can also be included in the revised manuscript upon reviewer request.

Figure 5 for Referee#3. Relative expression of SCD in breast cancer patient-derived samples from primary tumours, metastatic lesions and circulating tumour cells. The data was analysed by and obtained from ctcRbase, doi: 10.1093/database/baaa020. Whiskers represent highest and lowest value. Boxes span from first quartile values to third quartile values with line representing the median of the expression level.

14. The focus on SCD as "critical" is somewhat diminished by the findings around lipid peroxides in Figure 5F. For a paper on ferroptosis, further assessment of lipid peroxidation would be expected to ensure ferroptosis is indeed responsible for these phenotypes.

We agree with Referee #3 that the results in Figure 5F (corresponding to the revised Figure 5E) do not support a mechanistic model where SCD is critical to directly detoxify lipid peroxidation.

In line with our results, the exogenous supplementation with the SCD-product, oleic acid, does not significantly change the cellular levels of lipid peroxidation as assessed by FACS with the C11 BODIPY probe (PMID: 30686757). To our knowledge the current literature does not yet provide an accepted mechanism by which MUFAs could directly decrease lipid peroxides and consequently BODIPY oxidation.

However, our new data presented in Figure 5A-B show that SCDko cells are sensitized to selenium starvation and GPX4 inhibition compared to wild type cells. The revised Figure 5A also show that this defect can be at least in part rescued by Ferrostatin-1 and deferoxamine, demonstrating that SCD has an anti-ferroptotic role in TNBC cells that is not mediated by the direct dumping of lipid peroxides.

15. Figure 6C should be repeated with the other two KO lines from Figure 6B.

We addressed the request from Referee#3 and repeated the assay in Figure 6C with the sgSEPSECS cells. The new results reported in the revised Figure 6C are consistent with the results reported in the original manuscript. Unfortunately for technical reasons we could not perform the requested experiment in the sgPSTK cells where a functional decrease in the levels of the targeted gene was more difficult to achieve, as also indicated by the WB in Figure 6B. For these technical reasons, we decided to further validate the relevance of the two targets SEPHS2 and SEPSECS with new *in vivo* experiments that strengthened the validity of our original conclusions.

Please refer to points 16 and 17 below where these experiments are described in more details.

16. Tumour experiments in Figure 6 are essentially a single experiment with just 3 mice per arm. There is always variation in tumour take rate in individual animals, with these data likely to be underpowered to make the current conclusions. Particularly where the two groups are subject to individual cell counts, and a small variation in the number of injected cells could lead to changes in tumour growth. In addition, one of the control mice was subject to a "technical issue" which meant only a single tumour was implanted, which may affect the tumour size if the two tumours secrete factors that enhance/support their growth. Out of interest, it should be stated whether the "two tumours derived from a single injection" appeared in the mouse with only 1 injection, as perhaps this was actually 2 injections in the same fat pad, which would totally change the result. Additional mice in a separate experiment would assist in ensuring these data are more robust and scientifically valid. This needs to be performed since there are no significant differences in tumour size for the *ex vivo* analysis.

We would like to thank Referee#3 for carefully assessing the results presented in the original Figure 6. Firstly, we would like to clarify to Referee#3 that the issue regarding the "two tumours derived from a single injection" happened in a mouse that was injected only once. This was confirmed by the technical staff that carried out the surgical procedure to inject the cells in the mammary fat pad of mice.

We agree with Referee#3 that this technical issue as well as the variability in the preparation of the cell lines for the injections could potentially impact the significance of the differences observed between experimental groups constituted by only 3 mice. Therefore, we have repeated this particular experiment by orthotopically implanting bilaterally NTC and sgSEPHS2 cells in 8 mice per group.

In one of the 8 mice of the NTC group the two tumours implanted bilaterally in the mammary fat pad started to fuse early on during tumour development, a condition breaching the licence for animal work. For this reason, the mouse was culled before the experimental endpoint and censored from the study. The experimental groups were constituted by 7 mice with 14 tumours for the NTC, and 8 mice and 16 tumours for the sgSEPHS2 group. In addition to the request from Referee#3 we decided to expand the experiment and test the effects of a second gene in the selenocysteine biosynthesis pathway, SESEPCS for which we obtained a functional knock down (revised Figure 6E). For this, 8 mice were bilaterally implanted in the mammary fat pad with sgSEPSECS cells. In one of these mice one of the tumours was impairing the movement of the leg, and the mouse was culled before the experimental endpoint and censored from the study. Therefore, this experimental group was constituted by 7 mice with 2 tumours each. The *in vivo* and *ex vivo* results from this new study, reported in the revised Figure 6E-G and Figure S6 D-E, demonstrate that the knockout of SEPHS2 and SESEPCS do not have a significant effect on the growth and size of mammary tumours.

In addition, the same cells lines used to grow tumours were injected in the tail vein of 8 mice per group mice and their capacity to metastasize in the lung was measured *in vivo* by a luciferase-based reporter, as

well as *ex vivo* by immunohistochemistry. One of the 8 mice in the SEPHS2 died before the experimental endpoint for husbandry reasons independent of the experimental procedures, and it was censored from the study.

The results from this new study assessing the luciferase signal and the number of individual cancer cells in the lung (revised Figure 6H-K and Figure S6C) demonstrate that the knock down of SEPHS2 and SESEPCS significantly decrease the metastatic burden in the lungs.

To reassure Referee#3 that the observed differences are not biased by '*cell counts and a small variation in the number of injected cells*' we measured the luciferase signal 1 hour after the cells injection and found no significant differences between the experimental groups (Figure S6C).

Overall, the new results obtained with SEPHS2 and SEPSECS ko substantially strengthen the proposed mechanistic model where TNBC cells that lose the anti-ferroptotic environment of high-density cultures, or primary tumours, are sensitized to the inhibition of selenocysteine synthesis. Interestingly our results show that the selenocysteine synthesis pathway, generally considered essential for survival, is redundant in TNBC mammary tumours, providing evidence that a therapeutic window for targeting selenocysteine synthesis in metastatic TNBC cells might exist, and is warrant further pre-clinical investigations.

The key *in vitro* and *in vivo* findings of this study are graphically summarized in the new Synopsis.

17. No details are provided as to why the experiment stopped at 35 days - was this a predetermined endpoint or an ethical endpoint? These are quite small tumours by normal xenograft ethical standards, which also makes these data hard to interpret. Figure 6I should include protein expression rather than just mRNA expression for the knockout to ensure the protein levels are indeed lower in the knockout post tumour growth.

We would like to thank Referee#3 for carefully assessing the experimental details of the animal studies. The experiment reported in the original Figure 6 F-I was stopped at day35 because of pre-determined endpoint, defined in the project licence for animal work. The endpoint for mammary tumours consists of a maximum linear dimension of 15mm. The technical staff members performing the caliper measurements noted that some tumours reaching 11 mm in maximal length, started to fuse with the contralateral tumours in the mammary fat pad. The fusion of the two tumours would have resulted in a single tumour longer than 15 mm. Therefore, for these mice the termination of the study at day 35 was a legal obligation under UK law. Therefore, we decided to synchronously stop the experiment at day 35 for all the mice, to compare the *ex vivo* tumour measurements at a consistent time point post injection.

However, we would like to remark to Referee#3 that all mouse studies and the relative figure panels presented in the original submission have been replaced. The new studies with the TNBC cells orthotopically injected, described in the point 16 above, were terminated 37 days post injection. The results from the new studies are reported in the revised Figure 6H-K and Figure S6C. The new results from the tail-vein injection studies are reported in the revised Figure 6H-K and Figure S6C.

Finally, as requested by Referee#3 the protein levels of SEPHS2 in the tumours extracts is reported in Figure S6D. The anti-SEPSECS antibody (#ab236956) that we tested did not produce a specific band in the Western blot analysis of the tumour extracts. We therefore used GPX4 levels as functional proxy for the levels of SEPSECS and reported those in Figure S6D left inset. The results obtained from Western blot analysis show that the sgSEPHS2 and sgSEPSECS cells used for injections expressed significantly lower

levels of SEPHS2 and GPX4 respectively, compared to the NTC controls (Figure 6E). However, the protein levels in the tumours at endpoint were comparable between the groups, indicating that tumour evolution selected against the expression of the single guides.

18. In the discussion it is stated: "Our results complement these findings and suggest that primary tumours, where cells are found at very high density, could secrete MUFA-containing lipids and alter the lipid composition in circulation thereby protecting metastasizing cells from ferroptosis." This should have been looked at by analysis of the mice with tumours compared to control serum for MUFA-containing lipids (and indeed compared to SCDko cells).

We would like to thank Referee#3 for suggesting these additional *in vivo* studies. We followed Referee#3 suggestions and looked at the levels of total fatty acids in the serum of mice with tumours. For this we sampled serum from mice before they were implanted with MDA-MB-468 cells NTC or SCDko and after 42 days from implantation when mammary tumours were at least 9 mm and 250 mm³. The results shown below in the Figure 6 for Referee#3 show that there were no significant differences in the levels of saturated fatty acids and MUFAs between pre and post tumour implantation. These results suggest that tumours do not significantly alter the circulating levels of the pool of total fatty acids.

In addition to these measurements, we assessed the levels of total fatty acids in the interstitial fluid obtained from NTC and SCDko tumours. The new results reported in the revised Figure 4H, K show that oleic acid and palmitoleic acid, two main products of SCD activity, are less abundant in the interstitial fluid from SCDko tumours compared to wt (Figure 4K).

These results strengthen the relevance of our cell culture-based findings for *in vivo* tumour biology, and show that lipids produced by SCD-expressing tumours are released in the extracellular tumour microenvironment.

Figure 6 for Referee#3. Relative levels of the total pool (free and lipid-bound) fatty acids in the serum of NSG mice, before and 42 days after the implantation of NTC and SCDko MDA-MB-468 cells in the mammary fat pad. Each data point represents one mouse and the lines indicate the variation between pre- and post-implantation. P values refer to a two-way ANOVA test for multiple comparisons.

8th May 2024

Dear Dr. Tardito,

Thank you for submitting your revised study, which was evaluated by the three initial reviewers. As you will see below, while referees #1 and #2 are satisfied with the revisions, referee #3 mentions several points that have not been satisfactorily addressed. After further discussion with my colleagues, we agreed that the remaining points could be addressed in an additional round of revisions.

In particular, please address points #12, #13 and #15 as advised by the referee. Point #17 could be addressed experimentally or by adequate discussion (in the rebuttal letter and in the manuscript).

Please note that the revised manuscript will once again be subjected to review, and we cannot guarantee a positive outcome at this stage.

Moreover, please address the following editorial requests:

We require:

4) A .docx formatted letter INCLUDING the reviewers' reports and your detailed point-by-point responses to their comments. As part of the EMBO Press transparent editorial process, the point-by-point response is part of the Review Process File (RPF), which will be published alongside your paper.

5) A complete author checklist, which you can download from our author guidelines (<https://www.embopress.org/page/journal/17574684/authorguide#submissionofrevisions>). Please insert information in the checklist that is also reflected in the manuscript. The completed author checklist will also be part of the RPF.

6) It is mandatory to include a 'Data Availability' section after the Materials and Methods. Before submitting your revision, primary datasets produced in this study need to be deposited in an appropriate public database, and the accession numbers and database listed under 'Data Availability'. Please remember to provide a reviewer password if the datasets are not yet public (see <https://www.embopress.org/page/journal/17574684/authorguide#dataavailability>).

7) For data quantification: please specify the name of the statistical test used to generate error bars and P values, the number (n) of independent experiments (specify technical or biological replicates) underlying each data point and the test used to calculate p-values in each figure legend. The figure legends should contain a basic description of n, P and the test applied. Graphs must include a description of the bars and the error bars (s.d., s.e.m.). Please provide exact p values.

8) Our journal encourages inclusion of *data citations in the reference list* to directly cite datasets that were re-used and obtained from public databases. Data citations in the article text are distinct from normal bibliographical citations and should directly link to the database records from which the data can be accessed. In the main text, data citations are formatted as follows: "Data ref: Smith et al, 2001" or "Data ref: NCBI Sequence Read Archive PRJNA342805, 2017". In the Reference list, data citations must be labeled with "[DATASET]". A data reference must provide the database name, accession number/identifiers and a resolvable link to the landing page from which the data can be accessed at the end of the reference. Further instructions are available at .

9) We replaced Supplementary Information with Expanded View (EV) Figures and Tables that are collapsible/expandable online. A maximum of 5 EV Figures can be typeset. EV Figures should be cited as 'Figure EV1, Figure EV2' etc... in the text and their respective legends should be included in the main text after the legends of regular figures.

- For the figures that you do NOT wish to display as Expanded View figures, they should be bundled together with their legends in a single PDF file called *Appendix*, which should start with a short Table of Content. Appendix figures should be referred to in

the main text as: "Appendix Figure S1, Appendix Figure S2" etc.

10) The paper explained: EMBO Molecular Medicine articles are accompanied by a summary of the articles to emphasize the major findings in the paper and their medical implications for the non-specialist reader. Please provide a draft summary of your article highlighting

11) For more information: There is space at the end of each article to list relevant web links for further consultation by our readers. Could you identify some relevant ones and provide such information as well? Some examples are patient associations, relevant databases, OMIM/proteins/genes links, author's websites, etc...

12) Author contributions: CRediT has replaced the traditional author contributions section because it offers a systematic machine readable author contributions format that allows for more effective research assessment. Please remove the Authors Contributions from the manuscript and use the free text boxes beneath each contributing author's name in our system to add specific details on the author's contribution. More information is available in our guide to authors.

13) Disclosure statement and competing interests: We updated our journal's competing interests policy in January 2022 and request authors to consider both actual and perceived competing interests. Please review the policy <https://www.embopress.org/competing-interests> and update your competing interests if necessary.

14) Every published paper now includes a 'Synopsis' to further enhance discoverability. Synopses are displayed on the journal webpage and are freely accessible to all readers. They include a short stand first (maximum of 300 characters, including space) as well as 2-5 one-sentences bullet points that summarizes the paper. Please write the bullet points to summarize the key NEW findings. They should be designed to be complementary to the abstract - i.e. not repeat the same text. We encourage inclusion of key acronyms and quantitative information (maximum of 30 words / bullet point). Please use the passive voice. Please attach these in a separate file or send them by email, we will incorporate them accordingly.

15) As part of the EMBO Publications transparent editorial process initiative (see our Editorial at <http://embomolmed.embopress.org/content/2/9/329>), EMBO Molecular Medicine will publish online a Review Process File (RPF) to accompany accepted manuscripts.

In the event of acceptance, this file will be published in conjunction with your paper and will include the anonymous referee reports, your point-by-point response and all pertinent correspondence relating to the manuscript. Let us know whether you agree with the publication of the RPF and as here, if you want to remove or not any figures from it prior to publication. Please note that the Authors checklist will be published at the end of the RPF.

I look forward to receiving your revised manuscript.

Yours sincerely,

Lise Roth

**** Reviewer's comments ****

Referee #1 (Comments on Novelty/Model System for Author):

As stated in during the first revision, the current study sums to the importance and growing recognition of the importance of selenocysteine metabolism to ferroptosis regulations and metastasis. The overall medical relevance is currently hard to extrapolate and the impact will not be immediate, given the lack of tools to modulates this pathway. Nevertheless this should not detract from the study.

Referee #1 (Remarks for Author):

The authors thoroughly addressed all the points raised during the first round of revision. I have nothing more to add at this stage, congratulations.

Referee #2 (Remarks for Author):

The authors provided additional data and the manuscript was adequately revised. I have no further comments.

Referee #3 (Remarks for Author):

The authors have comprehensively addressed the suggestions, resulting in a strengthened paper. However, a few points below remain and were not sufficiently addressed in their point-by-point response:

12 - Why did you change SCD antibodies, this change should be justified, or all blots redone with the same antibody.

13 - Please add the human SCD expression data to the manuscript.

15 - If the Western blot consistently shows sgPSTK knockdown, but no "functional decrease" - does that not suggest this gene is not functionally important? The authors should either include all data on sgPSTK, or if they do not believe the knockdown is working, all data on the spPSTK should be removed from the manuscript.

17 - The lack of a change in protein expression at the end of the in vivo experiment is troubling, although this appears to have only been examined in the orthotopic model? The fact that this occurs suggests either the knockout is not robust, or perhaps that there is a critical role for these genes in primary tumours resulting in selection against the knockout.

Is this selection against the knockout also occurring in the metastatic model?

Since the model/Synopsis figure suggests GPX4 is critical for the metastatic phenotype, GPX4 levels, and indeed SEPHS2 and SEPECS levels, should have been examined in the adjacent sections of the lungs for the IV model. This is a key mechanistic point of the authors, and if indeed there is selection against the knockout, how do they postulate the decreased metastatic effect occurs? This needs to be better examined and discussed in the manuscript. Since there was no MUFA secretion into the circulation, this aspect is critical for how the metastatic cells resist ferroptosis.

***** Reviewer's comments *****

Referee #1 (Comments on Novelty/Model System for Author):

As stated in during the first revision, the current study sums to the importance and growing recognition of the importance of selenocysteine metabolism to ferroptosis regulations and metastasis. The overall medical relevance is currently hard to extrapolate and the impact will not be immediate, given the lack of tools to modulates this pathway. Nevertheless this should not detract from the study.

Referee #1 (Remarks for Author):

The authors thoroughly addressed all the points raised during the first round of revision. I have nothing more to add at this stage, congratulations.

We would like to thank the Referee #1 for the constructive criticisms, suggestions and appreciation of our revised study.

Referee #2 (Remarks for Author):

The authors provided additional data and the manuscript was adequately revised. I have no further comments.

We would like to thank the Referee #2 for the constructive criticisms, suggestions and appreciation of our revised study.

Referee #3 (Remarks for Author):

The authors have comprehensively addressed the suggestions, resulting in a strengthened paper. However, a few points below remain and were not sufficiently addressed in their point-by-point response:

12 - Why did you change SCD antibodies, this change should be justified, or all blots redone with the same antibody.

We would like to thank Referee #3 for carefully assessing the methodology of our study. In brief, we found that both antibodies employed in Western blots presented in our study,

Abcam (#ab19862) and Alpha diagnostic (#SCD11-A), detect a specific band corresponding to the human SCD. However, the antibody from Alpha diagnostic has higher affinity compared to the Abcam one and is more suitable for the detection of SCD in cell lines that express low levels of the protein.

Initially, we used the antibody from alpha diagnostic to perform Western blots for SCD on cellular extracts. We validated the specificity of this antibody in SCDko MDA-MB-468 cells as shown in Figure 4E and in the Figure 7A for Referee#3 below (rightmost two bands). However, this antibody did not produce detectable signal when assessing the expression of SCD in tumour tissue from MDA-MB-468 xenografts (See Figure 7A for Referee#3 below). Moreover, the same Alpha diagnostic antibody showed low intensity bands for BT549, Cal120 and MDA-MB-231 cells cultured at low density, where the expression of SCD is low (See Figure 7B for Referee#3 below).

To improve the detection of SCD, particularly in the tumour samples, we tested a second anti-SCD antibody (Abcam, #ab19862). This antibody produced a clear and specific band for SCD in tumour tissue from MDA-MB-468 xenografts and the specific signal was absent in samples from the tumours generated with SCDko cells (Figure 4H). These results prompted us to re-assess the expression of SCD with the Abcam, #ab19862 antibody also in the BT549, CAL120, MDA-MB-231 that express low levels of the enzyme especially when cultured at low density. The MCF7 cells that express high levels of SCD were used as a reference. The results with the Abcam antibody are presented in Figure 5G. Additionally Figure 7B for Referee#3 show that the alpha diagnostic antibody produced comparable results in these cell lines, strengthening the conclusion that cell density regulates the expression of SCD in TNBC cells. Unfortunately, the protein yield in the extracts from cells cultured at low density is very low, allowing us to load only one gel for Western blot analysis with the samples collected from the 3 independent experiments presented. Therefore, we decided to show the results for the BT549, CAL120, MDA-MB-231 and MCF7 obtained with the Abcam antibody to obtain a more reliable quantification with the cell lines that express low levels of SCD when cultured at low density.

We are confident about the robustness of the results presented in the revised manuscript and, in fact validated with two different antibodies in multiple cell lines, supporting the key message that SCD expression is regulated by cell density in TNBC cells.

We hope that this explanation and the additional data provided in the Figure 7 for Referee#3 will convince them about the robustness of these findings and the integrity of our study.

Figure 7 for Referee#3. A) Western blot analysis of SCD levels assessed with the antibody Alpha Diagnostic, #SCD11-A (1:1000) in tissue samples from tumours derived from NTC and SCDko MDA-MB-468 xenografts. The lower band visible only in the NTC MDA-MB-468 cells, corresponds to the SCD specific signal. Vinculin was used as loading control.

B) Western blot analysis of SCD levels assessed with the antibody Alpha Diagnostic, #SCD11-A (1:1000) in extracts from BT549, CAL120, MDA-MB-231, MCF7 seeded at low and high density and cultured for 2 days. A representative image of the western blot for SCD and vinculin (loading control) from one of the 3 independent experiments quantified in the upper panel. P value refers to a two-tailed, homoscedastic Student's t tests for unpaired samples.

13 - Please add the human SCD expression data to the manuscript.

We thank Referee#3 for appreciating the value of the results obtained by mining patient databases. We have included the data on the expression of SCD in human samples to the revised Figure 5I. These results are described in the revised manuscript and the methods employed described in the Materials and Methods section.

15 - If the Western blot consistently shows sgPSTK knockdown, but no "functional decrease" - does that not suggest this gene is not functionally important? The authors should either include all data on sgPSTK, or if they do not believe the knockdown is working, all data on the sgPSTK should be removed from the manuscript.

We appreciate the point raised from Referee#3. We have tested an antibody against PSTK from Thermo Fisher Scientific that unfortunately did not produce reliable bands in Western blot analysis of cell extracts. For this reason, we decided to use GPX4 level as a proxy for PSTK activity. In 2 out of 5 independent experiments initiated by infecting MDA-MB-468 cells with lentivirus transducing single guide RNA against PSTK, the activity of PSTK was decreased as indicated by GPX4 levels (see Figure 6B and "Experiment A" in the Figure 8 for Referee#3 below). In these two experiments the interference with PSTK expression resulted in a functional effect, decreasing the colony formation efficiency of MDA-MB-468 cells (Figure 6D). This evidence suggests that PSTK plays an important functional role in survival of TNBC cells. However, for what we believe are technical reasons, an efficient depletion of PSTK levels is difficult to achieve by lentiviral infection.

Overall, we believe that the results regarding PSTK shown in Figure 6D, in the context of the results obtained with the other two key enzymes of selenocysteine tRNA biosynthesis (SEPHS2 and SEPSECS), add value to the revised manuscript. We agree that firm conclusions about the specific role of PSTK would require additional experiments. To reflect the lack of statistical significance for the data reported for PSTK we toned down the results description as follow:

“In these ferroptosis-priming conditions, the interference with SEPSECS and SEPHS2 ablates the protective effect of selenium, while Ferrostatin-1 retains its anti-ferroptotic effect. A similar trend was observed for PSTK interference, indicating that selenocysteine synthesis inhibition triggers ferroptosis selectively in TNBC cells cultured at low density (Figure 6D and EV4B).”

Figure 8 for Referee#3. Western blot images of GPX4 and vinculin (loading control) in MDA MB 468 cells after lentiviral infection with NTC, sgSEPHS2, sgSEPSECS or sgPSTK1 and supplemented with 50 nM Na₂SeO₃ (Se) as indicated. Four independent experiments (#A-D) are shown.

17 - The lack of a change in protein expression at the end of the in vivo experiment is troubling, although this appears to have only been examined in the orthotopic model? The fact that this occurs suggests either the knockout is not robust, or perhaps that there is a critical role for these genes in primary tumours resulting in selection against the knockout.

Is this selection against the knockout also occurring in the metastatic model?

Since the model/Synopsis figure suggests GPX4 is critical for the metastatic phenotype, GPX4 levels, and indeed SEPHS2 and SEPSECS levels, should have been examined in the adjacent sections of the lungs for the IV model. This is a key mechanistic point of the authors, and if indeed there is selection against the knockout, how do they postulate the decreased metastatic effect occurs? This needs to be better examined

and discussed in the manuscript. Since there was no MUFA secretion into the circulation, this aspect is critical for how the metastatic cells resist ferroptosis.

We would like to thank Referee#3 for these thoughtful questions. We agree with the Referee that the knockout of SEPHS2 and SEPSECS is not stable during the *in vivo* experiments and that there is selection against the loss of expression of SEPHS2 and SEPSECS in the primary tumours.

We would like to highlight that for all the experiments targeting selenocysteine biosynthesis, we use pools of cells freshly selected for puromycin resistance after infection with the lentivirus transducing Cas9 and sgRNAs. Just before the injection in mice the cell populations interfered for selenocysteine synthesis show clear reductions in the levels of SEPHS2 and GPX4 compared to NTC controls (Figure 6E). In this context, any residual GPX4 expression comes from a subpopulation of cells that aren't deleted for SEPHS2 or SEPSECS, as also pointed out by Referee#1 point 5. Therefore, it is reasonable to assume that within the mixed cell population implanted in the mammary fat pad, the fraction of wild type cells outcompetes over time the SEPHS2 or SEPSECS deficient cells, explaining the comparable levels of expression between interfered and control mammary tumours at endpoint.

In the metastatic model, upon tail vein injection of the same cell populations, the interference with SEPHS2 or SEPSECS caused a detectable difference in luciferase signal 9 days after injection (Figure 6H), and lead to a significant difference in the metastatic burden at experimental endpoint.

We followed Referee#3 suggestion and we performed IHC analysis with anti-SEPHS2 and anti-GPX4 antibodies in lung sections from the metastatic models. Unfortunately, the staining for neither of these antigens was convincingly specific in IHC. We therefore optimized an IHC staining for Cas9 that is encoded by the same plasmid as the sgRNAs. We observed a significantly lower number of Cas9-positive cells in lungs of animals injected with sgSEPSECS or sgSEPHS2 cells compared to NTC cells. This indicates that cells expressing SEPHS2 or SEPSECS and Cas9 are indeed counter-selected during the metastatic lung colonization (Figure 6K).

Our interpretation of the differential results obtained in mammary tumours and metastatic models is that the experiments performed with 14-16 mammary tumours / group are underpowered to appreciate the effect on tumour growth caused by the selective pressure against sgSEPSECS- or sgSEPHS2-expressing cells.

On the contrary in the metastatic model, the selective pressure against these cells is stronger due to the different microenvironment and causes a significantly lower metastatic burden in mice injected with selenocysteine interfered cells compared to controls (Figure 6 H-K).

Following Referee#3 requests the text of manuscript has been revised to make these considerations more explicit and acknowledge the limitations of these *in vivo* studies:

“However, compared to NTC controls the lower levels of SEPHS2 and GPX4 expression observed pre-implantation (Figure 6E) were restored in sgSEPHS2 or sgSEPSECS

tumours at endpoint (Figure EV4D-E), suggesting that the counterselection of the interfered cells did not cause a tumour growth delay appreciable with the experimental settings employed.”

“Overall, these in vivo results demonstrate that the expression of SEPHS2 and SEPSECS is less critical for mammary breast tumour growth than for breast cancer cells to survive in the bloodstream and colonize the lungs.”

“In line with the working model proposed, the deletion of SEPHS2 and SEPSECS does not affect the proliferation of TNBC cultured at high density, and a transient interference with their expression does not appreciably delay the growth of orthotopic mammary tumours.”

8th Aug 2024

Dear Dr. Tardito,

Thank you for submitting your revised study. We have now received the feedback from referee #3 who evaluated your revised manuscript, and as you will see below, this referee is satisfied with the revisions. I am therefore pleased to inform you that I will be able to accept your manuscript once the following minor comments are addressed:

1/ Manuscript text:

- Please remove the yellow highlights and only keep in track changes mode any new modification.
- Please provide up to 5 keywords.
- Methods:
 - o This section should be placed after the discussion.
 - o Methods should be described using our 'Structured Methods' format. The Methods section includes a Reagents and Tools Table followed by a Methods and Protocols section describing the methods using a step-by-step protocol format. More information on how to adhere to this format as well as a downloadable template (.docx) for the Reagents and Tools Table can be found in our author guidelines:
<https://www.embopress.org/page/journal/17574684/authorguide#structuredmethods>
 - o Table 1 should be moved after the figure legends.
 - o Statistics: please include a statement on inclusion/exclusion criteria.
- Data availability: thank you for depositing your data in a public repository. The data must be publicly available before acceptance of the manuscript. Please remove the sentences: "Source data files that support the findings of this study are stored at the Cancer Research UK Scotland Institute. Requests for unique biological materials can be made to the corresponding author."
- Author contributions: CRediT has replaced the traditional author contributions section because it offers a systematic machine readable author contributions format that allows for more effective research assessment. Please remove the Authors Contributions from the manuscript and use the free text boxes beneath each contributing author's name in our system to add specific details on the author's contribution.
- Please replace "Competing interests" by "Disclosure statement and competing interests" (<https://www.embopress.org/competing-interests>).
- References: an article was recently published in EMM (PMID: 38926633) that reports complementary findings to your study. We thus believe it should be cited in your article.

2/ Figures:

- The figures should be removed from the manuscript, figure legends should be compiled after the References and the Expanded View figure legends after Table 1.
- Re-use of cell images: there must have been a typo and we apologize for it, but we noticed a possible re-use of cell images in Fig 1C (not 1B as previously stated). Please check and clarify.
- Figures 2E/4F: Thank you for indicating the individual data points and the mean. Please remove the error bars.
- Please indicate the statistical test used for data analysis in the legends of figures 1b. Please note that the error bars are not defined in the legends of figure 1b.

3/ Source Data:

Thank you for providing source data. Please carefully check the data provided for Figure 5I.

4/ Checklist:

- New materials and reagents: do any restriction apply? If not, please indicate "Not applicable" in the left column and remove the DAS from the right column.
- Experimental study design and statistics, randomization: please fill in the left column

5/ Thank you for providing a synopsis text. Please remove it from the manuscript text and upload it as a related manuscript file. We note that you provided a nice cover suggestion. We'll get back to you shortly on that.

6/ As part of the EMBO Publications transparent editorial process initiative (see our Editorial at <http://embomolmed.embopress.org/content/2/9/329>), EMBO Molecular Medicine will publish online a Review Process File (RPF) to accompany accepted manuscripts.

We note that you agree with the publication of the RPF.

I look forward to receiving your revised manuscript.

With kind regards,

Lise Roth

**** Reviewer's comments ****

Referee #3 (Remarks for Author):

The authors have addressed all my concerns.

The authors addressed the remaining editorial issues.

4th Sep 2024

Dear Dr. Tardito,

Thanks you for submitting your revised files. I am pleased to inform you that your manuscript is accepted for publication and is now being sent to our publisher to be included in the next available issue of EMBO Molecular Medicine!

Please note that that while the reference "Lorito et al" was indeed added in the introduction, it is not included in the reference list. Please make sure to correct at proof stage.

With kind regards,

Lise Roth
